# How far can the statistical error estimation problem be closed by collocated data?

Annika Vogel[1,2] and Richard Ménard[1]

[1]Air Quality Research Division, Environment and Climate Change Canada (ECCC), Dorval - QC, Canada
[2]Rhenish Institute for Environmental Research (RIU) at the University of Cologne, Cologne, Germany

**Correspondence:** Annika Vogel (annika.vogel@ec.gc.ca)

**Abstract.** Accurate specification of error statistics required for data assimilation remains an ongoing challenge, partly because their estimation is an underdetermined problem that requires statistical assumptions. Even with the common assumption that background and observation errors are uncorrelated, the problem remains underdetermined. One natural question that could arise is: Can the increasing amount of overlapping observations or other datasets help to reduce the total number of statistical assumptions, or do they introduce more statistical unknowns? In order to answer this question, this paper provides a conceptual view on the statistical error estimation problem for multiple collocated datasets, including a generalized mathematical formulation, an illustrative demonstration with synthetic data, as well as guidelines for setting up and solving the problem. It is demonstrated that the required number of statistical assumptions increases linearly with the number of datasets. However the number of error statistics that can be estimated increases quadratically, allowing for an estimation of an increasing number of error cross-statistics between datasets for more than three datasets. The presented generalized estimation of full error covariance- and cross-covariance matrices between datasets does not necessarily accumulate uncertainties of assumptions among error estimations of multiple datasets.

## 1 Introduction

Accurate specification of error statistics used for data assimilation has been an ongoing challenge. It is known that the accuracy of both background and observation error covariances have a strong impact on the performance of atmospheric data assimilation (e.g., Daley, 1992a, b; Mitchell and Houtekamer, 2000; Desroziers et al., 2005; Li et al., 2009). A number of approaches to estimate optimal error statistics make use of residuals, i.e. the innovations between observation and background states in observation space (Tandeo et al., 2020), but the error estimation problem remains underdetermined. Different approaches exist which aim at closing the error estimation problem, all of which rely on various assumptions. For example, error variances and correlations were estimated a-posteriori by Tangborn et al. (2002); Ménard and Deshaies-Jacques (2018); Voshtani et al. (2022) based on cross-validation of the analysis with independent observations withheld from the assimilation. However, these a-posteriori methods require an iterative calculation of the analysis and the global minimization criterion provides only spatial-mean estimates of optimal error statistics. In recent years, the amount of available datasets has increased rapidly, including overlapping

or collocated observations from several measurements systems. This raises the question whether multiple overlapping datasets can be used to estimate full spatial fields of optimal error statistics a-priori.

Outside of the field of data assimilation, two different methods were developed that allow for a statistically optimal estimation of scalar error variances for fully collocated datasets. Although similar, these two methods were developed independently from each other in different scientific fields. One method, called the three cornered hat (3-CH) method, is based on Grubbs (1948) and Gray and Allan (1974) who developed an estimation method for error variances of three datasets based on their residuals. This method has been widely used in physics for decades, but has only recently been exploited in meteorology (e.g., Anthes and Rieckh, 2018; Rieckh et al., 2021; Kren and Anthes, 2021; Xu and Zou, 2021). Nielsen et al. (2022) and Todling et al. (2022) were the first to independently use the generalized 3CH (G3CH) method to estimate full error covariances matrices. Todling et al. (2022) used a modification of the G3CH method to estimate the observation error covariance matrix in a data assimilation framework. They show that when the G3CH method is applied to the observations, background and analysis of variational assimilation procedures, this particular error estimation problem can only be closed under the assumption that the analysis is optimal.

Independent from these developments, Stoffelen (1998) used three collocated datasets for multiplicative calibration w.r.t. each other. Following this idea, the triple collocation (TC) method became a well-known tool to estimate scalar error variances from residual statistics in the fields of hydrology and oceanography (e.g., Scipal et al., 2008; McColl et al., 2014; Sjoberg et al., 2021). Up to now, there are only a few applications of scalar error variance estimation in data assimilation with the TC method (e.g., Crow and van den Berg, 2010; Crow and Yilmaz, 2014). The 3CH and TC methods use different error models leading to slightly different assumptions and formulations of error statistics. A detailed description, comparison and evaluation of the two methods is given in Sjoberg et al. (2021). Both methods have in common that they require fully spatio-temporally collocated datasets with random errors. These errors are assumed to be independent among the realizations of each dataset with common error statistics across all realizations (e.g., Zwieback et al., 2012; Su et al., 2014). In addition, error statistics of the three datasets are assumed to be pairwise independent, which is the most critical assumption of these methods (Pan et al., 2015; Sjoberg et al., 2021).

While the estimation of three error variances has been well-established for decades, recent developments propose different approaches to extend the method to a larger number of datasets. As observed e.g. by Su et al. (2014); Pan et al. (2015); Vogelzang and Stoffelen (2021), the problem of error variance estimation from pairwise residuals becomes overdetermined for more than three datasets. Su et al. (2014); Anthes and Rieckh (2018); Rieckh et al. (2021) averaged all possible solutions of each error variance which reduces the sensitivity of the error estimates to inaccurate assumptions. Pan et al. (2015) clustered their datasets into structural groups and performed a two-step estimation of the in-group errors and the mean errors of each group, which were assumed to be independent. Zwieback et al. (2012) were the first to propose the additional estimation of the scalar error cross-variances between two selected datasets (which they denote as covariances) instead of solving an overdetermined system. This extended collocation (EC) method was applied to scalar soil moisture datasets by Gruber et al. (2016) who estimated one cross-variance in addition to the error variances of four datasets. Also for four datasets, Vogelzang and Stoffelen (2021) demonstrated the ability to estimate two cross-variances in addition to the error variances. They observed

that the problem can not be solved for all possible combinations of cross-variances to be estimated. However, their approach failed for five dataset due to a missing generalized condition which is required to solve the problem.

This demonstrates that the different approaches available for more than three datasets provide only an incomplete picture of the problem, where each approach is tailored to the specific conditions of the respective application. Aiming for a more general analysis, this paper approaches the problem from a conceptual point-of-view. The main questions to be answered are: How many error statistics can be extracted from residual statistics between multiple collocated datasets? How many statistics remain to be assumed? How do inaccuracies in assumed error statistics affect different estimations of error statistics? And what are the general conditions to set up and solve the problem?

In order to answer these questions, the general framework of the estimation problem which builds the basis for the remaining sections is introduced in Sect. 2. It provides a conceptual analysis of the general problem w.r.t. the number of knowns and unknowns and the minimum number of assumptions required. Based on this, the mathematical formulation for non-scalar error matrices is derived in Sect. 3 and Sect. 4. The derivation is based on the formulation of residual statistics as function of error statistics which is introduced in Sect. 3.2. While the exact formulation for estimating error statistics in Sect. 3.3 remain underdetermined in real applications, approximate formulations which provide a closed system of equations are derived in Sect. 4. Some relations presented in these two sections were already formulated previously for scalar problems dealing with error variances only. However, we present formulations for full covariance matrices including off-diagonal covariances between single elements of the state-vector of the respective dataset, as well as cross-covariance matrices between different datasets. Overlap to previous studies is mainly restricted to the formulation for three datasets in Sect. 4.1 and noted accordingly. Based on this, Sect. 4.2 provides a new approach for the estimation of error statistics of all additional datasets which uses a minimal number of assumptions. The theoretical formulations are applied to four synthetic datasets in Sect. 5. It demonstrates the general ability to estimate full error covariances and cross-statistics as well as effects of inaccurate assumptions w.r.t different setups. The theoretical concept proposed in this study is summarized in Sect. 6. This summary aims at providing the most important results in a general context; answering the main research questions of this study without requiring knowledge of the full mathematical theory. It includes the formulation and illustration of rules to solve the problem for an arbitrary number of datasets and provides guidelines for the setup of those. Finally, Sect. 7 concludes the findings and discusses consequences of using the proposed method in the context of high-dimensional data assimilation.

## 2  General framework

Suppose a system of $I$ spatio-temporally collocated datasets, which may include various model forecasts, observations, analyses and any other datasets available in the same state space. The 2nd moment statistics of the random errors of this system (with respect to the truth) can be described by $I$ error covariances w.r.t. each dataset and $N_I$ error cross-covariances w.r.t. each pair of different datasets. In a discrete state space, (cross-)covariances are matrices and the cross-covariance of dataset A and B is the transposed of the cross-covariance of B and A (see Sect. 3.1 for an explicit definition). Considering this equivalence,

the number $N_I$ of error cross-covariances between all different pairs of datasets is:

$$N_I = \sum_{i=1}^{I-1} i = \frac{1}{2} \cdot I \cdot (I-1) \tag{1}$$

Thus, the total number $U_I$ of error statistics (error covariances and cross-covariances) is:

$$U_I = N_I + I = \frac{1}{2} \cdot I \cdot (I+1) \tag{2}$$

While error statistics w.r.t. the truth are usually unknown in real applications, residual covariances can be calculated from the residuals between each pair of different datasets. The main idea now is to express the known residual statistics as functions of unknown error statistics (Sect. 3.2) and combine these equations to eliminate single error statistics (Sect. 3.3, Sect. 4). Because of $j \neq i$ for residuals, each of the $I$ datasets can be combined with all other $I-1$ datasets. As residual statistics also do not change with the order of datasets in the residual (see Sect. 3.1), the number of known statistics of the system is also given by

$N_I$ as defined in Eq. (1). It will be shown in Sect. 3.2.3 that residual cross-covariances contain generally the same information as residual covariances; thus the $N_I$ residual statistics can be given in form of residual covariances or cross-covariances.

Because $N_I$ residual statistics are known, $N_I$ of the $U_I$ error statistics can be estimated and the remaining $I$ have to be assumed in order to close the problem. The set of error statistics to be estimated can generally be chosen according to the specific application, but it will be shown that there are some constraints. Based on the mathematical theory provided in the following

sections, Sect. 6.1 provides guidelines which ensure the solvability of the problem for a minimal number of assumptions.

In most applications of geophysical datasets like in data assimilation, the estimation of error covariances is highly crucial while their error cross-covariances are usually assumed to be negligible. Given the greater need to estimate the $I$ error covariances, the remaining number of error cross-covariances which can be additionally estimated $A_I$ is:

$$A_I = N_I - I = \frac{1}{2} \cdot I \cdot (I-3) \tag{3}$$

The relation between the number of datasets, residual covariances, assumed and estimated error statistics is visualized in Fig. 1. $I = 0$ represents the mathematical extension of the problem, where no error- and residual statistics are required when no dataset is considered. For less than three datasets ($0 < I < 3$), $A_I$ is negative because the number of (known) residual covariances is smaller than the number of (unknown) error covariances ($N_I < I$) and thus the problem is underdetermined even when all datasets are assumed to be independent (=zero error cross-covariances). As in the case in data assimilation of

two datasets ($I = 2$), additional assumptions on error statistics are required. The same holds when only one dataset is available ($I = 1$), where the error covariance of this dataset remains unknown because no residual covariance can be formed. For three datasets ($I = 3$), $A_I$ is zero meaning that the problem is fully determined under the assumption of independent errors ($N_I = I$, formulated in Sect. 4.1).

For more than three datasets ($I > 3$), the number of (known) residual covariances exceeds the number of error covariances

which would lead to an overdetermined problem assuming independence among all datasets. Instead of solving an overdetermined problem, the additional information can be used to calculate some error cross-covariances (formulated in Sect. 4.2).

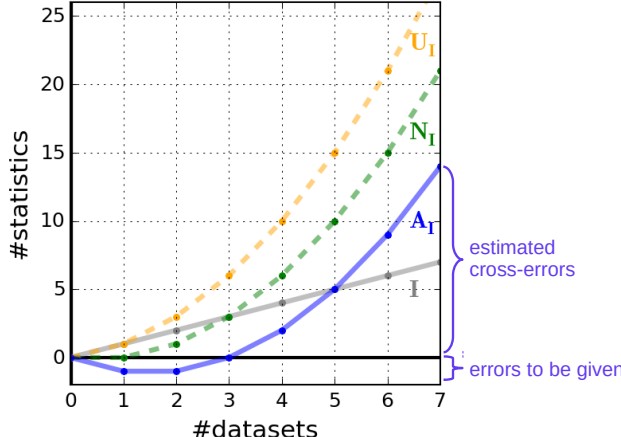

**Figure 1.** Relation between different numbers of statistics (covariances and cross-covariances) as function of the number of datasets. Shown are $I$ in solid gray (#datasets, #error covariances, #required assumptions), $U_I$ in dashed orange (#error statistics), $N_I$ in dashed green (#residual covariances, #error dependencies, #estimated error statistics), and $A_I$ in solid blue (#estimated error cross-covariances).

In other words, for $I > 3$ not all datasets need to be assumed to be independent; where $A_I$ gives the number of error cross-covariances which can be estimated in addition to the error covariances from all datasets. For example, half of the error cross-covariances can be estimated for $I = 5$ $\left(\frac{A_5}{N_5} = \frac{5}{10}\right)$, while two-thirds of them can be estimated for $I = 7$ $\left(\frac{A_7}{N_7} = \frac{14}{21}\right)$.

Although the relative amount of error cross-covariances which can be estimated increases with the number of datasets, an increasing number of $U_I - N_I = I$ assumptions – equal to the number of datasets – is required in order to close the problem because of $U_I > N_I , \forall I > 0$.

Note that almost all numbers presented above apply to the general case, where any combination of error covariances and cross-covariances may be given or assumed. While the interpretation of the numbers $I$, $N_I$ and $U_I$ remains the same in all

cases, the only difference is the interpretation of $A_I$ which is less meaningful when also error covariances are assumed.

## 3    Mathematical theory: Exact formulation

This section gives the theoretical formulation for exact statistical formulations of complete error covariance- and cross-covariance matrices from fully spatio-temporally collocated datasets. Similar to the 3CH method, the errors are assumed to be random, independent among different realizations, but with common error statistics for each dataset. The notation is introduced

in Sect. 3.1. While the true state and thus error statistics w.r.t. the truth are usually unknown, residual statistics can be calculated from residuals between each pair of datasets. At the same time, residual statistics contain information about error statistics of the datasets involved. The expression of residual statistics as function of error covariances and cross-covariances in Sect. 3.2 provides the basis for the subsequent mathematical theory. Based on these forward relations, inverse relations describe error statistics as function of residual statistics. The general equations of inverse relations are given in Sect. 3.3 which result in a

highly underdetermined system of equations. Closed formulations of error statistics for three and more datasets under certain assumptions will be formulated in the subsequent Sect. 4.

This first part of the mathematical theory includes the following new elements: (i) the separation of cross-statistics into a symmetric error dependency and an error asymmetry (Sect. 3.1), (ii) the general formulation of residual statistics as function of error statistics (Sect. 3.2.1 and 3.2.2), (iii) the demonstration of equivalence between residual covariances and cross-covariances (Sect. 3.2.3), (iv) the general formulation of exact relations between residual- and error statistics (Sect. 3.3).

## 3.1 Notation

Suppose $I$ datasets, each containing $R$ realizations of spatio-temporally collocated state vectors $\boldsymbol{x}_i$, $\forall\, i \in [1, I]$. Without loss of generality, the following formulation uses unbiased state vectors with zero mean. In practice, each index $i,j,k,l$ may represent any geophysical dataset like model forecasts, climatologies, in-situ- or remote sensing observations, or other datasets.

Let $\boldsymbol{\Gamma}_{i;j;k;l}$ be the *residual cross-covariance matrix* between dataset residuals $i-j$ and $k-l$ with $j \neq i$ and $l \neq k$, where each element $(p, q)$ is given by the expectation over all realizations:

$$\boldsymbol{\Gamma}_{i;j;k;l}(p, q) := \overline{\left[ \boldsymbol{x}_i(p) - \boldsymbol{x}_j(p) \right] \left[ \boldsymbol{x}_k(q) - \boldsymbol{x}_l(q) \right]} \tag{4}$$

and the *error cross-covariance matrix* $\mathbf{X}_{\widetilde{i};\widetilde{j}}$ between the errors of two datasets $i$ and $j$ w.r.t. the true state $\boldsymbol{x}_T$:

$$\mathbf{X}_{\widetilde{i};\widetilde{j}}(p, q) := \overline{\left[ \boldsymbol{x}_i(p) - \boldsymbol{x}_T(p) \right] \left[ \boldsymbol{x}_j(q) - \boldsymbol{x}_T(q) \right]} \tag{5}$$

where the tilde above a dataset index indicates its deviation from the truth and the overbar denotes the expectation over all $R$ realizations. Note that $x_i(p)$ is a scalar element of the dataset vector.

In the symmetric case, each element $(p, q)$ of the *residual covariance matrix* of $i-j$ with $j \neq i$, is given by:

$$\boldsymbol{\Gamma}_{i;j}(p, q) := \boldsymbol{\Gamma}_{i;j;i;j}(p, q) \overset{(4)}{=} \overline{\left[ \boldsymbol{x}_i(p) - \boldsymbol{x}_j(p) \right] \left[ \boldsymbol{x}_i(q) - \boldsymbol{x}_j(q) \right]} \tag{6}$$

and the *error covariance matrix* $\mathbf{C}_{\widetilde{i}}$ of a dataset $i$ w.r.t. the true state $\boldsymbol{x}_T$:

$$\mathbf{C}_{\widetilde{i}}(p, q) := \mathbf{X}_{\widetilde{i};\widetilde{i}}(p, q) \overset{(5)}{=} \overline{\left[ \boldsymbol{x}_i(p) - \boldsymbol{x}_T(p) \right] \left[ \boldsymbol{x}_i(q) - \boldsymbol{x}_T(q) \right]} \tag{7}$$

where numbers in parenthesis above an equal sign indicate other equations that were used to retrieve the right hand side.

Note that residual- and error cross-covariance matrices are generally asymmetric in the non-scalar formulation presented here, but the following relations hold for residual as well as similarly for error cross-covariance matrices:

$$\boldsymbol{\Gamma}_{i;j;k;l} \overset{(4)}{=} -\boldsymbol{\Gamma}_{j;i;k;l} \overset{(4)}{=} -\boldsymbol{\Gamma}_{i;j;l;k} \overset{(4)}{=} \boldsymbol{\Gamma}_{j;i;l;k} \tag{8}$$

$$\boldsymbol{\Gamma}_{i;j;k;l} \overset{(4)}{=} \left[ \boldsymbol{\Gamma}_{k;l;i;j} \right]^T \tag{9}$$

$$\mathbf{X}_{\widetilde{i};\widetilde{j}} \overset{(5)}{=} \left[ \mathbf{X}_{\widetilde{j};\widetilde{i}} \right]^T \tag{10}$$

The symmetric properties of residual- and error covariances follow directly from their definition:

$$\mathbf{\Gamma}_{i;j} \overset{(6)}{=} \mathbf{\Gamma}_{j;i} \tag{11}$$

$$\left[\mathbf{\Gamma}_{i;j}\right]^T \overset{(6)}{=} \mathbf{\Gamma}_{i;j} \tag{12}$$

The sum of an (asymmetric) cross-covariance matrix and its transposed is denoted as *dependency*. For example, the sum of error cross-covariance matrices between $i$ and $j$ is denoted as *error dependency matrix* $\mathbf{D}_{\widetilde{i};\widetilde{j}}$:

$$\mathbf{D}_{\widetilde{i};\widetilde{j}} := \mathbf{X}_{\widetilde{i};\widetilde{j}} + \mathbf{X}_{\widetilde{j};\widetilde{i}} \tag{13}$$

Although error cross-covariances may be asymmetric, the error dependency matrix is symmetric by definition:

$$\mathbf{D}_{\widetilde{i};\widetilde{j}} \overset{(13)}{=} \mathbf{X}_{\widetilde{i};\widetilde{j}} + \mathbf{X}_{\widetilde{j};\widetilde{i}} \overset{(13)}{=} \mathbf{D}_{\widetilde{j};\widetilde{i}} \tag{14}$$

$$\mathbf{D}_{\widetilde{i};\widetilde{j}} \overset{(13)}{=} \mathbf{X}_{\widetilde{i};\widetilde{j}} + \mathbf{X}_{\widetilde{j};\widetilde{i}} \overset{(10)}{=} \left[\mathbf{X}_{\widetilde{j};\widetilde{i}}\right]^T + \left[\mathbf{X}_{\widetilde{i};\widetilde{j}}\right]^T \overset{(13)}{=} \left[\mathbf{D}_{\widetilde{i};\widetilde{j}}\right]^T \tag{15}$$

Likewise, the sum of the residual cross-covariance matrices between $i-j$ and $k-l$ with $j \neq i$ and $l \neq k$, is denoted as *residual dependency matrix* $\mathbf{D}_{i;j;k;l}$:

$$\mathbf{D}_{i;j;k;l} := \mathbf{\Gamma}_{i;j;k;l} + \mathbf{\Gamma}_{k;l;i;j} \tag{16}$$

The difference between a cross-covariance matrix and its transposed is a measure of asymmetry in the cross-covariances and is therefore denoted as *asymmetry*. For example, difference between the error cross-covariance matrices between $i$ and $j$ is denoted as *error asymmetry matrix* $\mathbf{Y}_{\widetilde{i};\widetilde{j}}$:

$$\mathbf{Y}_{\widetilde{i};\widetilde{j}} := \mathbf{X}_{\widetilde{i};\widetilde{j}} - \mathbf{X}_{\widetilde{j};\widetilde{i}} \tag{17}$$

Likewise, the difference between the residual cross-covariance matrices between $i-j$ and $k-l$ with $j \neq i$ and $l \neq k$, is denoted as *residual asymmetry matrix* $\mathbf{Y}_{i;j;k;l}$:

$$\mathbf{Y}_{i;j;k;l} := \mathbf{\Gamma}_{i;j;k;l} - \mathbf{\Gamma}_{k;l;i;j} \tag{18}$$

## 3.2  Residual statistics

For real geophysical problems, the available statistical information are (i) residual covariance matrices of each pair of datasets and (ii) residual cross-covariance matrices between different residuals of datasets. The forward relations of residual covariances and residual cross-covariances as functions of error statistics are formulated in the following. For the estimation of error statistics, it is important to quantify the number of independent input statistics which determines the number of possible error estimations. Therefore, this section also includes an evaluation of the relation between residual cross-covariances and residual covariances in order to specify the additional information content of residual cross-covariances.

### 3.2.1 Residual covariances

Each element $(p, q)$ of the residual covariance matrix between two input datasets $i$ and $j$ can be written as function of their error statistics as follows:

$$\boldsymbol{\Gamma}_{i;j}(p,q) \overset{(6)}{=} \overline{\left\{ \left[ \boldsymbol{x}_i(p) - \boldsymbol{x}_T(p) \right] - \left[ \boldsymbol{x}_j(p) - \boldsymbol{x}_T(p) \right] \right\} \left\{ \left[ \boldsymbol{x}_i(q) - \boldsymbol{x}_T(q) \right] - \left[ \boldsymbol{x}_j(q) - \boldsymbol{x}_T(q) \right] \right\}}$$
$$\overset{(5),(7)}{=} \mathbf{C}_{\widetilde{i}}(p,q) - \mathbf{X}_{\widetilde{i};\widetilde{j}}(p,q) - \mathbf{X}_{\widetilde{j};\widetilde{i}}(p,q) + \mathbf{C}_{\widetilde{j}}(p,q) \tag{19}$$

Thus the complete residual covariance matrix of $i - j$ is expressed as:

$$\boldsymbol{\Gamma}_{i;j} \overset{(19)}{=} \underbrace{\mathbf{C}_{\widetilde{i}} + \mathbf{C}_{\widetilde{j}}}_{\text{"independent residual"}} - \underbrace{\left[ \mathbf{X}_{\widetilde{i};\widetilde{j}} + \mathbf{X}_{\widetilde{j};\widetilde{i}} \right]}_{\text{"error dependency"} =: \mathbf{D}_{\widetilde{i};\widetilde{j}}} \tag{20}$$

Equation (20) is an exact formulation of the complete residual covariance matrix of any pair of datasets $i - j$. It holds for all combinations of datasets without any further assumption like independent errors. Thus, the residual covariance of any dataset pair consists of (i) the independent residual associated with sum of the error covariances of each dataset, minus (ii) the error dependency corresponding to the sum of their error cross-covariances.

Note that although the error dependency matrix is symmetric by definition, it is the sum of two error cross-covariances which are generally asymmetric and thus differ in the non-scalar formulation. In the scalar case, the two error cross-covariances reduce to their common error cross-variance and the residual covariance reduces to the scalar formulation of the variance as e.g. in Anthes and Rieckh (2018); Sjoberg et al. (2021).

### 3.2.2 Residual cross-covariances

Each element $(p, q)$ of the residual cross-covariance matrix between two input datasets $i - j$ and $k - l$ can be written as function of their error cross-covariances:

$$\boldsymbol{\Gamma}_{i;j;k;l}(p,q) \overset{(4)}{=} \overline{\left\{ \left[ \boldsymbol{x}_i(p) - \boldsymbol{x}_T(p) \right] - \left[ \boldsymbol{x}_j(p) - \boldsymbol{x}_T(p) \right] \right\} \left\{ \left[ \boldsymbol{x}_k(q) - \boldsymbol{x}_T(q) \right] - \left[ \boldsymbol{x}_l(q) - \boldsymbol{x}_T(q) \right] \right\}}$$
$$\overset{(5)}{=} \mathbf{X}_{\widetilde{i};\widetilde{k}}(p,q) - \mathbf{X}_{\widetilde{i};\widetilde{l}}(p,q) - \mathbf{X}_{\widetilde{j};\widetilde{k}}(p,q) + \mathbf{X}_{\widetilde{j};\widetilde{l}}(p,q) \tag{21}$$

And thus the complete residual cross-covariance matrix between $i - j$ and $k - l$:

$$\boldsymbol{\Gamma}_{i;j;k;l} \overset{(21)}{=} \mathbf{X}_{\widetilde{i};\widetilde{k}} - \mathbf{X}_{\widetilde{i};\widetilde{l}} - \mathbf{X}_{\widetilde{j};\widetilde{k}} + \mathbf{X}_{\widetilde{j};\widetilde{l}} \tag{22}$$

Equation (22) is a generalized form of Eq. (20) with residuals between different datasets $(i - j; k - l)$. It consists of four error cross-covariances of the datasets involved. This formulation of residual statistics as function of error statistics provides the basis for the complete theoretical derivation of error estimates in this study. In contrast to the symmetric residual covariance matrix, the residual cross-covariance matrix may be asymmetric for asymmetric error cross-covariances.

### 3.2.3 Relation of residual statistics

In the following, it is demonstrated that combinations of residual cross-covariances contain the same statistical information as residual covariance matrices.

For $k = i$, the residual dependency between $i - j$ and $i - l$ can be expressed as combination of three residual covariances:

$$
\begin{aligned}
\mathbf{\Gamma}_{i;l} + \mathbf{\Gamma}_{j;i} - \mathbf{\Gamma}_{j;l} &\overset{(6)}{=} \overline{\left[\boldsymbol{x}_i(p) - \boldsymbol{x}_l(p)\right]\left[\boldsymbol{x}_i(q) - \boldsymbol{x}_l(q)\right]} + \overline{\left[\boldsymbol{x}_j(p) - \boldsymbol{x}_i(p)\right]\left[\boldsymbol{x}_j(q) - \boldsymbol{x}_i(q)\right]} - \overline{\left[\boldsymbol{x}_j(p) - \boldsymbol{x}_l(p)\right]\left[\boldsymbol{x}_j(q) - \boldsymbol{x}_l(q)\right]} \\
&= \overline{\left[\boldsymbol{x}_i(p)\right]\left[\boldsymbol{x}_i(q)\right]} - \overline{\left[\boldsymbol{x}_i(p)\right]\left[\boldsymbol{x}_l(q)\right]} - \overline{\left[\boldsymbol{x}_l(p)\right]\left[\boldsymbol{x}_i(q)\right]} + \overline{\left[\boldsymbol{x}_l(p)\right]\left[\boldsymbol{x}_l(q)\right]} \\
&\quad + \overline{\left[\boldsymbol{x}_j(p)\right]\left[\boldsymbol{x}_j(q)\right]} - \overline{\left[\boldsymbol{x}_j(p)\right]\left[\boldsymbol{x}_i(q)\right]} - \overline{\left[\boldsymbol{x}_i(p)\right]\left[\boldsymbol{x}_j(q)\right]} + \overline{\left[\boldsymbol{x}_i(p)\right]\left[\boldsymbol{x}_i(q)\right]} \\
&\quad - \overline{\left[\boldsymbol{x}_j(p)\right]\left[\boldsymbol{x}_j(q)\right]} + \overline{\left[\boldsymbol{x}_j(p)\right]\left[\boldsymbol{x}_l(q)\right]} + \overline{\left[\boldsymbol{x}_l(p)\right]\left[\boldsymbol{x}_j(q)\right]} - \overline{\left[\boldsymbol{x}_l(p)\right]\left[\boldsymbol{x}_l(q)\right]} \\
&= \overline{\left[\boldsymbol{x}_i(p) - \boldsymbol{x}_j(p)\right]\left[\boldsymbol{x}_i(q) - \boldsymbol{x}_l(q)\right]} + \overline{\left[\boldsymbol{x}_i(p) - \boldsymbol{x}_l(p)\right]\left[\boldsymbol{x}_i(q) - \boldsymbol{x}_j(q)\right]} \\
&\overset{(4)}{=} \mathbf{\Gamma}_{i;j;i;l} + \mathbf{\Gamma}_{i;l;i;j}
\end{aligned}
\tag{23}
$$

The relation between residual covariances and residual cross-covariances in Eq. (23) is exact and holds for all datasets without any further assumption. In case of symmetric residual cross-covariances $\left(\mathbf{\Gamma}_{i;j;i;l} = \mathbf{\Gamma}_{i;l;i;j} \overset{(23)}{=} \frac{1}{2}\left[\mathbf{\Gamma}_{i;l} + \mathbf{\Gamma}_{j;i} - \mathbf{\Gamma}_{j;l}\right]\right)$, the residual cross-covariance matrices are fully determined by the symmetric residual covariances.

In the general asymmetric case, Eq. (23) can be rewritten as:

$$
\begin{aligned}
\mathbf{\Gamma}_{i;l} + \mathbf{\Gamma}_{j;i} - \mathbf{\Gamma}_{j;l} &\overset{(23)}{=} \mathbf{\Gamma}_{i;j;i;l} + \mathbf{\Gamma}_{i;l;i;j} \overset{(18)}{=} \mathbf{\Gamma}_{i;j;i;l} + \left[\mathbf{\Gamma}_{i;j;i;l} - \mathbf{Y}_{i;j;i;l}\right] \\
&\Longleftrightarrow \quad \mathbf{\Gamma}_{i;j;i;l} = \frac{1}{2}\left[\mathbf{\Gamma}_{i;l} + \mathbf{\Gamma}_{j;i} - \mathbf{\Gamma}_{j;l}\right] + \frac{1}{2}\mathbf{Y}_{i;j;i;l}
\end{aligned}
\tag{24}
$$

Equation (24) shows that each individual residual cross-covariance consists of a symmetric contribution including residual covariances between the datasets and an asymmetric contribution being half of the related residual asymmetry matrix. Thus, residual cross-covariances may only provide additional information on asymmetries of error statistics, but not on symmetric statistics (like error covariances).

## 3.3 Exact error statistics

As an extension to previous work, this section provides generalized formulations of error covariances, cross-covariances, and dependencies in matrix form. These formulations are based on the relations between residual- and error statistics in Eq. (20) and Eq. (22). Note that the general formulations presented here do not provide a closed system of equations which can be solved in real applications. They serve as basis for the approximate solutions which are formulated in the subsequent section.

### 3.3.1 Error statistics from residual covariances

Equation (20) shows that each residual covariance matrix can be expressed by the error covariances of the two datasets involved and their error dependency. The goal is to find an inverse formulation of an error covariance matrix as function of residual covariances which does not include other (unknown) error covariances matrices. By combining the formulations of three

residuals $\mathbf{\Gamma}_{i;j}$, $\mathbf{\Gamma}_{j;k}$, and $\mathbf{\Gamma}_{k;i}$ between the same three datasets $i$, $j$, and $k$ and expressing each using Eq. (20), a single error covariance can be eliminated:

$$\mathbf{C}_{\widetilde{i}} \overset{(20)_{ij}}{=} \mathbf{\Gamma}_{i;j} + \mathbf{D}_{\widetilde{i};\widetilde{j}} - \mathbf{C}_{\widetilde{j}} \tag{25}$$

$$\overset{(20)_{jk}}{=} \mathbf{\Gamma}_{i;j} + \mathbf{D}_{\widetilde{i};\widetilde{j}} - \mathbf{\Gamma}_{j;k} - \mathbf{D}_{\widetilde{j};\widetilde{k}} + \mathbf{C}_{\widetilde{k}}$$

$$\overset{(20)_{ki}}{=} \mathbf{\Gamma}_{i;j} + \mathbf{D}_{\widetilde{i};\widetilde{j}} - \mathbf{\Gamma}_{j;k} - \mathbf{D}_{\widetilde{j};\widetilde{k}} + \mathbf{\Gamma}_{k;i} + \mathbf{D}_{\widetilde{k};\widetilde{i}} - \mathbf{C}_{\widetilde{i}} \tag{26}$$

$$\iff \quad \mathbf{C}_{\widetilde{i}} = \frac{1}{2} \Bigg[ \underbrace{\mathbf{\Gamma}_{i;j} + \mathbf{\Gamma}_{k;i} - \mathbf{\Gamma}_{j;k}}_{\text{"independent contribution"}} + \underbrace{\mathbf{D}_{\widetilde{i};\widetilde{j}} + \mathbf{D}_{\widetilde{k};\widetilde{i}} - \mathbf{D}_{\widetilde{j};\widetilde{k}}}_{\text{"dependent contribution"}} \Bigg] \tag{27}$$

where the indication of used equations above the equal signs are extended by indices which denote to which datasets this equation has been applied. For example, " $\overset{(20)_{ki}}{=}$ " indicates that the relation in Eq. (20) was applied to datasets $k$ and $i$ to achieve the right hand side.

Equation (27) provides a general formulation of error covariances as function of residual covariances and error dependencies. It holds for all combinations of datasets without any further assumption (e.g. independence). Thus, each error covariance can be formulated as a sum of an independent contribution of three residual covariances w.r.t. any pair of other datasets and a dependent contribution of the three related error dependencies. While the independent contribution can be calculated from residual statistics between input datasets, the dependent contribution is generally unknown in real applications.

Given $I$ datasets, the total number of different formulations of each error covariance in Eq. (27) is determined by the number of different pairs of the other datasets which is $\sum_{i=1}^{I-2} i = \frac{1}{2}(I-1)(I-2)$ (see also Sjoberg et al., 2021). The scalar equivalent of Eq. (27) where the dependency matrices reduce to twice the error cross-variances has been formulated previously in the 3CH method e.g. in Anthes and Rieckh (2018); Sjoberg et al. (2021). Very recently, the full matrix form was used by Nielsen et al. (2022); Todling et al. (2022). Note that in the literature, the dependent contribution in Eq. (27) is denoted as cross-covariances between the errors.

Equation (26) can be generalized by replacing the closed series of the 3 dataset-pairs $(i;j), (j;k), (k;i)$ with a closed series of $F$ dataset-pairs $(i_1;i_2), (i_2;i_3), \cdots, (i_{F-1};i_F), (i_F;i_1)$, for any $3 \leq F \leq I$ (and $I$ the number of datasets):

$$\mathbf{C}_{\widetilde{i_1}} \overset{(20)}{=} \sum_{f=1}^{F-1} (-1)^{f-1} \Big[ \mathbf{\Gamma}_{i_f;i_{f+1}} + \mathbf{D}_{\widetilde{i_f};\widetilde{i_{f+1}}} \Big] + (-1)^{F-1} \Big[ \mathbf{\Gamma}_{i_F;i_1} + \mathbf{D}_{\widetilde{i_F};\widetilde{i_1}} \Big] + (-1)^F \mathbf{C}_{\widetilde{i_1}} \tag{28}$$

Because of changing signs, Eq. (28) can only be solved for the error covariance $\mathbf{C}_{\widetilde{i_1}}$ if $F$ is odd. If $F$ is even, $\mathbf{C}_{\widetilde{i_1}}$ cancels out and cannot be eliminated, and Eq. (28) could be solved for one error dependency instead. If $F$ is odd, the generalized formulation for $\mathbf{C}_{\widetilde{i_1}}$ becomes:

$$\mathbf{C}_{\widetilde{i_1}} \overset{(28)}{=} \frac{1}{2} \Bigg[ \underbrace{\left( \sum_{f=1}^{F-1} (-1)^{f-1} \mathbf{\Gamma}_{i_f;i_{f+1}} \right) + \mathbf{\Gamma}_{i_F;i_1}}_{\text{"independent contribution"}} + \underbrace{\left( \sum_{f=1}^{F-1} (-1)^{f-1} \mathbf{D}_{\widetilde{i_f};\widetilde{i_{f+1}}} \right) + \mathbf{D}_{\widetilde{i_F};\widetilde{i_1}}}_{\text{"dependent contribution"}} \Bigg] \quad , \forall F \text{ odd} \wedge 3 \leq F \leq I \tag{29}$$

where Eq. (27) results for $F = 3$ with indices $i_1 = i$, $i_2 = j$ and $i_3 = k$. Note that in any case, the number of assumed and estimated error statistics remains consistent with the general framework in Sect. 2.

A formulation of each individual error dependency matrix as function of the error covariances of the two datasets and their residual covariance results directly from Eq. (20):

$$\mathbf{D}_{\widetilde{i};\widetilde{j}} \overset{(20)}{=} \mathbf{C}_{\widetilde{i}} + \mathbf{C}_{\widetilde{j}} - \mathbf{\Gamma}_{i;j} \tag{30}$$

Being a symmetric matrix, residual covariances cannot provide information on error asymmetries and thus on asymmetric components of error cross-covariances. Only the symmetric component of error cross-covariances could be estimated from half the error dependency which is equivalent to a zero error asymmetry matrix:

$$\mathbf{D}_{\widetilde{i};\widetilde{j}} + \mathbf{Y}_{\widetilde{i};\widetilde{j}} \overset{(13),(17)}{=} \left[ \mathbf{X}_{\widetilde{i};\widetilde{j}} + \cancel{\mathbf{X}_{\widetilde{j};\widetilde{i}}} \right] + \left[ \mathbf{X}_{\widetilde{i};\widetilde{j}} - \cancel{\mathbf{X}_{\widetilde{j};\widetilde{i}}} \right] \qquad \Longleftrightarrow \qquad \mathbf{X}_{\widetilde{i};\widetilde{j}} = \frac{1}{2} \left[ \mathbf{D}_{\widetilde{i};\widetilde{j}} + \mathbf{Y}_{\widetilde{i};\widetilde{j}} \right] \tag{31}$$

### 3.3.2 Error statistics from residual cross-covariances

The general forward formulation of residual cross-covariances in Eq. (22) consists of error cross-covariances of the four datasets involved. Setting for example $k = i$, provides an inverse formulation of error covariances of $i$:

$$\mathbf{\Gamma}_{i;j;i;l} \overset{(22)}{=} \mathbf{C}_{\widetilde{i}} - \mathbf{X}_{\widetilde{i};l} - \mathbf{X}_{\widetilde{j};\widetilde{i}} + \mathbf{X}_{\widetilde{j};l} \qquad \Longleftrightarrow \qquad \mathbf{C}_{\widetilde{i}} = \mathbf{\Gamma}_{i;j;i;l} + \mathbf{X}_{\widetilde{i};l} + \mathbf{X}_{\widetilde{j};\widetilde{i}} - \mathbf{X}_{\widetilde{j};l} \tag{32}$$

The scalar formulation of Eq. (32) was previously given in Zwieback et al. (2012).

Similarly to Eq. (27) from residual covariances, the number of formulations of each error covariance from different pairs of other datasets in Eq. (32) is $\sum_{i=1}^{I-2} i = \frac{1}{2}(I-1)(I-2)$. In addition, there are four possibilities to write each error covariances from the same pairs of other datasets using the relations of residual cross-covariances in Eq. (8). Each of the four possibilities results from setting both indices of one pair of datasets in definition of residual cross-covariances in Eq. (22) to the same value. Two of the error cross-covariances in Eq. (32) can be rewritten by applying Eq. (32) to the error covariance of dataset $j$:

$$\mathbf{C}_{\widetilde{j}} \overset{(32)j}{=} \mathbf{\Gamma}_{j;i;j;l} + \mathbf{X}_{\widetilde{j};l} + \mathbf{X}_{\widetilde{i};\widetilde{j}} - \mathbf{X}_{\widetilde{i};l} \qquad \Longleftrightarrow \qquad \mathbf{X}_{\widetilde{i};l} - \mathbf{X}_{\widetilde{j};l} = \mathbf{\Gamma}_{j;i;j;l} + \mathbf{X}_{\widetilde{i};\widetilde{j}} - \mathbf{C}_{\widetilde{j}} \tag{33}$$

With this, Eq. (32) becomes:

$$\mathbf{C}_{\widetilde{i}} \overset{(33)}{=} \mathbf{\Gamma}_{i;j;i;l} + \mathbf{\Gamma}_{j;i;j;l} - \mathbf{C}_{\widetilde{j}} + \mathbf{X}_{\widetilde{i};\widetilde{j}} + \mathbf{X}_{\widetilde{j};\widetilde{i}} \overset{(13)}{=} \mathbf{\Gamma}_{i;j;i;l} + \mathbf{\Gamma}_{j;i;j;l} - \mathbf{C}_{\widetilde{j}} + \mathbf{D}_{\widetilde{i};\widetilde{j}} \tag{34}$$

Because the residual cross-covariances can be rewritten as:

$$\mathbf{\Gamma}_{i;j;i;l} + \mathbf{\Gamma}_{j;i;j;l} \overset{(32),(33)}{=} \mathbf{C}_{\widetilde{i}} - \cancel{\mathbf{X}_{\widetilde{i};l}} - \mathbf{X}_{\widetilde{j};\widetilde{i}} + \cancel{\mathbf{X}_{\widetilde{j};l}} + \mathbf{C}_{\widetilde{j}} - \cancel{\mathbf{X}_{\widetilde{j};l}} - \mathbf{X}_{\widetilde{i};\widetilde{j}} + \cancel{\mathbf{X}_{\widetilde{i};l}} \overset{(13)}{=} \mathbf{C}_{\widetilde{i}} + \mathbf{C}_{\widetilde{j}} - \mathbf{D}_{\widetilde{i};\widetilde{j}} \overset{(20)}{=} \mathbf{\Gamma}_{i;j} \tag{35}$$

the formulation of error covariances based on residual cross-covariances in Eq. (34) is, as it must be, symmetric and equivalent to the formulation based on residual covariances from Eq. (25).

The forward formulation of residual cross-covariances does not allow for an elimination of one single error cross-covariance even when multiple equations are combined. One formulation of an error cross-covariance matrix as function of residual cross-covariances results directly from the forward relation:

$$\mathbf{X}_{\widetilde{j};\widetilde{l}} \overset{(32)}{=} \mathbf{\Gamma}_{i;j;i;l} - \mathbf{C}_{\widetilde{i}} + \mathbf{X}_{\widetilde{i};\widetilde{l}} + \mathbf{X}_{\widetilde{j};\widetilde{i}} \tag{36}$$

Note that the third dataset $i$ on the right hand side of Eq. (36) can be any other dataset ($i \neq j, i \neq l$). Thus for any $I > 2$, there are $I - 2$ formulations of each error cross-covariance $\mathbf{X}_{\widetilde{j};\widetilde{l}}$, which are all equivalent in the exact formulation.

Any of the formulations of error cross-covariances can also be used for a formulation of the error dependency matrix $\mathbf{D}_{\widetilde{j};\widetilde{l}}\big|_{\mathrm{cross}}$ which is equivalent to the formulation based on residual covariances $\mathbf{D}_{\widetilde{j};\widetilde{l}}\big|_{\mathrm{covar}}$:

$$
\begin{aligned}
\mathbf{D}_{\widetilde{j};\widetilde{l}}\big|_{\mathrm{cross}} &\overset{(13)}{=} \mathbf{X}_{\widetilde{j};\widetilde{l}} + \mathbf{X}_{\widetilde{l};\widetilde{j}} \overset{(36)}{=} \mathbf{\Gamma}_{j;i;l;i} - \mathbf{C}_{\widetilde{i}} + \mathbf{X}_{\widetilde{j};\widetilde{i}} + \mathbf{X}_{\widetilde{i};\widetilde{l}} + \mathbf{\Gamma}_{l;i;j;i} - \mathbf{C}_{\widetilde{i}} + \mathbf{X}_{\widetilde{i};\widetilde{j}} + \mathbf{X}_{\widetilde{l};\widetilde{i}} \\
&\overset{(13)}{=} \mathbf{\Gamma}_{j;i;l;i} + \mathbf{\Gamma}_{l;i;j;i} - 2\,\mathbf{C}_{\widetilde{i}} + \mathbf{D}_{\widetilde{i};\widetilde{j}} + \mathbf{D}_{\widetilde{i};\widetilde{l}} \overset{(23)}{=} \mathbf{\Gamma}_{i;j} + \mathbf{D}_{\widetilde{i};\widetilde{j}} + \mathbf{\Gamma}_{i;l} + \mathbf{D}_{\widetilde{i};\widetilde{l}} - \mathbf{\Gamma}_{j;l} - 2\,\mathbf{C}_{\widetilde{i}} \\
&\overset{(20)}{=} \cancel{\mathbf{C}_{\widetilde{i}}} + \mathbf{C}_{\widetilde{j}} + \cancel{\mathbf{C}_{\widetilde{i}}} + \mathbf{C}_{\widetilde{l}} - \mathbf{\Gamma}_{j;l} - \cancel{2\,\mathbf{C}_{\widetilde{i}}} = \mathbf{C}_{\widetilde{j}} + \mathbf{C}_{\widetilde{l}} - \mathbf{\Gamma}_{j;l} \overset{(30)}{=} \mathbf{D}_{\widetilde{j};\widetilde{l}}\big|_{\mathrm{covar}}
\end{aligned} \tag{37}
$$

The equivalence demonstrates that, as they must be, the exact formulations of error statistics from residual covariances and cross-covariances are consistent with each other. This consistency applies to the exact formulations of all symmetric error statistics (error covariances and dependencies) and results from the consistent definitions of residual covariances and cross-covariances in Eq. (4) and (6).

## 4 Mathematical theory: Approximate formulation

Based on the exact formulations in Sect. 3 which remain underdetermined in real applications, this section provides approximate formulations for three and more datasets which provide a closed system of equations. Section 4.1 describes the long-known closure of the system for three datasets, but for full covariance matrices. An extension for more than three datasets based on a minimal number of assumptions is introduced in Sect. 4.2. It includes the estimation, either direct or sequential, of additional error covariances, as well as of some error cross-statistics.

In addition to the optimal extension to more than three datasets, this second part of the mathematical theory includes the following new elements: (i) the analysis of differences between error estimates from residual covariances and cross-covariances (Sect. 4.1.2), (ii) the determination of uncertainties resulting from possible errors in the assumed error statistics (Sect. 4.1.3 and 4.2.4), and (iii) the comparison of the approximations from direct and sequential estimates (Sect. 4.2.5).

### 4.1 Approximation for three datasets

As demonstrated in Sect. 2, at least three collocated datasets are required to estimate all error covariances ($A_I \geq 0$). For three datasets ($I = 3$), three residual covariances ($N_3 = 3$) can be calculated between each pair of datasets. At the same time, there are six unknown error statistics ($U_3 = 6$): three error covariances and three error cross-statistics (cross-covariances or

dependencies). Thus, the problem is under-determined and three error statistics ($U_3 - N_3 = 3$) have to be assumed in order to close the system. The most common approach, which is also used in 3CH and TC methods, is to assume zero error cross-statistics between all pairs of datasets: $\mathbf{X}_{\widetilde{i};\widetilde{j}} = 0 \ \Rightarrow \ \mathbf{D}_{\widetilde{i};\widetilde{j}} = 0 \ , \forall \ i,j \in [1,3], \ j \neq i$. The approximation of the three error covariances can also be formulated in a Hilbert space which allows for an illustrative geometric interpretation as in Pan et al. (2015) (their Fig. 1). Because the assumption of zero error cross-covariance implies zero error correlation which is often used as proxy for independence, it is denoted as "assumption of independence" or "independence assumption" hereafter.

The independence assumption resembles the innovation covariance consistency of data assimilation, where the residual covariance between background and observation datasets - denoted as innovation covariance - is assumed to be equal to the sum of their error covariances in the formulation of the analysis (e.g., Daley, 1992b; Ménard, 2016):

$$\mathbf{\Gamma}_{i;j} \underset{\{in\}}{\overset{(20)}{\approx}} \mathbf{C}_{\widetilde{i}} + \mathbf{C}_{\widetilde{j}} \tag{38}$$

where " $\underset{\{in\}}{\approx}$ " indicates the assumption of independence between the two datasets, i.e. $\mathbf{X}_{\widetilde{i};\widetilde{j}} = 0$.

Because all error cross-statistics need to be assumed in this setup, approximations of these cross-covariances and dependencies only reproduces the initially assumed statistics and do not provide any new information.

### 4.1.1 Error covariance estimates

Assuming independent error statistics among all three datasets, or similarly that error dependencies are negligible compared to residual covariances $\mathbf{D}_{\widetilde{i};\widetilde{j}} \ll \mathbf{\Gamma}_{i;j} \ , \forall \ j \neq i$ gives an estimate of each error covariance matrix as function of three residual covariances:

$$\mathbf{C}_{\widetilde{i}} \underset{\{in3\}}{\overset{(27)}{\approx}} \frac{1}{2} \left[ \mathbf{\Gamma}_{i;j} + \mathbf{\Gamma}_{k;i} - \mathbf{\Gamma}_{j;k} \right] \tag{39}$$

where " $\underset{\{in3\}}{\approx}$ " indicates the assumption of independence among all three datasets involved.

In the scalar case, Eq. (39) reduces to the equivalent formulation for error variances known from the TC and 3CH method (e.g., Pan et al., 2015; Sjoberg et al., 2021). Thus, the long-known 3CH estimation of error variances from residual variances among three datasets holds similarly for complete error covariance matrices from residual covariances under the independence assumption. In fact, the approximation in Eq. (39) requires only the assumption that the dependent contribution of Eq. (27) vanishes. However combining this condition for the error covariance estimates of all three datasets results in the need for each error dependency to be zero.

Under the assumption of independence among all three datasets $\mathbf{X}_{\widetilde{i};\widetilde{j}} = 0 \ , \forall \ i,j$, their error covariance matrices can also be directly estimated from residual cross-covariances:

$$\mathbf{C}_{\widetilde{i}} \underset{\{in3\}}{\overset{(32)}{\approx}} \mathbf{\Gamma}_{i;j;i;l} \tag{40}$$

And likewise:

$$\mathbf{C}_{\widetilde{i}} \underset{\{in3\}}{\overset{(32)}{\approx}} \mathbf{\Gamma}_{i;l;i;j} \tag{41}$$

As described in Sect. 3.3.2 on exact cross-covariance statistics, every error covariance from residual cross-covariances has four equivalent formulations which provide the same result in the exact case, but might differ in the approximate formulation. Equation (40) and (41) provide two different approximations of each error covariance matrix from residual cross-covariances based on each pair of other datasets. In the simplified case of scalar statistics, the two different formulations in Eq. (40) and (41) reduce to the same residual cross-variance which was previously formulated by e.g. Crow and van den Berg (2010); Zwieback et al. (2012); Pan et al. (2015).

### 4.1.2 Differences

Equations (39) to (41) provide three different estimates of an error covariance matrix. Using the relation between residual covariances and cross-covariances from Sect. 3.2.3 and the symmetric properties of residual statistics allow for a comparison of the three estimates:

$$\mathbf{C}_{\widetilde{i}}\Big|_{(40)} \overset{(40)}{\underset{\{in3\}}{\approx}} \mathbf{\Gamma}_{i;j;i;l} \overset{(24),(39)}{=} \mathbf{C}_{\widetilde{i}}\Big|_{(39)} + \frac{1}{2}\mathbf{Y}_{i;j;i;l} \tag{42}$$

$$\mathbf{C}_{\widetilde{i}}\Big|_{(41)} \overset{(41)}{\underset{\{in3\}}{\approx}} \mathbf{\Gamma}_{i;l;i;j} \overset{(24),(39)}{=} \mathbf{C}_{\widetilde{i}}\Big|_{(39)} - \frac{1}{2}\mathbf{Y}_{i;j;i;l} \tag{43}$$

The three independent estimates of a error covariance matrix from the same pair of other datasets differ only by their residual asymmetry. Thus, differences between the estimates from Eq. (39) to (41) provide no additional information about symmetric error statistics.

While the estimation from residual covariances remains symmetric by definition, the estimates of error covariances from residual cross-covariances may become asymmetric. This asymmetry can be eliminated using the residual asymmetry matrix which is also equivalent to averaging both formulations of error covariances from residual cross-covariances:

$$\mathbf{C}_{\widetilde{i}} \overset{(39)}{\underset{\{in3\}}{\approx}} \frac{1}{2}\Big[\mathbf{\Gamma}_{i;j} + \mathbf{\Gamma}_{l;i} - \mathbf{\Gamma}_{j;l}\Big] \overset{(42)}{=} \mathbf{\Gamma}_{i;j;i;l} - \frac{1}{2}\mathbf{Y}_{i;j;i;l} \overset{(43)}{=} \mathbf{\Gamma}_{i;l;i;j} + \frac{1}{2}\mathbf{Y}_{i;j;i;l} \tag{44}$$

All three estimates become equivalent if the residual cross-covariances and thus, error cross-covariances are symmetric ( $\rightarrow \mathbf{X}_{\widetilde{i};\widetilde{j}} = \frac{1}{2}\mathbf{D}_{\widetilde{i};\widetilde{j}} = \mathbf{X}_{\widetilde{j};\widetilde{i}}$ , $\forall\, i,j$ ). This is also the case for scalar statistics, where the equivalence between scalar error variance estimates from residual variances and cross-variances was previously shown by Pan et al. (2015). However, none of the estimates ensures positive definiteness of the estimated error covariances.

### 4.1.3 Uncertainties of approximation

The independence assumption introduces the following absolute uncertainties $\Delta\mathbf{C}_{\widetilde{i}}$ of the three different estimates for each dataset $i$:

$$\Delta\mathbf{C}_{\widetilde{i}}\Big|_{(39)} := \mathbf{C}_{\widetilde{i}}\Big|_{\text{true}} - \mathbf{C}_{\widetilde{i}}\Big|_{(39)} \overset{(27),(39)}{=} \frac{1}{2}\Big[\Delta\mathbf{D}_{\widetilde{i};\widetilde{j}} + \Delta\mathbf{D}_{\widetilde{i};\widetilde{k}} - \Delta\mathbf{D}_{\widetilde{j};\widetilde{k}}\Big] \tag{45}$$

$$\Delta\mathbf{C}_{\widetilde{i}}\Big|_{(40)} := \mathbf{C}_{\widetilde{i}}\Big|_{\text{true}} - \mathbf{C}_{\widetilde{i}}\Big|_{(40)} \overset{(32),(40)}{=} \Delta\mathbf{X}_{\widetilde{j};\widetilde{i}} + \Delta\mathbf{X}_{\widetilde{i};\widetilde{k}} - \Delta\mathbf{X}_{\widetilde{j};\widetilde{k}} \tag{46}$$

$$\Delta\mathbf{C}_{\widetilde{i}}\Big|_{(41)} := \mathbf{C}_{\widetilde{i}}\Big|_{\text{true}} - \mathbf{C}_{\widetilde{i}}\Big|_{(41)} \overset{(32),(41)}{=} \Delta\mathbf{X}_{\widetilde{i};\widetilde{j}} + \Delta\mathbf{X}_{\widetilde{k};\widetilde{i}} - \Delta\mathbf{X}_{\widetilde{k};\widetilde{j}} \tag{47}$$

where $\Delta \mathbf{D}_{\widetilde{i};\widetilde{j}}$ and $\Delta \mathbf{X}_{\widetilde{i};\widetilde{j}}$ are the uncertainties of the estimated error dependencies and cross-covariances, respectively.

The absolute uncertainty of the estimates depends similarly on the (neglected) error cross-covariances or dependencies among the three datasets. While the error dependencies to the two other datasets contribute positively, the dependency between the two others is subtracted. If these dependencies cancel out ($\Delta \mathbf{D}_{\widetilde{i};\widetilde{j}} + \Delta \mathbf{D}_{\widetilde{i};\widetilde{k}} = \Delta \mathbf{D}_{\widetilde{j};\widetilde{k}}$), the estimate of one dataset might be exact even if all three dependencies are non-zero. However, two exact estimates can only be achieved if one (e.g. $\Delta \mathbf{D}_{\widetilde{i};\widetilde{j}} = 0 \wedge \Delta \mathbf{D}_{\widetilde{i};\widetilde{k}} = \Delta \mathbf{D}_{\widetilde{j};\widetilde{k}}$) or all three dependencies are zero. A special case was observed by Todling et al. (2022) who showed that the estimations of background, observation and analysis errors in a variational data assimilation system become exact if the analysis is optimal. In this particular case, no assumptions on dependencies are required because the optimality of the analysis induces vanishing dependencies.

Estimated error covariances might even contain negative values if error dependencies are large compared to the true error covariance of a dataset. If the true error covariances differ significantly among highly correlated datasets, the neglected error dependency between two datasets might become much larger than the smaller error covariance, e.g. $\Delta \mathbf{D}_{\widetilde{k};\widetilde{i}} - \Delta \mathbf{D}_{\widetilde{j};\widetilde{k}} \approx 0$, $\frac{1}{2}\Delta \mathbf{D}_{\widetilde{i};\widetilde{j}} > \mathbf{C}_{\widetilde{i}}\big|_{\text{true}}$. This phenomena was also described and demonstrated by Sjoberg et al. (2021) for scalar problems, but the generalization to covariances matrices is expected to increase the occurrence of negative values in off-diagonal elements. Because spatial correlations and thus true covariances may become small compared to uncertainties in the assumptions or sampling noise, estimated error covariances at these locations might become negative. However, the occurrence of negative elements does not affect the positive definiteness of a covariance matrix, which is determined by the sign of its eigenvalues.

## 4.2 Approximation for more than three datasets

While independence among all datasets is required to estimate the error covariances of three datasets ($I = 3$), the use of more than three datasets ($I > 3$) enables the additional estimation of some error dependencies or cross-covariances (compare Sect. 2). Although this potential of cross-statistic estimation was previously indicated by Gruber et al. (2016); Vogelzang and Stoffelen (2021) for scalar problems, a generalized formulation exploiting its full potential by minimizing the number of assumptions is yet missing.

As described in Sect. 2, for $I > 3$ datasets, $A_I > 0$ gives the number of error cross-statistics which can potentially be estimated in addition to all error covariances. Consequentially, the independent assumption between all pairs of datasets can be relaxed to a "partial independence assumption" where one independent dataset-pair is required for each dataset $I$. The estimation of error covariances can be generalized in two ways: Firstly, the direct formulation for three datasets in Sect. 4.1.1 is generalized to a direct estimation of more than three datasets in Sect. 4.2.1. Secondly, Sect. 4.2.2 introduces the sequential estimation of error covariances of any additional dataset. This estimation procedure of additional error covariances is denoted as "sequential estimation" because is requires the error covariance estimate of a prior dataset, in contrast to the "direct estimation" from an independent triplet of datasets ("triangular estimation" in Sect. 4.1) or generally from a closed series of pairwise-independent datasets ("polygonal estimation" in Sect. 4.2.1).

### 4.2.1 Direct error covariance estimates

For more than three datasets $I > 3$, the estimation from three residual covariances in Eq. (39) can be generalized to estimations of error covariances from a closed series of $F$ residual covariances (compare Sect. 3.3.1). For any odd $F$ with $3 \leq F \leq I$, each error covariance can be estimated under the assumption of vanishing error dependencies along the closed series of datasets $\mathbf{D}_{\widetilde{i_f};\widetilde{i_{f+1}}} \ \forall \ f \in [1, F-1]$ and $\mathbf{D}_{\widetilde{i_F};\widetilde{1}}$:

$$\mathbf{C}_{\widetilde{i_1}} \underset{\{inF\}}{\overset{(29)}{\approx}} \frac{1}{2} \left[ \left( \sum_{f=1}^{F-1} (-1)^{f-1} \mathbf{\Gamma}_{i_f;i_{f+1}} \right) + \mathbf{\Gamma}_{i_F;i_1} \right] \qquad , \ \forall \ F \ \text{odd} \ \wedge \ 3 \leq F \leq I \tag{48}$$

where " $\underset{\{inF\}}{\approx}$ " indicates the assumption of neglectable error dependencies along the series of datasets. As shown in Sect. 2, the problem cannot be closed for less than 3 datasets, even under the independent assumption. For $F = 3$ datasets, Eq. (39) is a special case of Eq. (48) with indices $i_1 = i$, $i_2 = j$ and $i_3 = k$.

### 4.2.2 Sequential error covariance estimates

Similar to the estimation for three datasets $I = 3$ in Sect. 4.1.1, the error covariances of the first three datasets can be directly estimated from residual covariances or cross-covariances using Eq. (39), (40) or (41). This triplet of the first three datasets which are assumed to be pairwise independent is denoted as "basic triangle". Similarly, a "basic polygon" can be defined from a closed series of $F$ pairwise-independent datasets, where each two successive datasets in the series as well as the last and first element are independent from each other (compare Sect. 4.2.1). Then, the error covariance of each dataset in the series can be directly estimated from Eq. (48).

Based on this, the remaining error covariances can be calculated sequentially. For each additional dataset $i$ with $F < i < I$, its cross-statistics to one prior dataset $\mathrm{ref}(i) < i$ is needed to be assumed in order to close the problem. This prior dataset $\mathrm{ref}(i)$ is denoted as "reference dataset" of dataset $i$. With this, the remaining error covariances can be estimated from residual covariances under the partial independence assumption $\mathbf{X}_{\widetilde{i};\widetilde{\mathrm{ref}(i)}} = 0$:

$$\mathbf{C}_{\widetilde{i}} \underset{\{inI\}}{\overset{(25)}{\approx}} \mathbf{\Gamma}_{i;\mathrm{ref}(i)} - \mathbf{C}_{\widetilde{\mathrm{ref}(i)}} \tag{49}$$

where " $\underset{\{inI\}}{\approx}$ " indicates the assumption of independence to the reference dataset, i.e. $\mathbf{X}_{\widetilde{i};\widetilde{\mathrm{ref}(i)}} = 0$.

Similarly, each additional error covariance can be estimated from two residual cross-covariances w.r.t its reference dataset $\mathrm{ref}(i)$ and any other dataset $j$:

$$\mathbf{C}_{\widetilde{i}} \underset{\{inI\}}{\overset{(34)}{\approx}} \mathbf{\Gamma}_{i;\mathrm{ref}(i);i;j} + \mathbf{\Gamma}_{\mathrm{ref}(i);i;\mathrm{ref}(i);j} - \mathbf{C}_{\widetilde{\mathrm{ref}(i)}} \tag{50}$$

From the equivalence of residual statistics in Eq. (35) it follows that the two formulations of error covariances in Eq. (49) and Eq. (50), respectively, are equivalent and produce exactly the same estimates even if the underlying assumptions are not perfectly fulfilled.

### 4.2.3 Error cross-covariance and dependency estimates

Once the error covariances are estimated, the remaining residual covariances can be used to calculate the error dependencies to all other prior datasets $j \neq \mathrm{ref}(i), j < i$:

$$\mathbf{D}_{\widetilde{i};\widetilde{j}} \overset{(30)}{=} \mathbf{C}_{\widetilde{i}} + \mathbf{C}_{\widetilde{j}} - \mathbf{\Gamma}_{i;j} \tag{51}$$

In contrast to residual covariances, the asymmetric formulation of residual cross-covariances allows for an estimation of remaining error cross-covariances including their asymmetric components. The error cross-covariance to each other prior dataset $j \neq \mathrm{ref}(i), j < i$ can be estimated sequentially using again the reference dataset $\mathrm{ref}(i)$:

$$\mathbf{X}_{\widetilde{i};\widetilde{j}} \overset{(36)}{\underset{\{inI\}}{\approx}} \mathbf{\Gamma}_{\mathrm{ref}(i);i;\mathrm{ref}(i);j} - \mathbf{C}_{\widetilde{\mathrm{ref}(i)}} + \mathbf{X}_{\widetilde{\mathrm{ref}(i)};\widetilde{j}} \tag{52}$$

Based on this, the symmetric error dependencies can be estimated from its definition in Eq. (13). The equivalence between the formulations of error dependencies from residual covariances and cross-covariances was shown in Eq. (37).

Note that the error cross-covariances $\mathbf{X}_{\widetilde{j};\widetilde{i}}$ and dependencies $\mathbf{D}_{\widetilde{j};\widetilde{i}}$ of each subsequent dataset $j > i$ to dataset $j$ result directly from their symmetric properties in Eq. (10) and Eq. (14), respectively.

### 4.2.4 Uncertainties in approximation

As generalization of Eq. (45), the absolute uncertainty $\Delta \mathbf{C}_{\widetilde{i_1}}$ of a polgonal error covariance estimate introduced by the assumption of pairwise-independence along the closed series of $F$ datasets, with $F$ odd and $3 \leq F \leq I$, is given by:

$$\Delta \mathbf{C}_{\widetilde{i_1}}\Big|_{(48)} := \mathbf{C}_{\widetilde{i_1}}\Big|_{\mathrm{true}} - \mathbf{C}_{\widetilde{i_1}}\Big|_{(48)} \overset{(29),(48)}{=} \frac{1}{2}\left[\left(\sum_{f=1}^{F-1}(-1)^{f-1}\Delta \mathbf{D}_{\widetilde{i_f};\widetilde{i_{f+1}}}\right) + \Delta \mathbf{D}_{\widetilde{i_F};\widetilde{i_1}}\right] \quad , \forall\, F \text{ odd} \wedge 3 \leq F \leq I \tag{53}$$

Due to the changing sign of error dependencies along the series of datasets, the absolute uncertainty of the error covariance estimates does not necessary increase with the size of the polygon $F$.

The absolute uncertainty $\Delta \mathbf{C}_{\widetilde{i}}$ of a sequential error covariance estimate of any additional dataset $i$ with $F < i < I$ is formulated recursively w.r.t. its reference dataset $\mathrm{ref}(i)$:

$$\Delta \mathbf{C}_{\widetilde{i}}\Big|_{(49)} := \mathbf{C}_{\widetilde{i}}\Big|_{\mathrm{true}} - \mathbf{C}_{\widetilde{i}}\Big|_{(49)} \overset{(25),(49)}{=} \Delta \mathbf{D}_{\widetilde{i;\mathrm{ref}(i)}} - \Delta \mathbf{C}_{\widetilde{\mathrm{ref}(i)}} \tag{54}$$

$$\Delta \mathbf{C}_{\widetilde{i}}\Big|_{(50)} := \mathbf{C}_{\widetilde{i}}\Big|_{\mathrm{true}} - \mathbf{C}_{\widetilde{i}}\Big|_{(50)} \overset{(34),(50)}{=} \Delta \mathbf{D}_{\widetilde{i;\mathrm{ref}(i)}} - \Delta \mathbf{C}_{\widetilde{\mathrm{ref}(i)}} \tag{55}$$

The two sequential estimates of error covariances from residual covariances in Eq. (54) and from cross-covariances in Eq. (55) are equivalent, and the uncertainty of the latter is independent of the selection of the third dataset $j$ in the residual cross-covariances (compare Eq. (50)). Thus, the absolute uncertainties of error estimations from residual covariances and cross-covariances differ only in the uncertainties w.r.t. the basic polygon given in Eq. (45) to (48).

With this, a series of reference datasets $\{m_g\} = m_1, \ldots, m_G$, with $m_G$ being the reference of $i$, and $m_{G-1}$ the reference of $m_G$ and so on, with $m_{g-1} < m_g < i$ ,$\forall g$ and $m_1 = j \leq 3$ are defined from the target dataset to the basic triangle as example

for a basic polygon. Then, the absolute uncertainty $\Delta \mathbf{C}_{\widetilde{i}}$ of each error covariance estimate is:

$$\Delta \mathbf{C}_{\widetilde{i}} \stackrel{(54)}{=} \Delta \mathbf{D}_{\widetilde{i};\widetilde{m_G}} - \Delta \mathbf{C}_{\widetilde{m_G}} = \Delta \mathbf{D}_{\widetilde{i};\widetilde{m_G}} - \Delta \mathbf{D}_{\widetilde{m_G};\widetilde{m_{G-1}}} + \Delta \mathbf{C}_{\widetilde{m_{G-1}}} = \dots$$

$$\stackrel{(45)}{=} \Delta \mathbf{D}_{\widetilde{i};\widetilde{m_G}} + \sum_{g=G-1}^{1} \left[ (-1)^{G-g} \cdot \Delta \mathbf{D}_{\widetilde{m_{g+1}};\widetilde{m_g}} \right] + (-1)^G \cdot \frac{1}{2} \left[ \Delta \mathbf{D}_{\widetilde{j};\widetilde{k}} + \Delta \mathbf{D}_{\widetilde{j};\widetilde{l}} - \Delta \mathbf{D}_{\widetilde{k};\widetilde{l}} \right] \tag{56}$$

where $k, l \leq 3$ are the other two datasets in the basic triangle.

According to Eq. (56), uncertainties in the sequential estimations of additional error covariances result from the partial in-
dependence assumption of the additional datasets in the series of reference datasets and the independence assumption in the basic triangle. Due to the changing sign between the intermediate dependencies as well as within the basic triangle (or basic polygon), the individual uncertainties may cancel out. Thus, absolute uncertainties do not necessarily increase with more intermediate reference datasets.

Although Eq. (51) is exact, the error dependency estimate of each additional pair of datasets $(i;j)$ is influenced by uncertainties in the estimations of the related error covariances:

$$\Delta \mathbf{D}_{\widetilde{i};\widetilde{j}} := \mathbf{D}_{\widetilde{i};\widetilde{j}}\Big|_{\text{true}} - \mathbf{D}_{\widetilde{i};\widetilde{j}}\Big|_{(51)} \stackrel{(30),(51)}{=} \Delta \mathbf{C}_{\widetilde{i}} + \Delta \mathbf{C}_{\widetilde{j}} \tag{57}$$

where the uncertainties of the two error covariances are given in Eq. (53) to (56).

And the absolute uncertainties of estimates of additional error cross-covariances based on residual cross-covariances can be
determined recursively using Eq. (56):

$$\Delta \mathbf{X}_{\widetilde{i};\widetilde{j}} := \mathbf{X}_{\widetilde{i};\widetilde{j}}\Big|_{\text{true}} - \mathbf{X}_{\widetilde{i};\widetilde{j}}\Big|_{(52)} \stackrel{(36),(52)}{=} \Delta \mathbf{X}_{\widetilde{\text{ref}(i)};\widetilde{j}} + \Delta \mathbf{X}_{\widetilde{i};\widetilde{\text{ref}(i)}} - \Delta \mathbf{C}_{\widetilde{\text{ref}(i)}} \tag{58}$$

In contrast to error covariances, the uncertainties of error cross-covariances sum up in the two series of reference datasets. However, this sum is subtracted by the two sums of uncertainties in error covariances of these datasets, whose elements may cancel partially (not shown).

### 4.2.5 Comparison to approximation from three datasets

It can be shown that the sequential formulation of an error covariance from its reference dataset is consistent with the triangular formulation from three independent datasets in Sect. 4.1 in the basic triangle. Given the triangular estimate of one error covariance $\mathbf{C}_{\widetilde{i}}\big|_{\triangleleft}$ from Eq. (39), the error covariances $\mathbf{C}_{\widetilde{j}}\big|_{\triangleleft}$ of the other two datasets in the basic triangle are equal to their sequential formulation $\mathbf{C}_{\widetilde{j}}\big|_{\vdash}$ from Eq. (49) with reference dataset $\text{ref}(j) = i$:

$$\mathbf{C}_{\widetilde{j}}\big|_{\vdash} \underset{\{inI\}}{\overset{(49)}{\approx}} \mathbf{\Gamma}_{i;j} - \mathbf{C}_{\widetilde{i}}\big|_{\triangleleft} \underset{\{in3\}}{\overset{(39)_i}{\approx}} \mathbf{\Gamma}_{j;i} - \frac{1}{2}\left[\mathbf{\Gamma}_{i;j} + \mathbf{\Gamma}_{k;i} - \mathbf{\Gamma}_{j;k}\right] = \frac{1}{2}\left[\mathbf{\Gamma}_{i;j} + \mathbf{\Gamma}_{j;k} - \mathbf{\Gamma}_{i;k}\right] \underset{\{in3\}}{\overset{(39)_j}{\approx}} \mathbf{C}_{\widetilde{j}}\big|_{\triangleleft} \tag{59}$$

This can also be generalized for the estimation of any error covariance $\mathbf{C}_{\widetilde{i_2}|\vdash}$ given its reference $\mathbf{C}_{\widetilde{i_1}|\diamond}$ estimated with the polygonal formulation for a closed series of $F$ pairwise-independent datasets for any odd $F$ with $3 \leq F \leq I$:

$$
\begin{aligned}
\mathbf{C}_{\widetilde{i_2}|\vdash} &\overset{(49)}{\underset{\{inI\}}{\approx}} \boldsymbol{\Gamma}_{i_1;i_2} - \mathbf{C}_{\widetilde{i_1}|\diamond} \\
&\overset{(48)_{i_1}}{\underset{\{inF\}}{\approx}} \boldsymbol{\Gamma}_{i_1;i_2} - \frac{1}{2}\left[\left(\sum_{f=1}^{F-1}(-1)^{f-1}\boldsymbol{\Gamma}_{i_f;i_{f+1}}\right) + \boldsymbol{\Gamma}_{i_F;i_1}\right] \\
&= \frac{1}{2}\left[\left(\sum_{f=2}^{F-1}(-1)^{f-2}\boldsymbol{\Gamma}_{i_f;i_{f+1}}\right) - \boldsymbol{\Gamma}_{i_F;i_1} + \boldsymbol{\Gamma}_{i_1;i_2}\right] \overset{(48)_{i_2}}{\underset{\{inF\}}{\approx}} \mathbf{C}_{\widetilde{i_2}|\diamond} \quad , \forall\, F \text{ odd } \wedge\ 3 \leq F \leq I \quad (60)
\end{aligned}
$$

The consistency between direct and sequential error covariance estimates results directly from their common underlying definition of residual covariances in Eq. (20) and holds not only for the approximate formulations but similarly for the full expressions including error dependencies (compare Sect. 3.3.1). Thus, only one error covariance needs to be calculated with Eq. (39), or more general Eq. (48), while all other can be estimated from Eq. (49). Note that although even if only $\mathbf{C}_{\widetilde{i}}$ is calculated from the fully independent formulation in the basic polygon, the independence assumption among all pairs of datasets in the basic polygon remains.

Instead of using the sequential estimation for additional datasets $i$ with $F < i < I$, the error covariances could also be estimated by defining another pairwise-independent polygon, e.g. independent triangle $(i;j;k)$, with $k = \mathrm{ref}(j)$, $j = \mathrm{ref}(i)$. Because the definition of another independent triangle requires an additional independence assumption between $(i;k)$ (i.e. $\mathbf{X}_{\widetilde{i};\widetilde{k}} = 0 \Rightarrow \mathbf{D}_{\widetilde{i};\widetilde{k}} = 0$), this triangular estimate $\mathbf{C}_{\widetilde{i}|\triangleleft}$ from Eq. (39) differs from the sequential estimate $\mathbf{C}_{\widetilde{i}|\vdash}$ from Eq. (49) using its reference dataset ($\mathbf{C}_{\widetilde{j}} \to \mathbf{C}_{\widetilde{i}}$), where their absolute errors compare as follows:

$$
\left|\Delta\mathbf{C}_{\widetilde{i}|\vdash}\right| - \left|\Delta\mathbf{C}_{\widetilde{i}|\triangleleft}\right| \overset{(45),(54)}{=} \left|\Delta\mathbf{D}_{\widetilde{i};\widetilde{j}} - \Delta\mathbf{C}_{\widetilde{j}}\right| - \frac{1}{2}\left|\Delta\mathbf{D}_{\widetilde{i};\widetilde{j}} + \Delta\mathbf{D}_{\widetilde{i};\widetilde{k}} - \Delta\mathbf{D}_{\widetilde{j};\widetilde{k}}\right| \quad (61)
$$

The sequential estimation of an error covariance becomes favourable if the error covariance estimate of its reference dataset is as least as accurate as the assumed dependency between these two datasets $\left(\Delta\mathbf{C}_{\widetilde{j}} \to \Delta\mathbf{D}_{\widetilde{i};\widetilde{j}}\right)$. And the triangular estimation becomes favourable if the accuracy of the additional independence assumption is of the order of the difference between the uncertainties of the other two error dependencies $\left(\Delta\mathbf{D}_{\widetilde{i};\widetilde{k}} \to \Delta\mathbf{D}_{\widetilde{i};\widetilde{j}} - \Delta\mathbf{D}_{\widetilde{j};\widetilde{k}}\right)$; i.e. if the accuracy of the additional independence assumption is similar to that of the other two assumptions. This holds similarly for any polygonal estimation, where the additional independence assumption which closes the series of pairwise-independent datasets has to be of similar accuracy as the other independent assumptions.

Note that the absolute uncertainties presented here only account for uncertainties due to the underlying assumptions on error cross-statistics and not due to imperfect residual statistics occurring e.g. from finite sampling. A discussion of those effects for scalar problems can be found in Sjoberg et al. (2021).

## 5   Experiments

This section illustrates the capabilities for estimating full error covariance matrices of all datasets and some error dependencies. Three different experiments are presented with four collocated datasets ($I = 4$) on a 1D domain with 25 grid-points. For each experiment, the datasets are generated synthetically from 20,000 random realizations around the true value of $5.0$ with predefined error statistics. The experiments use predefined error statistics which are artificially generated to fulfill certain properties concerning error covariances and dependencies. Although also being generated by a finite sample of 20,000 realizations, these predefined error statistics are used to calculate residual statistics and thus represent the true error statistics that would be unknown in real applications. Here, the artificial generation of sampled true error statistics – denoted as "true error statistics" hereafter – allows for an evaluation of uncertainties of the "estimated error statistics" that are estimated with the proposed method. The experiments presented in this section are based on the symmetric estimations from residual covariances derived in Sect. 4 which are summarized in Algorithm A1. Similar results would be obtained from estimations from cross-covariances given in Algorithm A2, but this short illustration is restricted to a general demonstration using symmetric statistics only.

The error statistics of the four datasets consist of 10 matrices ($U_4 = 10$, compare Sect. 2): four error covariances (for each dataset, $I = 4$) and six error dependencies (between each pair of datasets, $N_4 = 6$). The three experiments differ in the true error dependency between datasets $(2; 3)$, which increases from experiment one to three. The general structures of the other true error statistics are the same among all experiments, however some local differences occur between the experiments due to the different dependencies and random sampling. The six residual covariances ($N_4 = 6$) between each pair of datasets are calculated from the true error statistics. Because these residual covariances are the statistical information that would be available for real applications were the truth remains unknown, they provide the input for the calculation of estimated error statistics.

From the six residual covariances given, all four error covariances and maximal two error dependencies can be estimated ($A_4 = 2$, compare Sect. 2). The remaining error dependencies that need to be assumed are set to zero for all experiments ("independent assumption") which is consistent to the mathematical formulation in Sect. 4.1 and 4.2. For each experiment, the error statistics were estimated with two different setups (subplot a and b of each figure, respectively). Both setups use a basic triangle between datasets $(1; 2; 3)$ to estimate their error covariances from Eq. (39). This triangular estimate assumes independence among those three datasets which is fulfilled in the first experiment but not in experiments two and three.

Based on this, the first setup uses a sequential estimation of the error covariance of the additional dataset $4$ w.r.t. its reference dataset 1 from Eq. (49) (ref$(4) = 1$); the independent assumption between those is fulfilled in all experiments. In contrast, the second setup uses another independent triangle between datasets $(1; 2; 4)$ to estimate the error covariance of dataset 4 from Eq. (39). In comparison to the sequential estimation, this additional triangular estimation requires an additional independent assumption between datasets $(2; 4)$ which is not fulfilled in any of the three experiments. Finally, both setups use the same formulation in Eq. (51) to estimate two error dependencies $(2; 4)$ and $(3; 4)$ based on the estimated error covariances of the two datasets involved $(2; 4)$ and $(3; 4)$, respectively. Note that the second setup is inconsistent because it assumes independence between $(2; 4)$ in the error covariance estimation of $4$, but uses this estimate to estimate the error dependency $(2; 4)$ which

was assumed to be zero before. The comparison between the two setups of each experiment shows the different effects of uncertainties in the underlying assumptions for sequential and direct error estimates.

In the following, the accuracy of the estimated error statistics from the two setups is evaluated for each experiment. In the first experiment in Sect. 5.1, the true error dependencies are constructed to fulfill the independent assumption in the basic triangle $(1; 2; 3)$. In experiment two and three in Sect. 5.2 and 5.3, a true error dependency between datasets $(2; 3)$ is introduced which is not in accordance with the independent assumption.

    The plots are structured as follows: Each subplot combines two covariance matrices; one shown in the upper-left part and
575 the other in the lower-right part. Because all matrices involved are symmetric, it is sufficient to show only one half of each matrix. The two matrices are separated by a thick gray diagonal bar and shifted off-diagonal so that diagonal variances are right above/below the gray bar, respectively. Statistics that might become negative are shown as absolute quantities in order to show them with the same color-code. In each row, the upper-left parts are matrices which are usually unknown in real applications (as they require the knowledge of the truth) and the lower-right parts are known/estimated matrices. The 1st row contains the
580 error dependencies and residual covariances of each dataset pair. Here, gray asterisks in the upper-left subplot indicate that these error dependency matrices are assumed to be zero in the estimation. The 2nd row contains the true and estimated error covariances and dependencies. The 3rd row gives the absolute difference between the true and estimated matrices. Note that the lower-right part of each subplot in the 3rd row does not contain any data.

## 5.1   Uncertainties in additional dependencies

Fig. 2 shows the error statistics of the first experiment where only true error dependencies are generated between datasets $(2; 4)$ and $(3; 4)$ (upper-left part of 1st row in Fig. 2a and b). This is in accordance to the estimation from the first setup shown in Fig. 2a which assumes independence in the basic triangle $(1; 2; 3)$ and between datasets $(1; 4)$ (independent assumptions indicated by gray asterisks). In contrast, the second setup shown in Fig. 2b requires an independent assumption between datasets $(2; 4)$ which is violated in this experiment. Thus, this experiment demonstrates the effects of uncertainties in this
additional assumption.

    By construction, the true error dependency matrices within the basic triangle – i.e. between $(1; 2)$, $(1; 3)$, $(2; 3)$ – and along the sequential estimation between $(1; 4)$ are zero in this experiment (upper-left part of 1st row, column 1-4 in Fig. 2a and b). Because the first setup only assumes independence of these dataset-pairs, it is able to estimate all four error covariance matrices and the two error dependency matrices between $(2; 4)$ and $(3; 4)$ accurately. Thus, the estimated error statistics match exactly
the true ones (2nd row in Fig. 2a) and their absolute difference is zero (upper-left part of 3rd row in Fig. 2a).

    In contrast, the additional triangular estimate in the second setup assumes an additional independence between datasets $(2; 4)$ which is not fulfilled. This neglected error dependency affects the triangular estimation of the error covariances of dataset 4, which is underestimated by half the neglected dependency as given in Eq. (45). This agrees with the experimental results shown in Fig. 2b where the neglected error dependency$(2; 4)$ with diagonal values around 1.2 (orange colors in upper-left part of 1st
row, column 5) induces an absolute uncertainty of the estimated error covariance 4 with diagonal values around 0.6 (purple colors in 3rd row, column 4). The sign of the uncertainty which corresponds to the underestimation can be seen from comparing

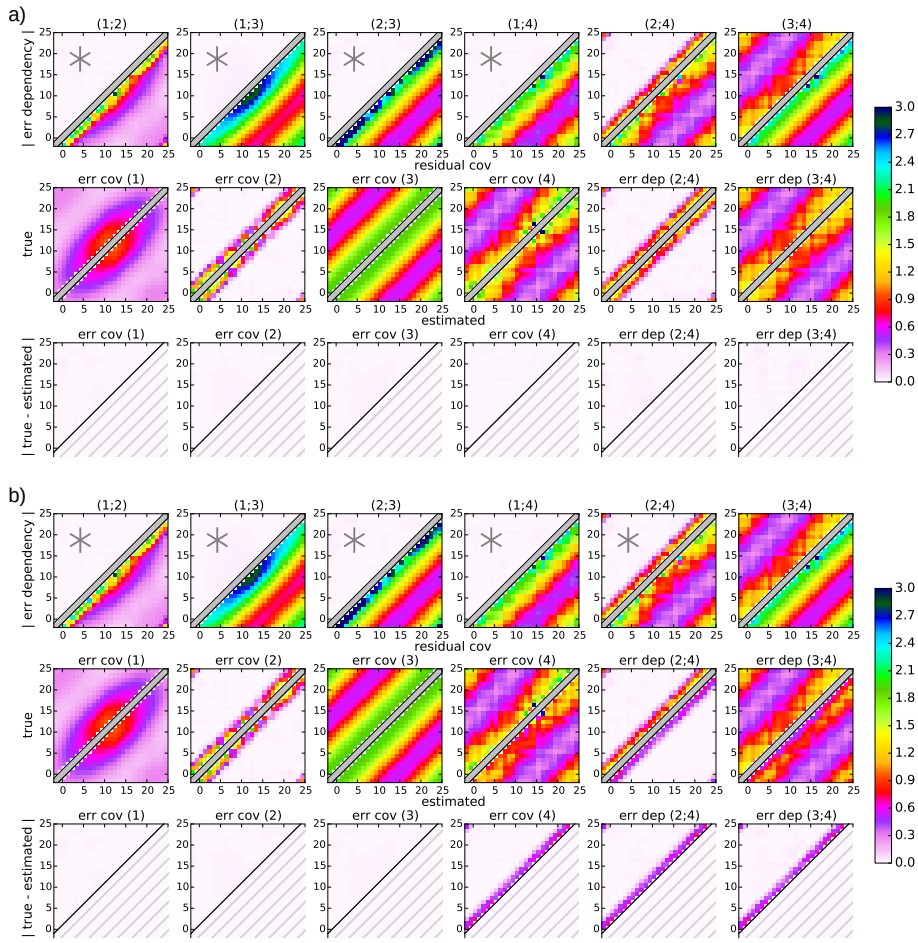

**Figure 2.** Experiment 1: Covariance matrices for 4 datasets ($I = 4$) with true dependencies of datasets $(2;4)$ and $(3;4)$. Datasets $(1;2;3)$ build the basic triangle. Dataset 4 is estimated (a) from its reference dataset 1 ("sequential estimation") and (b) from an additional independent triangle $(1;2;4)$ ("triangular estimation"). For each subplot, gray asterisks in the upper-left part of the 1st row indicate that these error dependencies are assumed to be zero in the estimation. Note that the lower-right part of each subplot in the 3rd row does not contain any data.

the true and estimated error covariances matrices of dataset 4 (2nd row, column 4). This uncertainty in the error covariances estimate of dataset 4 also affects the subsequent estimates of the error dependencies $(2;4)$ and $(3;4)$ which are expected to transfer the uncertainty of the error covariances with the same amplitude as given in Eq. (57). This can be confirmed by Fig. 2b where the uncertainties of the two estimated error dependencies equal the uncertainty of error covariance 4 (3rd row, column 4 and 5-6) and thus the dependency estimates are underestimated by half the neglected dependency $(2;4)$ (sign of uncertainty visible in 2nd row, column 5-6).

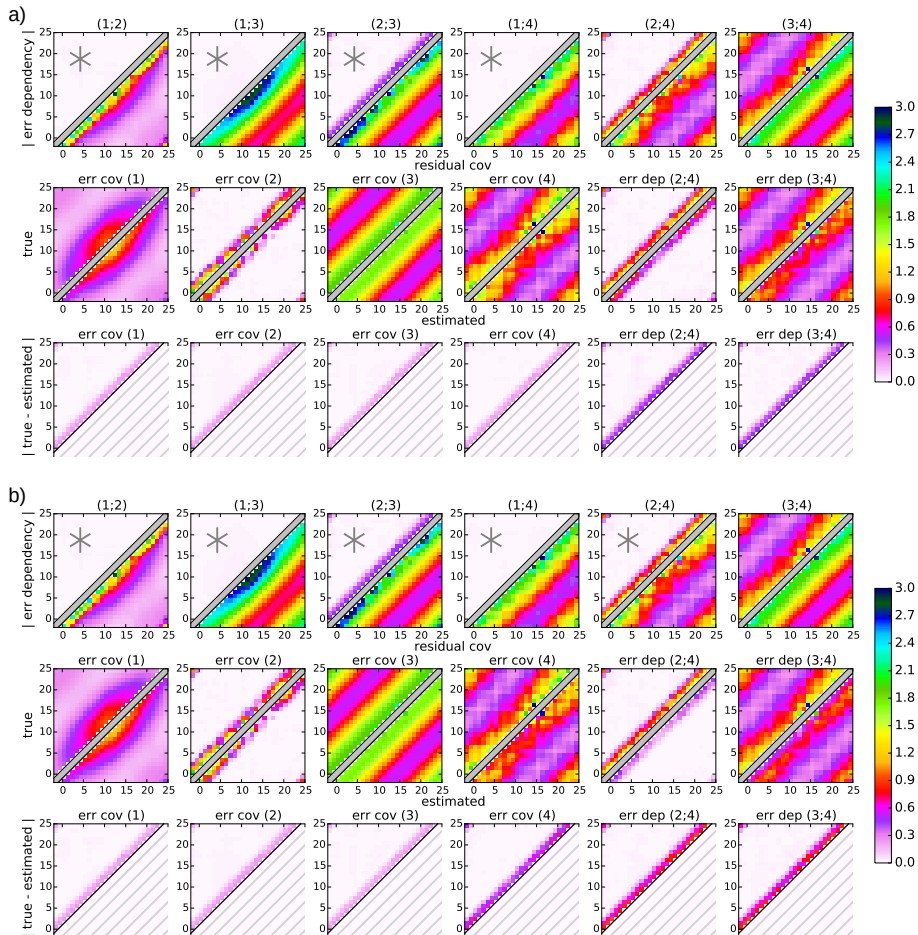

**Figure 3.** Experiment 2: Covariance matrices for 4 datasets ($I = 4$) with true dependencies of datasets (2;3), (2;4) and (3;4). As in Fig. 2, but with a neglected dependency in the basic triangle between datasets (2;3).

This experiment demonstrates the potential to estimate accurately complete error covariances and some dependencies for more than three datasets, if the underlying assumptions are sufficiently fulfilled. Note that this accurate estimation is independent of the complexity of the statistics like spatial variations or correlations. It also shows that an inaccurate independence assumption in an error covariance – here in the additional triangular estimation – may introduce uncertainties to all subsequent estimates of error covariances and dependencies, which is in accordance to the theoretical formulations above. The comparison of the two setup demonstrates the advantage of the sequential estimation for more than three datasets compared to only using triangular estimations.

## 5.2 Small uncertainties in basic triangle

Fig. 3 and 4 show the error statistics of the second and third experiment, respectively, where the independent assumption in the basic triangle is violated by introducing a non-zero dependency between datasets $(2; 3)$. The remaining true error statistics are the same as in the first experiment. Thus, both experiments have in total three non-zero error dependencies between datasets $(2; 3)$, $(2; 4)$ and $(3; 4)$, where the neglected dependency$(2; 3)$ is increased from experiment two to experiment three (upper-left part of 1st row in Fig. 3 and Fig. 4). These experiments demonstrate the effects of uncertainties in the basic triangle on the error estimates with the two setups.

Because both setups use the independent triangle $(1; 2; 3)$, the non-zero error dependency$(2; 3)$ violates this independence assumption and induces the same uncertainties in their error covariance estimates for both setups (compare Eq. (45) ). Comparing the estimated error covariance matrices of datasets 1, 2 and 3 with the true matrices in Fig. 3a and b shows that all three matrices are similarly affected. While the magnitude of uncertainties is the same (3rd row, column 1-3), their sign differs between the datasets which is in accorance to Eq. (45). For the two datasets involved 2 and 3, the neglected positive dependency$(2; 3)$ is transferred with the same sign, leading to an underestimation of their error covariances (row 2, column 2-3). In contrast, the impact on the error covariance of the remaining dataset in the triangle 1 is reversed, leading to an overestimation of the true error covariance (2nd row, column 1). As expected from Eq. (45), the magnitude of uncertainty of the three estimated error covariances with diagonal elements around 0.4 (light purple colors in 3rd row, colomn 1-3) is half the neglected error dependency with diagonal elements around 0.2 (dark purple colors in upper-left part of 1st row, column 3).

The two different setups differ in the estimation of the error covariance of dataset 4 which affects the estimated dependencies$(2; 4)$ and $(3; 4)$ as described in the first experiment. For the sequential estimation of error covariance 4 in Fig. 3a (first setup), the uncertainty of its reference error covariance 1 is transferred with same amplitude but opposite sign (compare Eq. (54) ), resulting in an underestimation of the error covariance matrix 4 (2nd and 3rd row, column 4). For the additional triangular estimation from $(1; 2; 4)$ in Fig. 3b (second setup), the uncertainty of error covariance 4 remains the same as in the first experiment where the independent triangle was accurate (2nd and 3rd row, column 4 of Fig. 3b vs. Fig. 2b). This is because the accuracy of a triangular estimation of a error covariance is only dependent on the assumed error dependencies between the dataset-pairs – which are accurate in this experiment –, but not on the other error covariance estimates (compare Eq. (45) ).

For both setups, the uncertainties in the two estimated error dependencies$(2; 4)$ and $(3; 4)$ are the sum of the uncertainties of the error covariance estimates of the two datasets involved, i.e. $(2; 4)$ and $(3; 4)$, respectively. For the first setup in Fig. 3a, the two error dependencies are underestimated by the same amplitude as the neglected error dependency$(2; 3)$ (upper-left part of 1st row, column 3 vs. 2nd and 3rd row, column 5-6) because of its impact on both error covariance estimates, with half amplitude each (2nd and 3rd row, column 2-4). For the second setup in Fig. 3b , the two estimated error dependencies$(2; 4)$ and $(3; 4)$ are affected by both neglected error dependencies$(2; 3)$ and $(2; 4)$ due to their impact on the two error covariance estimates involved $(2; 4)$ and $(3; 4)$, respectively. Because the two uncertainties in the error covariances sum up, the estimated error dependencies are underestimated by half the sum of the two neglected error dependencies (upper-left part of 1st row, column 3 and 5 vs. 2nd and 3rd row, column 5-6).

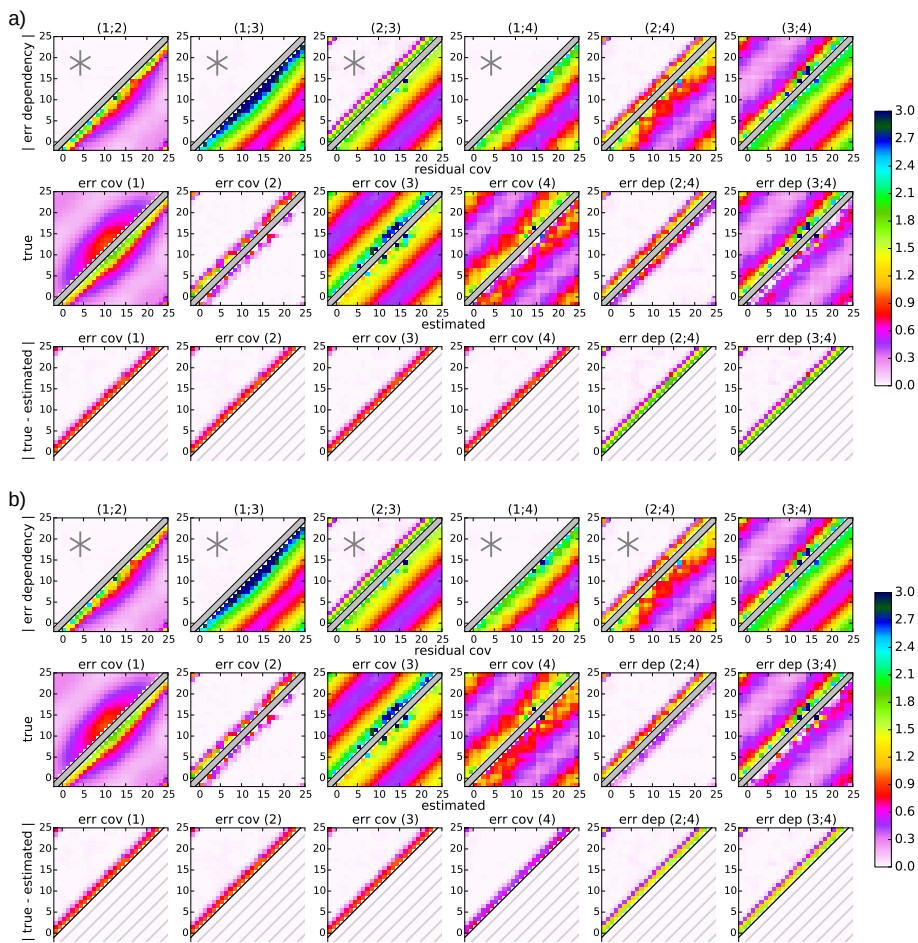

**Figure 4.** Experiment 3: Covariance matrices for 4 datasets ($I = 4$) with true dependencies of datasets (2;3), (2;4) and (3;4). As in Fig. 2, but with an increased dependency in the basic triangle between datasets (2;3).

Consequently, the sequential estimation of the additional dataset 4 is more accurate in this experiment because the uncertainties in the basic triangle $(1;2;3)$ are smaller than the uncertainty in the assumed dependency$(2;4)$ required for the additional triangular estimation, which can also be seen from Eq. (61).

## 5.3 Large uncertainties in basic triangle

This changes in the third experiment in Fig. 4, where the neglected dependency$(2;3)$ in the basic triangle is larger than the neglected dependency$(2;4)$ (upper-left parts of 1st row, column 3 and 5). Note that the increased error dependency$(2;3)$ is even larger than the true error covariance 2 at some locations (upper-left parts of 1st row, column 5 and 2nd row, column 2). For the first setup in Fig.4a, it can be seen that the increased uncertainty of the error covariance estimates in the basic triangle affects the estimates of all error statistics; the uncertainty of all estimated error statistics is increased proportional

to the increase in the neglected error dependency$(2;3)$. As in the second experiment, the uncertainty amplitude is half the neglected error dependency for all error covariance estimates and equals the neglected error dependency for the estimated error dependencies$(2;4)$ and $(3;4)$ (upper-left part of 1st row, column 3 and 3rd row of Fig. 4a vs. Fig. 3a).

The same holds for the error covariances estimates in the basic triangle $(1;2;3)$ in the second setup in Fig. 4b. In contrast, the additional triangular estimation of the error covariances of dataset $4$ remains again the same as in the other two experiments. Because the independent assumption in the additional triangle $(1;2;4)$ is more accurate than the basic triangle $(1;2;3)$ in this experiment, the additional triangular estimation of error covariance $4$ is more accurate than the sequential estimation (2nd and 3rd row, column 4 of Fig. 4a and b). If the other two error covariances $1$ and $2$ had also been estimated from the additional triangle $(1;2;4)$ instead of the basic triangle $(1;2;3)$, their estimation would also be more accurate (not shown).

The more accurate error covariance estimate of dataset $4$ with the second setup also leads to more accurate estimates of the two error dependencies$(2;3)$ and $(3;4)$, because of the summation of the two error covariance estimates involved (2nd and 3rd row, column 5-6 of Fig. 4a and b). In this particular example, the uncertainty of the error dependency$(2;4)$ is even larger than the true dependency (upper-left parts of 1st and 3rd row, column 5 of Fig. 4a and b), leading to negative dependencies for both estimates (lower-right part of 2nd row, column 5 of Fig. 4a and b). Similarly, the estimated error dependence matrix$(3;4)$ looses its diagonal-dominance, where the diagonal elements are almost zero but the more distant dependencies remain positive and similar to the true values (lower-right part of 2nd row, column 6 of Fig. 4a and b). This behaviour is caused by the different spatial correlation scales of the two datasets $3$ and $4$ and might give an indication for inaccurate assumptions in real applications. Note that the error dependency$(2;4)$ estimated with the second setup is more accurate in this experiment despite its inconsistency concerning the assumption of zero error dependency$(2;4)$ in the estimation of error covariance $4$. However, because of their negative dependency estimates, the independent assumption would be more accurate than the actual estimates from both setups in this case.

The large variation of uncertainties of error estimates from the two setups among the different experiments demonstrates the importance of selecting an appropriate setup for the error estimation problem which will be discussed in Sect. 6.2.

## 6 Conceptual summary and guidelines

This section provides a summary of the statistical error estimation method proposed in this study focusing on its technical application. Section 6.1 summarises the general assumptions and provides rules for the minimal conditions to solve the problem including an illustrative visualisation. Section 6.2 formulates guidelines for the selection of an appropriate setup of datasets under imperfect assumptions. An algorithmic summary for the calculation of error statistics from residual covariances and cross-covariances, respectively, is given in Appendix A.

## 6.1 Minimal conditions

This section provides a conceptual discussion of different conditions which need to be fulfilled in order to be able to solve the error estimation problem. The discussion is based on the previous sections, but formulated in a qualitative way without providing mathematical details.

For error statistics that need to be assumed, their specific formulation may have different forms. The easiest and most common assumption is to set their error correlations and thus the error cross-covariances and dependencies to zero. This assumption used in Sect. 4.1 and 4.2 is equivalent to the 3CH and TC methods. However, any non-zero error statistics can be defined and used in the general form which is summarized in Appendix. A. This also includes assuming error statistics as function of other error statistics including the ones estimated during the calculation. The only restriction is that all assumed error statistics must be fully determined by other error statistics or predefined values.

The number of error statistics that can be estimated for a given number of datasets ($N_I$) was introduced in Sect. 2. However not every possible choice of error statistics to be estimated provides a solution, which was also observed by Vogelzang and Stoffelen (2021) in the scalar case. The following discussion only considers setups where all error covariances and as many error cross-statistics as possible are estimated.

In the first step, some error covariances need to be estimated directly "from scratch", i.e. with no other error covariances available. Given the basic formulation of residual covariances in Eq. (20), a single error covariance ($C_i$) can only be eliminated when the other one ($C_j$) is replaced. Because every replacement of an error covariance of the same form introduces another error covariance, all other error covariances can only be removed if the final replacement introduces again the initial error covariance ($C_i$).

However, the resulting equation that involves a closed series of residuals can not always be solved for the initial error covariance. For less than three residuals involved ($F < 3$), the estimation of error covariances requires additional assumptions (compare Sect. 2). Because of the changing sign of error covariances in the equation, the initial error covariance ($\mathbf{C}_{\widetilde{i}}$) cancels out and cannot be eliminated if the number of involved residuals is even (compare Sect. 3.3.1). Note that the equation could then be used to estimate one error dependency, thus the number of estimated error statistics remains consistent with Sect. 2.

In addition, the error cross-covariances or dependencies between each involved dataset-pair have to be assumed in order to close the estimation problem. Thus the initial error covariance can only be estimated from a closed series of $F$ datasets with pairwise-assumed error cross-covariances or dependencies ($\mathbf{D}_{\widetilde{i_f};\widetilde{i_{f+1}}}$, $\mathbf{D}_{\widetilde{i_F};\widetilde{1}}$), where the number of involved datasets ($F$) is odd and larger or equal three (compare Sect. 4.2.1).

In the second step, all remaining error covariances can be estimated sequentially from their residual to a prior dataset – denoted as reference dataset – with previously estimated error covariance (compare Sect. 4.2.2). This estimation also requires the assumption of the error cross-covarianes or dependencies related to the residuals involved. And finally, the error cross-statistics which are not required in the estimation of error covariances ($A_I > 0$) can be estimated from their respective residual covariances (compare Sect. 4.2.3).

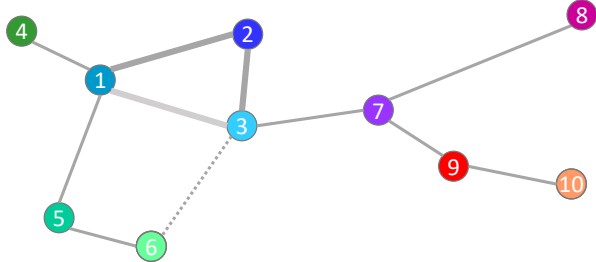

**Figure 5.** Independence tree: Illustrative example of assumed error dependencies (gray lines) between 10 datasets (colored dots). The assumed dependencies in the basic triangle (1;2;3) are indicated by thicker lines. An alternative setup with an basic pentagon is indicated by using the dotted line (3;6) instead of the lighter gray line (1;3).

Based on this, two general rules for the setup of datasets can be formulated which ensure the solvability of the problem in the case where all error covariances and as many error cross-statisitcs as possible (cross-covariances or dependencies) are estimated:

(i) all error cross-statistics along a closed series of dataset-pairs, with the number of involved datasets odd and larger or equal three, are needed (this closed series of datasets is called "basic polygon" or "basic triangle" in case of three datasets), and

(ii) at least one error cross-statistic of each additional dataset to any prior datasets is needed (this prior dataset is called "reference dataset" of the referring additional datasets).

Previously, Vogelzang and Stoffelen (2021) observed that some setups for four and five datasets do not produce a solution for the problem, but without discussing the general requirements. The limited solvability was also found by Gruber et al. (2016) for four datasets, who came up with an unnecessarily strong requirement that each dataset has to be part of an independent triangle.

An illustrative example of assumed dependencies for $I = 10$ datasets is visualized in Fig. 5. Note that this is one of many possible setups which are determined by the two rules above. The error dependencies among three datasets (1;2;3) are needed to be assumed ("basic triangle"). Then, one error dependency of each additional dataset $i > 3$ to any prior dataset $j$ (with $j < i$) is assumed ("sequential estimation"). Alternatively, the basic triangle could be replaced e.g. by a basic polygon of five datasets ("basic pentagon": 1;2;3;5;4), if the dependency(3;5) is assumed instead of the dependency(3;1).

## 6.2   Selection of setup

The general rules given in Sect. 6.1 allow for multiple different setups of datasets which all solve the error estimation problem. However in real applications, there might be significant differences in estimated error statistics from different setups as observed e.g. by Vogelzang and Stoffelen (2021) in the scalar case. The optimal selection is specific for each application and may depend on several requirements related to the actual purpose or use (e.g. available knowledge, accuracy of each estimate). This section

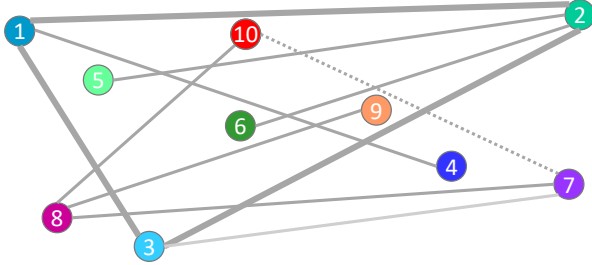

**Figure 6.** Improved independence tree: As Fig. 5, but with modified setup for more accurate error estimates. Distances between datasets represent the accuracy of assumed dependencies between the error statistics. While locations are the same, the numbers and colors of the datasets has been changed according to the modified setup. An alternative setup with an additional independent triangle is indicated by using the dotted line (7;10) instead of the lighter gray line (3;7).

provides some general guidelines on the selection of an appropriate setup among the various possible solutions w.r.t. the uncertainties introduced by statistical assumptions.

The relative accuracy of an error covariance estimate is proportional to the ratio between the residual covariance $\mathbf{\Gamma}_{i;j}$ and the absolute uncertainty $\Delta\mathbf{D}_{\tilde{i};\tilde{j}}$ of the assumed error dependency, which can be interpreted as being similar to a signal-to-noise ratio. In other words, the larger the residual covariance and the better the absolute estimate of the error dependency to the reference dataset, the more accurate is the estimated error covariance. Because uncertainties in error estimate do not necessarily sum up for a large basic polgon or along a branch of the independence tree (compare Sect. 4.2.4), a large residual-to-dependency ratio w.r.t. of the assumed cross-statistics is more important than a low number of intermediate datasets. In order to achieve sufficiently accurate estimates, the setup of datasets should be selected according to the expected accuracy of estimated dependencies which minimize the residual-to-dependency ratio for each datasets:

$$\max_j \left( \frac{\mathbf{\Gamma}_{i;j}}{\Delta\mathbf{D}_{\tilde{i};\tilde{j}}} \right) \ : j \to \mathrm{ref}(i) \quad \Longleftrightarrow \quad \min_j \left( \Delta\rho_{\tilde{i};\tilde{j}} \right) \ : j \to \mathrm{ref}(i) \quad , \forall\, i \tag{62}$$

The maximal residual-to-dependency ratio is equivalent to the minimal uncertainty in normalized error correlations $\Delta\rho_{\tilde{i};\tilde{j}} := \frac{\Delta\mathbf{D}_{\tilde{i};\tilde{j}}}{\sqrt{\mathbf{C}_{\tilde{i}}\,\mathbf{C}_{\tilde{j}}}}$. For example, if the error correlation of one dataset to another is known to some degree of accuracy, this dataset is well suited as reference dataset. If assumed error dependencies are set to zero, the dataset to which the independence assumption is most certain should be selected as reference dataset. Supposing that distances between datasets indicate their expected degree of independence in the independence tree, the setup visualized in Fig. 5 is not an appropriate selection. An example for an alternative setup is shown in Fig. 6, which is expected to provide more accurate error estimates.

While uncertainties in the basic polygon only contribute half, they affect the estimations of error statistics of all datasets (compare Sect. 4.1.3, 4.2.4, and Sect. 5.2, 5.3). This has two implications: Firstly, the basic polygon which is defined as the closed series of dataset that have the smallest error correlations produces the smallest overall uncertainty w.r.t. all error estimates. Ideally, the basic polygon should be set as a closed series of datasets which are highly pairwise-independent – or at least with reasonably small dependencies among each pair.

Secondly, if another pairwise-independent polygon can be assumed for an additional dataset with similar accuracy as the dependency to its reference dataset, the additional error estimate may be more accurate using the direct estimation from this additional pairwise-independent polygon rather than the sequential estimation (compare Sect. 4.2.5 and 5.3). The additional pairwise-independent polygon does not need to be connected to the basic polygon and may also have multiple independence branches, thus acting as additional basic polygon. For example, in the setup shown in Fig. 6, the estimation of dataset 7 is sensitive to the dependency(3;7) to its reference dataset, and less sensitive to dependencies in the basic triangle(1;2;3). If the dependency(7;10) could be assumed with higher accuracy than these dependencies, the error covariances of dataset 7 can alternatively be calculated from the independent triangle(7;8;10) and the independence assumption between (3;7) can be dropped. Thus, multiple independence trees can be defined around multiple separated basic triangles or basic polygons.

Furthermore, it is also possible to average the estimated error statistics of a dataset from multiple pairwise-independent polygons similar to an application of the N-cornered hat method (N-CH, e.g., Sjoberg et al., 2021) for an arbitrary subset of datasets. This setup builds an overestimated problem which requires the assumption of more error dependencies than the minimal requirements (compare Sect. 2 and 6.1). However, it might be beneficial if multiple pairwise-independent polygons containing the same dataset can be estimated with similar accuracy. In this case, potential uncertainties in the assumptions are expected to be reduced by the average over similar accurate estimates. Also an extension to weighted averages of different estimations is possible, where the weights reflect the expected accuracy of each estimation formulation w.r.t. the others.

# 7 Conclusions

Despite the generalized matrix-formulation, the main features of the presented approach are (i) its generality defining a flexible setup for any number of datasets according to the specific application, (ii) its optimality w.r.t. a minimal number of assumptions required, and (iii) its suitability to include expected non-zero dependencies between any pair of datasets. In contrast, the scalar N-CH method (N-cornered hat method) averages all estimates of each dataset which is equivalent to assuming that the independence assumption among each dataset triplet is fulfilled with the same accuracy. However, this is not the case for most applications to geophysical datasets. For example, Rieckh et al. (2021) applied the N-CH method to multiple atmospheric model and observational datasets and discussed neglected levels of independence between different datasets, which are expected to vary significantly. Pan et al. (2015) tried to account for such variations by clustering the datasets into structural groups; which however requires more assumptions than necessary and makes the result highly sensitive to the selected grouping. In contrast, the method presented here provides an optimal and flexible approach to handle multiple datasets with different levels of expected independence. Depending on the specific application, the estimation may be based on the minimal number of assumptions required or a (weighted) average over any number of estimations with similar expected accuracies.

An important application of the presented method is expected to be numerical weather prediction (NWP), where short-term forecasts from multiple national centers can be used to estimate error statistics required for data assimilation. In contrast to previous statistical methods, potential dependencies among the forecasts, i.e. due to the assimilation of similar observations, can be considered in the error estimation and even explicitly quantified. Future work will show how this statistical approach

compares to state-of-the-art background error estimates based on computation-expensive Monte-Carlo- or ensemble-methods. While the presented method can be formulated to provide symmetric error covariances, there remains a risk that negative values might occur for real applications due to inaccurate assumptions or sampling uncertainties.

In comparison to a-posteriori methods which statistically estimate optimal error covariances for data assimilation, an a-priori error estimation of collocated datasets has three main advantages: (i) optimal error statistics are calculated analytically without requiring an iterative minimization including multiple executions of the assimilation, (ii) complete covariance matrices provide spatially-resolved fields of error statistics at each collocated location including spatial- and cross-species correlations, and (iii) error statistics of all datasets are estimated without selecting one dataset as reference. This enables the consideration of more

than two datasets in the assimilation. Given sufficiently estimated error statistics, the final analysis w.r.t. to all datasets will be closer to the truth than any analysis between two datasets only. Thus, the rapidly increasing number of geophysical observations and model forecast enables improved analyses through increasingly overlapping datasets, where optimal error statistics can be calculated for example with the method presented here. Especially the possibility to estimate optimal error cross-covariances between datasets provides important information for data assimilation, where the violation of the independence assumption

remains a major challenge (Tandeo et al., 2020).

However, current data assimilation schemes are not suited for multiple overlapping datasets and cross-errors between datasets are assumed to be negligible. In contrast, the statistical error estimation method presented in this study is explicitly tailored to multiple datasets which cannot be assumed to be independent. Thus, the estimated error covariances are not consistent with assimilation algorithms assuming (two) independent datasets. If the estimated error dependencies among all assimilated datasets

are small, the independence assumption may be regarded as sufficiently fulfilled. The error estimation method then provides error covariances for assimilation and information on the accuracy of the independence assumption. Otherwise, generalized assimilation schemes are need to be developed for a proper use of this additional statistical information in data assimilation. Although increasing their complexity, such generalized assimilation schemes enable fundamental improvements in terms of an optimal analysis from multiple datasets w.r.t. their error covariances and cross-statistics.

*Data availability.* The error statistics of the three synthetic experiments will be provided by the authors upon request.

## Appendix A: Algorithms

The general estimation procedure of error statistics for $I \geq 3$ datasets is summarized in Algorithm A1 and A2. The algorithms require respectively, residual covariances or cross-covariances among all $I$ datasets (calculated from residual statistics) and $I$ assumed error dependencies or cross-covariances. Based on this, the first error covariance matrix is calculated with a polygonal estimation. Then, error statistics of the remaining datasets are calculated sequentially in an iterative procedure; introducing a new dataset $i$ with given residual statistics (covariances or cross-covariances) to dataset ref$(i)$ for each $i \in [2, I]$ with ref$(i) < i$. Note that this is equivalent to a direct estimation of all error covariances in the basic polygon and a sequential estimation of the additional error covariances of datasets $i > F$ (compare Sect. 4.2.5).

Algorithm A1 is formulated for symmetric statistic matrices, where error covariances $\texttt{errcov}(i;:,:)$ of each dataset $i$ and error dependency matrices $\texttt{errdep}(i;j;:;:)$ between each pair $(i;j)$ are estimated from symmetric residual covariances $\texttt{rescov}(i;j:;:;)$. In this algorithm, the generalized formulation of a basic polygon of $F \leq I$ residuals, for any odd $F \geq 3$, is used for the estimation of the first error covariance. In Algorithm A2, the error covariance- and cross-covariance matrices $\texttt{errcross}(i;j;:;:)$ of each pair $(i;j)$ are estimated from residual cross-covariances $\texttt{rescross}(i;j;i;k;:;:)$ between $(i\text{-}j;i\text{-}k)$. Here, the third dataset $k$ in the residual cross-covariances can be freely selected and does not affect the accuracy of the estimates (compare Sect. 4.2.4). This algorithm uses a basic triangle as example for a basic polygon for the estimation first error covariance. Each operation applies element-wise to each matrix-element indicated by the last two indices $(:;:)$, where matrices may contain different locations of the same quantity as well as different fields for multiple quantities of any dimension (=multivariate covariances). Transposed matrices w.r.t. the two location indices are indicated by $[\,]^T$.

The equations relate to the general exact formulations which requires some error dependencies or cross-covariances to be given (compare Sect. 3). The explicit calculation of the error cross-statistics (dependencies or cross-covariances) is not needed if only error covariances are of interest. In theory, both algorithms provide the same error estimations (compare Sect. 3.2.3). The decision to estimate error statistics from residual covariances (Algorithm A1) or cross-covariances (Algorithm A2) depends on the availability of residual statistics, the need for symmetric estimations of error covariances – which is only intrinsically guaranteed in Algorithm A1 –, and the need for estimating asymmetric components of error cross-covariances – which can only be estimated with Algorithm A2 (compare Sect. 3.3.1). Note that the generalized basic polygon can also be used for the estimation of the first error covariance in Algorithm A2.

*Author contributions.* AV developed the approach, derived the theory, performed the experiments, and wrote the manuscript. RM supervised the work, and revised the manuscript.

*Competing interests.* The authors declare that they have no conflict of interest.

**Algorithm A1** Iterative calculation of error covariances and dependencies for $I$ datasets from residual covariances with a general basic polygon of $F \leq I$ datasets.

---

**Require:** $\mathtt{rescov}(i; \mathrm{ref}(i); :, :) \; \forall \, i \in [2, I], \quad \mathtt{rescov}(\mathrm{F}; 1; :, :)$

**Require:** $\mathtt{errdep}(i; \mathrm{ref}(i); :, :) \; \forall \, i \in [2, I], \quad \mathtt{errdep}(\mathrm{F}; 1; :, :)$

   *– first datatset –*

   $\mathtt{errcov}(1; :, :) \leftarrow 0.5 \cdot \Big[ \Big( \sum_{f=1}^{F-1} (-1)^{f-1} \cdot \mathtt{rescov}(\mathrm{f+1}; \mathrm{f}; :, :) \Big) + \mathtt{rescov}(\mathrm{F}; 1; :, :)$

                     $+ \Big( \sum_{f=1}^{F-1} (-1)^{f-1} \cdot \mathtt{errdep}(\mathrm{f+1}; \mathrm{f}; :, :) \Big) + \mathtt{errdep}(\mathrm{F}; 1; :, :) \Big]$      *{∼ Eq. (29)}*

   *– loop over datasets –*

   **for** $i = 2, I$ **do**

      $\mathtt{errcov}(i; :, :) \leftarrow \mathtt{rescov}(i; \mathrm{ref}(i); :, :) + \mathtt{errdep}(i; \mathrm{ref}(i); :, :) - \mathtt{errcov}(\mathrm{ref}(i); :, :)$      *{∼ Eq. (25)}*

      *– remaining cross-statistics –*

      **for** $j = 1, i - 1$ **do**

         **if** $j \neq \mathrm{ref}(i)$ **then**

            $\mathtt{errdep}(i; j; :, :) \leftarrow \mathtt{errcov}(i; :, :) + \mathtt{errcov}(j; :, :) - \mathtt{rescov}(i; j; :, :)$      *{∼ Eq. (30)}*

         **end if**

         $\mathtt{errdep}(j; i; :, :) \leftarrow \mathtt{errdep}(i; j; :, :)$      *{∼ Eq. (14)}*

      **end for**

   **end for**

---

*Acknowledgements.* The authors thank Olivier Talagrand as editor, as well as Ricardo Todling and the two anonymous reviewers for their exceptionally thoughtful and valuable feedback on the manuscript.

**Algorithm A2** Iterative calculation of error covariances and cross-covariances for $I$ datasets from residual cross-covariances with a basic triangle of three datasets.

**Require:** $\texttt{rescross}(i; \text{ref}(i); i; j; :; :; :)$, $\quad \texttt{rescross}(\text{ref}(i); i; \text{ref}(i); j; :; :; :) \, \forall \, i \in [2, I], j \neq \text{ref}(i), j \neq i$, $\quad \texttt{rescross}(1; 2; 1; 3; :; :; :)$

**Require:** $\texttt{errcross}(i; \text{ref}(i); :; :; :) \, \forall \, i \in [2, I]$, $\quad \texttt{errcross}(1; 3; :; :; :)$

> **for** $i = 2, I$ **do**
>> $\texttt{errcross}(\text{ref}(i); i; :; :; :) \leftarrow \texttt{errcross}(i; \text{ref}(i); :; :; :)^T$ $\qquad \{\sim Eq.\ (10)\}$
>
> **end for**
>
> *– first dataset –*
>
> $\texttt{errcov}(1; :; :; :) \leftarrow \texttt{rescross}(1; 2; 1; 3; :; :; :) + \texttt{errcross}(1; 3; :; :; :) + \texttt{errcross}(2; 1; :; :; :) - \texttt{errcross}(2; 3; :; :; :)$ $\qquad \{\sim Eq.\ (32)\}$
>
> *– loop over datasets –*
>
> **for** $i = 2, I$ **do**
>> $\texttt{errcov}(i; :; :; :) \leftarrow \texttt{rescross}(i; \text{ref}(i); i; j; :; :; :) + \texttt{rescross}(\text{ref}(i); i; \text{ref}(i); j; :; :; :)$
>> $\qquad\qquad - \texttt{errcov}(\text{ref}(i); :; :; :) + \texttt{errcross}(i; \text{ref}(i); :; :; :) + \texttt{errcross}(\text{ref}(i); i; :; :; :)$ $\qquad \{\sim Eq.\ (34)\}$
>>
>> *– remaining cross-statistics –*
>>
>> **for** $j = 1, i - 1$ **do**
>>> **if** $j \neq \text{ref}(i)$ **then**
>>>> $\texttt{errcross}(i; j; :; :; :) \leftarrow \texttt{rescross}(\text{ref}(i); i; \text{ref}(i); j; :; :; :) - \texttt{errcov}(\text{ref}(i); :; :; :)$
>>>> $\qquad\qquad + \texttt{errcross}(\text{ref}(i); j; :; :; :) + \texttt{errcross}(i; \text{ref}(i); :; :; :)$ $\qquad \{\sim Eq.\ (36)\}$
>>>>
>>>> $\texttt{errcross}(j; i; :; :; :) \leftarrow \texttt{errcross}(i; j; :; :; :)^T$ $\qquad \{\sim Eq.\ (10)\}$
>>>
>>> **end if**
>>
>> **end for**
>
> **end for**

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
