# Peer review of "How far can the statistical error estimation problem be closed by collocated data?"

_EGUsphere, 2022_

## Referee Comment (RC1)

**How far can the error estimation problem in data assimilation be closed by collocated data?**

by A. Vogel and R. Ménard

*Reviewer's Report by R. Todling*

**Recommendation**: Accept with revisions.

The present work revisits the problem of estimating relevant statistical information for data assimilation by employing residual-based collocation methods. The work presents a generalization of three-cornered-hat (3CH) and traditional collocation methods establishing precise statements about how many relevant statistics can be inferred from a given number of datasets that include different estimates of sought out quantities. The work also provides for an understanding of what one can expect to estimate given various dependencies among differing datasets. The work is full of insight and provides illustration from idealized settings.

I my view the work is sound, mathematically meaningful and represents an important contribution to the field. I recommend some revision in the text, mostly minor points. I do have a couple of broad comments which are presented below before the minor points.

Main concerns:

**A** My first main concern refers to the wide use of the word *innovation*. Although I understand the main motivation behind the work is data assimilation applications, the framework in the present article is general - it deals with second moment statistics of variables regardless of the context in which these appear. The difference fields appearing in equations such as (4) are what would be better referred to as *residuals*. I strongly suggest replacement of the word innovation with residual. Indeed, unless commenting on related works truly using innovations (e.g., Tandeo et al. 2020; Todling et al 2022; and others), most of the time the authors can omit either of the words; especially once stated initially that the covariances and cross-covariances dealt with in the work are really *residual* covariances and cross-covariances.

**B** Another issue for me relates to notation. It starts around line 139, when the authors introduce eq. (4). I understand that subscripts such as $i - j$ represent differences (residuals) derived from estimates $\mathbf{x}_i$ and $\mathbf{x}_j$ for datasets $i$ and $j$ respectively. It is never said that in such case, $i$ must never equal $j$, as it would not make sense to calculate residuals of a dataset against itself. An alternative notation for the subscripts of $\boldsymbol{\Gamma}$ would be $i, j; k, l$ - in this, the pairs being use to calculate the difference vectors making up $\boldsymbol{\Gamma}$ are separated by a semicolon. This notation would also be more consistent with the notation in eq. (5), when the truth is introduced and the matrix represents an error covariance.

**C** I believe that in the considerations in section 3, and specially section 4, a relevant possibility for how possibly to get the precise estimates when dependence exists among the datasets has been overlooked. The others talk a lot about what happens

when the dataset are truly independent, or when there is dependence. But never really point out the important case when the dependent contribution in, say, eq. (26) vanishes as a whole. That is when, the datasets are such that

$$\mathbf{D}_{\tilde{i};\tilde{j}} + \mathbf{D}_{\tilde{k};\tilde{i}} - \mathbf{D}_{\tilde{j};\tilde{k}} = \mathbf{0}$$

The above is at the core of the Todling et al. (2022) findings. That is when the three datasets, (i,j,k) here, are connected in some very special (particular) way, that is, through the DA system, (i.e., these being the analysis, background and observation). I believe the possibility of finding special combination of datasets (for which the above holds) should be discussed in your work. Clearly, datasets that combine in such particular way are rather rare.

Minor points:

1. I wonder about the title a little bit. The work here is very general, I know data assimilation is the primary motivation for the application of the method(s) discussed and the work done in this work. But the fact is that the technique here applies generally, and independently of DA. Perhaps a better title could be: "When do collocated data provide for a closed error estimation problem?"

2. l. 24: "arises the question if" should read "raises the question whether".

3. With extreme respect to the authors, I recommend a close revision of the writing itself. I find use of very uncommon English words, which although not incorrect, seem rather usual, e.g., exemplary, approximative; there are also a number of articles, and other wording in the paper that could benefit from some attention. I try to point out some of these in what follows, but I only show so much. I can anticipate that most of the time a word like "exemplary" is better read as "example", and "approximative" as "approximate".

4. l. 30: "since decades" should read "for decades".

5. . 30: "recently be exploited" should read "recently been exploited".

6. l. 31: The works of Nielsen et al. (2022) and Todling et al. (2022) were done concurrently, basically with either being unaware of the details of the other. I believe your statement here would more fairly read "Nielsen et al. (2022) and Todling et al. (2022) were the first to independently use the generalized ...".

7. ll 33-35: I think the authors need to rephrase what comes after "However". A better sentence would perhaps be: "... framework. Indeed, Todling et al. (2022) shows that when the corners of G3CH are identified with the observation, background and analysis of variational assimilation procedures, only under the assumptions of optimality does the method obtains closed estimates for the three corners; in general, the problem cannot be closed." Notice this comment goes along comment C made above.

8. The "dot" notation used in eq. (4), and many others, has not been introduced. I authors should state that
$$\mathbf{x} \cdot \mathbf{y} = \mathbf{x}\mathbf{y}^T$$

9. Eqs. (6) and (7) do not require the term exposed in their second equalities.

10. Eq (6): I confess the notation of using superscripts in the equality signs in various equations is new to me. I have mixed feelings about it, but regardless of my feelings, the authors should explain what these are after they first appear in eqs. (6) and (7). That is, somewhere it should be stated that "superscripts and subscripts in the equality signs indicate what other equations were used to arrive at the given result".

11. l. 47: "since decades" should read "for decades".

12. l. 53: "additionally" should read "additional".

13. l. 70: word "approximative" would better read "approximate". The word approximative appears numerous times, I believe all instances would read better as "approximate" instead.

14. l. 76: word "exemplary" should be removed in this case - without loss of clarity.

15. l. 79: "requiring the knowledge" would better read "requiring knowledge".

16. ll. 84-85: "analyses or any" would better read "analysis and any".

17. l. 138: there needs to be an explanation (definition) for the meaning of the subscript notation with the standing up bar, as in $i|r$, $i$ given $r$? Why do you need this notation here when it is not used anywhere else in the article?

18. Eq. (4): I find it somewhat unnecessary to have the notation include the points $(p, q)$ explicitly. Given that $\mathbf{x}$ is a vector quantity the $(p, q)$ indexes can be implicitly understood. In fact, most of your eqs. do not carry them.

19. ll. 173-174: This sentence should be moved to the definition statements made around eq. (4).

20. ll. 190-193: This would better read: "Thus, the covariance of any two datasets consists of the sum of the independent covariances associated with each dataset minus the error dependency covariance; this latter corresponding to the sum of the error covariances associated with each dataset, eq. (16)."

21. l. 240: "formulated as sum" should read "formulated as a sum".

22. paragr. ll. 243-247: you might want to add here that all the works mentioned in this paragraph associate what the authors call "dependent contribution" with the *cross-covariances of the random errors.*

23. l. 243: please replace "by Eq. (26)" with "in Eq. (26)".

24. ll. 260-261: There are lots of instances of the word "formulation" in these two sentences; the author might want to work on the text.

25. ll. 278-279: This sentence is very confusing. I think I understand what the authors mean, but I suggest rephrasing.

26. ll. 306-308: I believe the authors want to say that the "*independence* assumption *resembles* the innovation consistency *statement* of data assimilation, where the innovation covariance ..." —- notice that here, this is one of those places where the word *innovation* can and should be used.

27. l. 314: word "neglectable" should read "negligible".

28. l. 325: word "between" should be replaced with "among". Please notice there are other instances of "between all three" that should be revised accordingly.

29. l. 337: "a error" should read "an error"

30. l. 339: "allow a comparison" should read "allow for a comparison".

31. l. 396: " Eq. (32) is follows" should read " Eq. (32) follows".

32. l. 418: spell: "beeing".

33. l. 461: 'An discussion" should read "A discussion".

34. l. 464: "provides an exemplary demonstration" would better read "provides some demonstration".

35. l. 485: word "calculated" is not needed.

36. l. 495: "exemplary demonstrated" should better read "illustrated".

37. l. 511: "does only affect" should read "affect only".

38. Fig. 2a: why are the errors (bottom row) so diagonally dominant? Shouldn't these bottom panels be more like random patterns everywhere? Why aren't the errors in the diagonal of the order of the off-diagonal terms?

39. Figs. 2b and 3: why are the errors (bottom row plots) so asymmetrically dominant?

40. l. 515: "it's" should read "its".

41. l. 516: "requirement of assumptions" should better read simply "assumptions".

42. l. 522: "and 4.2 and is" should better read "and 4.2 is".

43. l. 523: please spell out "Apx".

44. l. 530: "solution of the problem" reads better as "solution to the problem".

45. l. 533: word "whoever" should be removed.

46. l. 533: "came up with a too strong requirement" would better read "came up with an unnecessarily strong requirement that *ldots*"

47. l. 539: "estimates" should be in the singular.

48. l. 575: duplicate "of the".

49. p. 24, conclusions: in regards to your last two paragraphs, and the generality of the method as you propose here, can you comment on the viability of the method to be used for, say, deriving estimates of background errors by using a combination of background fields from multiple DA systems. For example, suppose we collect 6-hour forecasts from IFS, GFS, CMC, GMAO, US Navy, etc — there is some dependency among all these datasets since for most part the background fields (short-range forecasts) are based on the assimilation of similar observations in all these systems — do you think your method would be able to infer reliable and perhaps better forecast error estimates than what we typically get from the NMC or ensemble methods? The same question can be made wrt analysis errors. Can you comment on this - if not in the paper, at least here to this reviewer.

---

## Referee Comment (RC2)

One major question in data assimilation is to determine the statistics of the errors affecting the data to be assimilated. It is those statistics that define the weights to be given to the data in the assimilation. However, they can never be fully determined without external hypotheses, *i. e.* hypotheses that cannot be objectively validated on the basis of the data alone.

The authors present and discuss an approach that is appropriate for the situations in which a number of sets of collocated data are available. They consider only second-order statistical moments (first-order moments, *i. e.* biases, are also required, but their identification is an independent problem). Covariances and cross-covariances of the differences ('*innovations*') between those different sets of data are known from the data, and are linearly related to the statistics of the underlying data errors. By appropriate *a priori* specification of a number of those data errors statistics (the external hypotheses), all, or part, of the remaining error statistics are solution of a system of linear matrix equations. That approach, which originated from the so-called *three cornered hat* (*3CH*) method, has been used in a number of applications, but not much so far in assimilation of geophysical data.

Given $I$ sets of collocated data, the unknowns (covariances and cross-covariances of data errors) are in number $U_I = (1/2) I (I+1)$ (Eq. 2 of the paper). Concerning the innovations, their second order moments are not independent, and they are combinations of only $N_I = (1/2) I (I-1)$ of them (l. 95). This leads to a linear system of $N_I$ matrix equations with $U_I$ unknowns (that system is basically expressed, although in what is to me a cursory passing remark, by Eq. 22). The degree of underdeterminacy of the system is $U_I - N_I = I$. The view that is suffices to choose *a priori I* of the unknowns to close algebraically the system is correct for $I = 3$, but not necessarily for larger values (at least if, as the authors want, no error covariance is specified *a priori*). The purpose of the authors, in addition to stating precisely and discussing the problem, is to determine minimal conditions for its solution (… *what are the minimal and optimal conditions to solve the problem?*, l. 65). They also present numerical results obtained from synthetic data.

The article is instructive, and certainly contains material that is worth publishing. But it needs in my opinion substantial improvement.

1. My main comment is that I have found it very difficult to understand the very logic of the paper (and I am actually still not even sure I have fully understood). A succinct analysis shows that, for $I > 1$, system (22) (strictly speaking, a system of $N_I$ equations which is equivalent to 22) is of rank $N_I$, which shows that by appropriately choosing $I$ of the unknown error covariances and cross-covariances, one can obtain the values of all the other unknowns. My understanding is that the authors show that these $I$ *a priori* chosen error covariances and cross-covariances cannot be chosen arbitrarily, and that there are constraints in that choice (especially in the case considered by the authors, in which only cross-covariances are to be chosen *a priori*). If it is so, I think it must be stated more explicitly.

2. Subsection 6.1 (*Minimal conditions*) contains what I understand are the authors' main conclusions. That Subsection states two conditions (ll. 527-529) that are presented as the minimal conditions ensuring existence and uniqueness of the solution of system (22) (at least, that is my understanding)

- (i) *all three error dependencies between one triple of datasets are needed (this triple of independent datasets is called "basic triangle")*

- (ii) *at least one error dependency of each additional dataset to any prior datasets is needed*

I is not clear to me whether these two conditions are mathematically exact (if yes, explain more clearly where they are proven in the paper, or give a reference ; if not, say clearly they are only reasonable conjectures).

3. I find that Sections 3 and 4, although they boil down to elementary algebraic manipulations, are intricate and difficult to follow.

*a*. Eq. (22) expresses the basic links between innovation and error statistics (denoted respectively $\Gamma$ and X). Although algebraically obvious, it is the crux of the method, and should be stressed more strongly as such.

*b*. The derivation of Eq. 23 (ll. 211-213) is strange, since it suggests (l. 211) that one must go through the error statistics **X** to obtain the equation, while the latter expresses necessarily links between the innovation statistics $\Gamma$, and can be easily be proved directly.

*c*. Eq. (34) is also strange in that in purports to show the 'equivalence' between two expressions for the error dependencies **D**. Those two expressions are basically obtained from Eq. (22), and the reader would think they must necessarily be the same. I presume the authors want to stress that inappropriate choice of the *a priori* chosen error cross-covariances can lead to inconsistencies. But, rather than demonstrating consistency, it would be preferable to show an explicit example of inconsistency. Actually, my understanding is that Eqs (39-40) precisely show an example of inconsistency. If I am mistaken about the significance of Eq. (34), say more explicitly what that significance is.

*d*. The authors, for some unspecified reason, consider only the 'error-dependencies', *i.e.* the symmetric part of the error cross-correlations matrices (Eq. 20), and ignore the anti-symmetric part. Why so ?

*e*. It is not clearly said why the number of independent innovation covariances and cross-covariances is equal to $N_I = (1/2)\ I\ (I\text{-}1)$ (that is rather simple, but must be said more clearly). The mutual dependence between those quantities is expressed by Eq. 23, the significance of which (in addition to my remark *b* above) should be stressed more strongly.

These are only examples of places that can cause confusion in the mind of a reader who is a newcomer to the approach described in the paper, as elementary as that approach may fundamentally be. I think Sections 3 and 4 could be rewritten in a clearer and more concise way, with more stress on the logic of the approach and on the two fundamental aspects upon which it is based. First, that the observed innovation covariances and cross-covariances are redundant. Second, the basic link between between the innovation and errors covariances and cross-covariances, expressed by Eq. (22) (or any other equivalent equation for that matter).

4. And, for a final (but I think important) comment, any algebraic solution to system (22) will not be acceptable in then present context. It must also define a proper (symmetric non negative) global error covariance matrix (in particular, the estimated error covariance matrices $C_i$ of the various individual datasets, in addition to being symmetric, must be non-negative). The authors hardly mention this point. Do the conditions (i-ii) stated in subsection (6.1) lead to a proper global covariance matrix ? Since system (22) expresses necessary conditions between error and innovation variances and covariances, I presume that if the *a priori* specified variances and cross-covariances are compatible with a globally symmetric non-negative matrix that is itself compatible with the $\Gamma_{i,\,j\,;\,k\,l}$ 's (Eq. 22), the estimated

variances and cross-covariances will also be. I do not ask the authors to necessarily give a full answer to that question, but it should be clearly mentioned and at least briefly discussed. In particular, if the authors do not have a full answer to that question, it should clearly stated as remaining an open question.

It may that the response to some of the questions I raise above is available in the literature, in particular in the literature the authors mention. If so, please give precise references.

I would have a number of other comments, bearing on both scientific and editing aspects of the paper, but they are of lesser importance, and I will wait for a possible revised version for mentioning them.

---

## Editor Comment (EC1)

Two referees have now sent their reports on the paper, and the open discussion of the paper has been closed. The authors have been asked to post their responses to the referees' comments by 17 Jan 2023.

Referee 1 has let his name known, and is Ricardo Todling. He considers that the work presented in the paper *is sound, mathematically meaningful and represents an important contribution to the field*. He asks for minor revisions, and gives a number of suggestions to that end, some of which are rather broad, while others bear on editing aspects of the paper (in particular on the English).

Referee 2, who has remained anonymous, is more critical. He writes that he has not fully understood the very logic of the paper. What is the problem, beyond solving an underdetermined system of linear equations by *a priori* specification of the values of a sufficient number of unknowns ? He also raises the question of the symmetric non-negative character of the variance-covariance matrices produced by the estimation process described in the paper. He asks for major revision of the paper.

As Editor, I suggest to the authors to prepare of course their responses to the referees, and also (if they have not already done so) to prepare a revised version of their paper. This new version must take into account the questions, comments and suggestions of both referees. In case the authors disagree with a particular referee comment, or decide not to follow a particular suggestion, they will have to give explicitly their reasons for that in their responses.

As Editor, I will in any case not take any decision until the authors have responded to the referees' comments. The authors can in the meantime get in touch with me if they wish, either directly or through the public discussion.

---

## Author Comment (AC1)

**Response to Reviewer 1 - Ricardo Todling:**

We thank the reviewer for the thoughtful and detailed evaluation and valuable remarks. We hope that we could reply and adopt the manuscript in a sufficient way.

The present work revisits the problem of estimating relevant statistical information for data assimilation by employing residual-based collocation methods. The work presents a generalization of three-cornered-hat (3CH) and traditional collocation methods establishing precise statements about how many relevant statistics can be inferred from a given number of datasets that include different estimates of sought out quantities. The work also provides for an understanding of what one can expect to estimate given various dependencies among differing datasets. The work is full of insight and provides illustration from idealized settings.

I my view the work is sound, mathematically meaningful and represents an important contribution to the field. I recommend some revision in the text, mostly minor points. I do have a couple of broad comments which are presented below before the minor points.

**Main concerns:**

A  **My first main concern refers to the wide use of the word innovation. Although I understand the main motivation behind the work is data assimilation applications, the framework in the present article is general - it deals with second moment statistics of variables regardless of the context in which these appear. The difference fields appearing in equations such as (4) are what would be better referred to as residuals. I strongly suggest replacement of the word innovation with residual. Indeed, unless commenting on related works truly using innovations (e.g., Tandeo et al. 2020; Todling et al 2022; and others), most of the time the authors can omit either of the words; especially once stated initially that the covariances and cross-covariances dealt with in the work are really residual covariances and cross-covariances.**

Reply: This comment includes 2 points.

1) The replacement of the word "innovation" by "residual", which has been applied everywhere not explicitly referring to data assimilation (incl. Fig. 2-4 and Apx. A). Those sentence has been slightly modified to clarify the relation of the word "innovation" in data assimilation to "residuals" (ll.16-18, new count):

   *" A number of approaches to estimate optimal error statistics make use of residuals, i.e. the innovations between observation and background states in observation space (Tandeo et al., 2020), but the error estimation problem remains underdetermined. "*

   And under consideration of Minor comment 26 below (ll.327-328, new count):

   *" The independence assumption resembles the innovation covariance consistency of data assimilation, were the residual covariance between background and observation datasets - denoted as innovation covariance - ... "*

   In the labels of Fig. 2-4, "innovation" was replaced by "residual cov" and in addition the following other labels were extended to be more specific and consistent: "dependency" → "err dependency", "err(*)" → "err cov (*)", and "dep(*)" → "err dep (*)" (see Fig. 1

below, the other figures were modified accordingly, which are Fig. 2-4 in the new version of the manuscript).

2) The suggestion to omit either of the words. With great respect to the reviewer, we believe that it is important to keep the word "residual" (or "innovation") in the text. We see that the word appears quite often, but it the manuscript deals with error and residual statistics at the same time and a clear distinction among them is essential for the understanding.

B **Another issue for me relates to notation. It starts around line 139, when the authors introduce eq. (4). I understand that subscripts such as $i - j$ represent differences (residuals) derived from estimates $x_i$ and $x_j$ for datasets i and j respectively. It is never said that in such case, i must never equal j, as it would not make sense to calculate residuals of a dataset against itself. An alternative notation for the subscripts of $\Gamma$ would be $i, j; k, l$ - in this, the pairs being use to calculate the difference vectors making up $\Gamma$ are separated by a semicolon. This notation would also be more consistent with the notation in eq. (5), when the truth is introduced and the matrix represents an error covariance.**

Reply: This comment also includes 2 points.

1) As suggested, a note on non-zero residuals $j \neq i$ and $l \neq k$ was added to the definitions of residual cross-covariances (l.150, new count):

    " *Let $\Gamma_{i-j;k-l}$ be the residual cross-covariance matrix between dataset residuals $i - j$ and $k - l$ with $j \neq i$ and $l \neq k$, were …* "

    And similarly in the definitions of residual covariance, dependency, and asymmetry (l.157, l.176, and l.183, new count).

2) Concerning the index-notation of residual statistics, we decided to keep the current notation. In our point of view, the both notations for residuals are consistent with the tilde-notation for error statistics in Eq. (5). However, we find that the suggested comma-notation $\Gamma_{i,j;k,l}$ makes it difficult to distinguish between the two pairs of datasets as commas and semicolons look quite similar, especially for small subscripts. We choose the minus-notation $\Gamma_{i-j;k-l}$ because we think that the explicit formulation of the differences used makes the interpretation highly intuitive.

C **I believe that in the considerations in section 3, and specially section 4, a relevant possibility for how possibly to get the precise estimates when dependence exists among the datasets has been overlooked. The others talk a lot about what happens when the dataset are truly independent, or when there is dependence. But never really point out the important case when the dependent contribution in, say, eq. (26) vanishes as a whole. That is when, the datasets are such that**

$$\mathbf{D}_{\tilde{i};\tilde{j}} + \mathbf{D}_{\tilde{k};\tilde{i}} - \mathbf{D}_{\tilde{j};\tilde{k}} = 0$$

**The above is at the core of the Todling et al. (2022) findings. That is when the three datasets, (i,j,k) here, are connected in some very special (particular) way, that is, through the DA system, (i.e., these being the analysis, background and observation). I believe the possibility of finding special combination of datasets (for which the above holds) should be discussed in your work. Clearly, datasets that combine in such particular way are rather rare.**

Reply: Indeed, the special case of vanishing dependencies is an important point. We already had a short note in Sect. 4.1.3 stating that one dependent contribution might vanish even if all three dependencies are non-zero (compare l.382, new count). We extended this discussion w.r.t. the special case of data assimilation (ll.384-386, new count):

*" A special case was observed by Todling et al. (2022) who showed that the estimations of background, observation and analysis errors in a variational data assimilation system become exact if the analysis is optimal. In this particular case, no assumptions on dependencies are required because the optimality of the analysis induces vanishing dependencies. "*

In addition, the particular case of variational data assimilation referring to Todling et al. (2022) was also added in the introduction as suggested in Minor comment 7 (compare reply to this comment below).

**Minor points:**

1. **I wonder about the title a little bit. The work here is very general, I know data assimilation is the primary motivation for the application of the method(s) discussed and the work done in this work. But the fact is that the technique here applies generally, and independently of DA. Perhaps a better title could be: "When do collocated data provide for a closed error estimation problem?"**

   Reply: The words *"in data assimilation"* were removed from the title to express the generality of the work. The title reads now:
   *" How far can the statistical error estimation problem be closed by collocated data? "*

2. **l. 24: "arises the question if" should read "raises the question whether".**

   Reply: Corrected.

3. **With extreme respect to the authors, I recommend a close revision of the writing itself. I find use of very uncommon English words, which although not incorrect, seem rather usual, e.g., exemplary, approximative; there are also a number of articles, and other wording in the paper that could benefit from some attention. I try to point out some of these in what follows, but I only show so much. I can anticipate that most of the time a word like "exemplary" is better read as "example", and "approximative" as "approximate".**

   Reply: We corrected the two examples given here as well as all words given in the other comments below. In addition, we:

   - corrected some typos ("accuracies", "structural" in Sect. 1, "triangle" in Sect. 5)
   - did some corrections to achieve a consistent hyphen ("-") notation for words with two parts (e.g. "state-vector", "grid-points", "element-wise") and in listing of words (e.g. "covariance- and cross-covariance matrices")
   - we also removed repeating abbreviations "Eq." and "Sect." in listings of several equation or section numbers (e.g. "Eq. (36), (37), or (38)", l.410, new count)

4. **l. 30: "since decades" should read "for decades".**
   Reply: Corrected.

5. **l.30: "recently be exploited" should read "recently been exploited".**

   Reply: Corrected.

6. **l. 31: The works of Nielsen et al. (2022) and Todling et al. (2022) were done concurrently, basically with either being unaware of the details of the other. I believe your statement here would more fairly read "Nielsen et al. (2022) and Todling et al. (2022) were the first to independently use the generalized ...".**

   Reply: Modified accordingly.

7. **ll 33-35: I think the authors need to rephrase what comes after "However". A better sentence would perhaps be: "...framework. Indeed, Todling et al. (2022) shows that when the corners of G3CH are identified with the observation, background and analysis of variational assimilation procedures, only under the assumptions of optimality does the method obtains closed estimates for the three corners; in general, the problem cannot be closed." Notice this comment goes along comment C made above.**

   Reply: Rephrased accordingly, the sentence reads now (ll.34-36, new count):
   *" They show that when the corners of G3CH are identified with the observation, background and analysis of variational assimilation procedures, this particular error estimation problem can only be closed under the assumption of optimally. "*

8. **The "dot" notation used in eq. (4), and many others, has not been introduced. I authors should state that**
$$x \cdot y = xy^T$$

   Reply: Actually, $x_i(p)$ denotes one element of the dataset vector, thus the dot denotes a scalar multiplication with no need for a transposed. We removed the dot for all multiplications of vector-elements (Eq.(4)-(7),(19),(21)) and added the following note for clarification (l.156, new count):
   *" Note that $x_i(p)$ is a scalar element of the dataset vector. "*

9. **Eqs. (6) and (7) do not require the term exposed in their second equalities.**

   Reply: That's right. However, the authors believe that an explicit formulation of these quantities helps the reader when re-checking the definitions without reading the whole section.

10. **Eq (6): I confess the notation of using superscripts in the equality signs in various equations is new to me. I have mixed feelings about it, but regardless of my feelings, the authors should explain what these are after they first appear in eqs. (6) and (7). That is, somewhere it should be stated that "superscripts and subscripts in the equality signs indicate what other equations were used to arrive at the given result".**

    Reply: Although not being widely used, superscripts above equal signs have been used previously to indicate other equations used (e.g. Vogel and Elbern, 2021, GMD). Regarding the number of other equations that were used in some derivations in this manuscript (e.g. Eq. (34)) we decided to use this notation to help the reader following the derivations in a compressed way that does not affect the reading flow significantly. But we agree that the notation has to be introduced and thus removed the superscripts in Sect. 2 (in Eq. (2) and Eq. (3)) and added the following sentence were it first appears in Sect. 3 (l.161, new count):

" ... were numbers in parenthesis above an equal sign indicate other equations that were used to retrieve the right hand side. "

Concerning the use of indices of these superscripts, we removed these indices whenever not necessary (Eq. (27), (29), (45), (46), (47), (48), and (53)) and only kept them were the same equations were applied to different datasets in order to avoid potential confusion. In these cases, the descriptions of the equations were extended. For Eq. (25) (ll.245-247, new count):
" By combining the formulations of three residuals $\Gamma_{i-j}$, $\Gamma_{j-k}$, and $\Gamma_{k-i}$ between the same three datasets $i$, $j$, and $k$ and expressing each using Eq. (20), a single error covariance can be eliminated: ... "
and for Eq. (30) (l.283, new count):
" Two of the error cross-covariances in Eq. (29) can be rewritten by applying Eq. (29) to the error covariance of dataset $j$: ... "

We also added a description were these indices were used for the first time (ll.252-254, new count):
" ... were the indication of used equations above the equal signs are extended by indices which denote to which datasets this equation has been applied. For example, " $\stackrel{(20)_{ki}}{=}$ " indicates that the relation in Eq. (20) was applied to datasets $k$ and $i$ to achieve the right hand side. "

The meaning of subscripts of equal signs (indicating the assumptions used in this relation) was already described in the manuscript were there first appear (l.331. l.339, and l.414, new count), were l.331 slightly extended to:
" ... were " $\underset{\{in\}}{\approx}$ " indicates the assumption of independence between the two datasets, i.e. $X_{\tilde{i};\tilde{j}} = 0$. "

And similarly l.414:
" ... were " $\underset{\{inI\}}{\approx}$ " indicates the assumption of independence to the reference dataset, i.e. $X_{\tilde{i};\widetilde{ref(i)}} = 0$. "

11. **l. 47: "since decades" should read "for decades".**
    Reply: Corrected.

12. **l. 53: "additionally" should read "additional".**
    Reply: Corrected.

13. **l. 70: word "approximative" would better read "approximate". The word approximative appears numerous times, I believe all instances would read better as "approximate" instead.**
    Reply: Replaced everywhere were it appeared.

14. **l. 76: word "exemplary" should be removed in this case - without loss of clarity.**
    Reply: Removed.

15. **l. 79: "requiring the knowledge" would better read "requiring knowledge".**
    Reply: Corrected.

16. **ll. 84-85: "analyses or any" would better read "analysis and any".**

    Reply: "or" replaced by "and". "analyses" was kept in its plural form to remain consistent with the other listed items (l.87, new count).

17. **l. 138: there needs to be an explanation (definition) for the meaning of the subscript notation with the standing up bar, as in $i|r$, i given r? Why do you need this notation here when it is not used anywhere else in the article?**

    Reply: We agree that this notation might be confusing and is not important for the rest of the manuscript. We removed the explicit indication of the realization in the definition of the state vectors and deleted the equation in the 2nd sentence, which now reads (ll.147-148, new count):
    *" Suppose I datasets, each containing R realizations of spatio-temporally collocated state vectors $x_i \forall i \in [1, I]$. Without loss of generality, the following formulation uses unbiased state vectors with zero mean. "*

    Instead, we added a note on the meaning of the overbar w.r.t. realizations before Eq.(4):
    *" ... were each element $(p, q)$ is given by the expectation over all realizations: ... "*

    And after Eq.(5):
    *" ... and the overbar denotes the expectation over all R realizations. "*

18. **Eq. (4): I find it somewhat unnecessary to have the notation include the points (p; q) explicitly. Given that x is a vector quantity the (p; q) indexes can be implicitly understood. In fact, most of your eqs. do not carry them.**

    Reply: Although most equations refer to complete matrices, we believe that the complexity of the terms itself - especially when carrying multiple indices - justify the explicit definition of a single element of the matrix. While it might be obvious to some readers, the formulation of a single element avoids misunderstanding e.g. in Eq. (21).

19. **ll. 173-174: This sentence should be moved to the definition statements made around eq. (4).**

    Reply: The sentence was moved to the end of the introduction paragraph of this section, before Eq.(4) (now ll.48-49, new count).

20. **ll. 190-193: This would better read: "Thus, the covariance of any two datasets consists of the sum of the independent covariances associated with each dataset minus the error dependency covariance; this latter corresponding to the sum of the error covariances associated with each dataset, eq. (16)."**

    Reply: Rephrased based on the reviewer's suggestion. The sentence now reads (ll.201-203, new count):
    *" Thus, the residual covariance of any dataset pair consists of (i) the independent residual associated with sum of the error covariances of each dataset, minus (ii) the error dependency corresponding to the sum of their error cross-covariances. "*

21. **l. 240: "formulated as sum" should read "formulated as a sum".**

    Reply: Corrected.

22. **paragr. ll. 243-247: you might want to add here that all the works mentioned in this paragraph associate what the authors call "dependent contribution" with the cross-covariances of the random errors.**

    Reply: The following sentence was added (ll.264-265, new count):
    *" Note that in the literature, the dependent contribution in Eq. (26) is denoted as cross-covariances between the errors. "*

23. **l. 243: please replace "by Eq. (26)" with "in Eq. (26)".**

    Reply: Done.

24. **ll. 260-261: There are lots of instances of the word "formulation" in these two sentences; the author might want to work on the text.**

    Reply: Rephrased. The sentences read now (l.278-279, new count):
    *" The scalar formulation of Eq. (29) was previously given in Zwieback et al. (2012). Similarly to Eq. (26) from residual covariances, the number of formulations .... "*

25. **ll. 278-279: This sentence is very confusing. I think I understand what the authors mean, but I suggest rephrasing.**

    Reply: Rephrased. The sentence now reads (ll.296-297, new count):
    *" Note that the third dataset $i$ on the right hand side of Eq. (33) can be any other dataset ($i \neq j, i \neq l$). Thus for any $I > 2$, there are $I - 2$ formulations of each error cross-covariance $\mathbf{X}_{\tilde{j};\tilde{l}}$, which are all equivalent in the exact formulation. "*

26. **ll. 306-308: I believe the authors want to say that the "*independence* assumption *resembles* the innovation consistency *statement* of data assimilation, where the innovation covariance..." - notice that here, this is one of those places where the word *innovation* can and should be used.**

    Reply: Replaced accordingly; also at all other locations in the manuscript were "independent assumption" appeared. As suggested by the reviewer, the term innovation was kept here, with slight modification as described in reply to Main concern A.1.

27. **l. 314: word "neglectable" should read "negligible".**

    Reply: Replaced; also at all other locations in the manuscript were it appeared.

28. **l. 325: word "between" should be replaced with "among". Please notice there are other instances of "between all three" that should be revised accordingly.**

    Reply: Replaced whenever referring to three or multiple datasets.

29. **l. 337: "a error" should read "an error"**

    Reply: Corrected.

30. **l. 339: "allow a comparison" should read "allow for a comparison".**

    Reply: Corrected.

31. **l. 396: " Eq. (32) is follows" should read " Eq. (32) follows".**

    Reply: Typo. Replaced by "it follows" (l.418, new count).

32. **l. 418: spell: "beeing".**

    Reply: Corrected.

33. **l. 461: "An discussion" should read "A discussion".**

    Reply: Replaced.

34. **l. 464: "provides an exemplary demonstration" would better read "provides some demonstration".**

    Reply: Replaced with "illustrates" to be consistent with the comment on l.495 below (l.488, new count).

35. **l. 485: word "calculated" is not needed.**

    Reply: Removed.

36. **l. 495: "exemplary demonstrated" should better read "illustrated".**

    Reply: Replaced.

37. **l. 511: "does only affect" should read "affect only".**

    Reply: Replaced.

38. **Fig. 2a: why are the errors (bottom row) so diagonally dominant? Shouldn't these bottom panels be more like random patterns everywhere? Why aren't the errors in the diagonal of the order of the off-diagonal terms?**

    Reply: We are not entirely sure if we understand this comment the right way, we see two possible interpretations which we will both consider in the the following reply.

    (a) Looking at the bottom rows, the upper left part shows the absolute differences between true and estimated statistics. In the case of Fig. 2a, all assumptions are sufficiently fulfilled and the estimated statistics becomes equal to the true statistics. Indeed, what remains is some minor random noise. The reviewer might refer to the gray diagonal bar which separates the triangular matrices and has nothing to do with the fields itself. We see that the existence of the gray line might lead to miss-interpretation in the 3rd row and removed it accordingly. Additionally, we filled the lower right triangle with gray stripes to indicate that there is no data (rather than zero-values) shown in this part (see Fig. 1, the other figures were modified accordingly, which are Fig. 2-4 in the new version of the manuscript). The description of the plots in the manuscript was also extended for clarification (ll.500-502):

    *" Because all matrices involved are symmetric, it is sufficient to show only one half of each matrix. The two matrices are separated by a thick gray diagonal bar and shifted off-diagonal so that diagonal variances are right above/below the gray bar, respectively. "*

[Figure]

Figure 1: Covariance matrices for 4 datasets ($I = 4$) with true dependencies of datasets (2;4) and (3;4). Datasets (1;2;3) build the basic triangle. Dataset 4 is estimated (a) from its reference dataset 1 ("sequential estimation") and (b) from an additional independent triangle (1;2;4) ("triangular estimation").

(b) Otherwise, the reviewer might refer to the fact that differences between true and estimated statistics are only visible close to the diagonal in Fig. 2b,3,4. This is a result of the shape of error dependencies that were neglected in the estimation (upper part of 1st row, indicated by a gray asterisk) which are created to be proportional to the referring residual statistics (lower part of 1st row). Only the smaller amplitude might make the uncertainty appear more diagonal-dominant than the residual covariance, if this is what the reviewer is referring to.

39. **Figs. 2b and 3: why are the errors (bottom row plots) so asymmetrically dominant?**

Reply: This comment is closely related to the previous one. The gray diagonal bar together with the upper-triangular field and the missing data in the lower triangle might have appeared as asymmetric field. See reply to previous comment 38, the applied corrections should also clarify this point.

40. **l. 515: "it's" should read "its".**

Reply: Corrected.

41. **l. 516: "requirement of assumptions" should better read simply "assumptions".**

Reply: Replaced by (l.543, new count):
*" assumptions and requirements "*

42. **l. 522: "and 4.2 and is" should better read "and 4.2 is".**

Reply: Removed.

43. **l. 523: please spell out "Apx".**

Reply: Done; also were it appears elsewhere in the manuscript.

44. **l. 530: "solution of the problem" reads better as "solution to the problem".**

Reply: Replaced.

45. **l. 533: word "whoever" should be removed.**

Reply: Removed.

46. **l. 533: "came up with a too strong requirement" would better read "came up with an unnecessarily strong requirement that ldots"**

Reply: Replaced.

47. **l. 539: "estimates" should be in the singular.**

Reply: Corrected.

48. **l. 575: duplicate "of the".**

Reply: Removed.

49. **p. 24, conclusions: in regards to your last two paragraphs, and the generality of the method as you propose here, can you comment on the viability of the method to be used for, say, deriving estimates of background errors by using a combination of background fields from multiple DA systems. For example, suppose we collect 6-hour forecasts from IFS, GFS, CMC, GMAO, US Navy, etc - there is some dependency among all these datasets since for most part the background filds (short-range forecasts) are based on the assimilation of similar observations in all these systems - do you think your method would be able to infer reliable and perhaps better forecast error estimates than what we typically get from the NMC or ensemble methods? The same question can be made wrt analysis errors. Can you comment on this - if not in the paper, at least here to this reviewer.**

Reply: Thank you for this thoughtful suggestion. We added the following paragraph in the conclusions (ll.622-626, new count):
*" An important application of the presented method is expected to be numerical weather prediction (NWP) were short-term forecasts from multiple national centers can be used to estimate error statistics required for data assimilation. In contrast to previous statistical methods, potential dependencies among the forecasts, i.e. due to the assimilation of similar observations, can be considered in the error*

*estimation and even explicitly quantified. Future work will show how this statistical approach compares to state-of-the-art background error estimates based on computation-expensive Monte-Carlo- or ensemble-methods. "*

We decided to keep the discussion of how the presented method might compare to state-of-the-art estimates rather short in the manuscript. This will be left for a follow-up investigation were the method is actually applied to real geophysical datasets. The authors expect the main advantage of the proposed method to be its very low computation effort given that a large number of overlapping datasets are already available (eg. global gridded opertational forecasts from different weather centers worldwide which only need to be interpolated to a common grid). Another advantage that is also mentioned in the manuscript is the explicit estimation of error dependencies (or cross-covariances), which however require the development of novel data assimilation schemes.

Despite the need for collocated datasets, the main disadvantage of this method lies in its attempt to estimate statistical covariances only. The need for a large set of realizations which sample the same truth is expected to be a limitation for many real applications. This aspect will also be further discussed and different solutions will be proposed in the upcoming work.

---

## Author Comment (AC2)

**Response to Reviewer 2:**

We thank the reviewer for the insightful remarks and hope that we could reply and adopt the manuscript in a sufficient way.

One major question in data assimilation is to determine the statistics of the errors affecting the data to be assimilated. It is those statistics that define the weights to be given to the data in the assimilation. However, they can never be fully determined without external hypotheses, i. e. hypotheses that cannot be objectively validated on the basis of the data alone.

The authors present and discuss an approach that is appropriate for the situations in which a number of sets of collocated data are available. They consider only second-order statistical moments (first-order moments, i. e. biases, are also required, but their identification is an independent problem). Covariances and cross-covariances of the differences ('innovations') between those different sets of data are known from the data, and are linearly related to the statistics of the underlying data errors. By appropriate a priori specification of a number of those data errors statistics (the external hypotheses), all, or part, of the remaining error statistics are solution of a system of linear matrix equations. That approach, which originated from the so-called three cornered hat (3CH) method, has been used in a number of applications, but not much so far in assimilation of geophysical data.

Given $I$ sets of collocated data, the unknowns (covariances and cross-covariances of data errors) are in number $U_I = (1/2)I(I+1)$ (**Eq. 2** of the paper). Concerning the innovations, their second order moments are not independent, and they are combinations of only $N_I = (1/2)I(I-1)$ of them (l. **95**). This leads to a linear system of $N_I$ matrix equations with $U_I$ unknowns (that system is basically expressed, although in what is to me a cursory passing remark, by Eq. **22**). The degree of underdeterminacy of the system is $U_I - N_I = I$. The view that is suffices to choose a priori $I$ of the unknowns to close algebraically the system is correct for $I = 3$, but not necessarily for larger values (at least if, as the authors want, no error covariance is specified a priori). The purpose of the authors, in addition to stating precisely and discussing the problem, is to determine minimal conditions for its solution (… what are the minimal and optimal conditions to solve the problem?, l. **65**). They also present numerical results obtained from synthetic data.

The article is instructive, and certainly contains material that is worth publishing. But it needs in my opinion substantial improvement.

1. My main comment is that I have found it very difficult to understand the very logic of the paper (and I am actually still not even sure I have fully understood). A succinct analysis shows that, for $I > 1$, system (**22**) (strictly speaking, a system of $N_I$ equations which is equivalent to **22**) is of rank $N_I$, which shows that by appropriately choosing I of the unknown error covariances and cross-covariances, one can obtain the values of all the other unknowns. My understanding is that the authors show that these I a priori chosen error covariances and cross-covariances cannot be chosen arbitrarily, and that there are constraints in that choice (especially in the case considered by the authors, in which only cross-covariances are to be chosen a priori). If it is so, I think it must be stated more explicitly.

   Reply1: The reviewer is right in his interpretation. We intended to describe this important

aspect in Sect. 2, but we see now that it was not formulated clearly enough. We reformulated the referring paragraph, which now reads (ll.103-105, new count):

" *The set of error statistics to be estimated can generally be chosen according to the specific application, but it will be shown that there are some constraints. Based on the mathematical theory provided in the following sections, the actual minimal conditions to solve the problem will be discussed in Sect. 6.1.* "

We also added a new paragraph at the end of this section (ll.128-130):

" *Note that almost all numbers presented above apply to the general case were any combination of error covariances and cross-covariances may be given or assumed. While the interpretation of the numbers $I$, $N_I$ and $U_I$ remains the same in all cases, the only difference is the interpretation of DI which is less meaningful when also error covariances are assumed.* "

We ensured that the actual formulation of the minimal conditions (Sect. 6.1) already includes the information that those are formulated for the common case that only cross-covariances are assumed (see eg. l.553). Additionally, we added a note that similar conditions holding for other cases at the end of the subsection (ll.569-570, new count):

" *Note that similar conditions can be derived for cases were also error covariances are given or assumed, which is not part of this paper.* "

2. **Subsection 6.1 (Minimal conditions) contains what I understand are the authors' main conclusions. That Subsection states two conditions (ll. 527-529) that are presented as the minimal conditions ensuring existence and uniqueness of the solution of system (22) (at least, that is my understanding)**

   (i) **all three error dependencies between one triple of datasets are needed (this triple of independent datasets is called "basic triangle")**

   (ii) **at least one error dependency of each additional dataset to any prior datasets is needed**

   **I is not clear to me whether these two conditions are mathematically exact (if yes, explain more clearly where they are proven in the paper, or give a reference ; if not, say clearly they are only reasonable conjectures).**

   Reply2: Based on the mathematical derivations in Sect. 3 and 4, the two minimal conditions are logical conclusions that are valid for all number of datasets. They provide the necessary conditions for the existence of a solution; which is demonstrated by giving the explicit formulations of error statistics in Sect. 4.1.1, 4.2.1 and 4.2.2. The uniqueness of this solution is achieved when - and exactly when - the required assumptions are accurate; i.e. the assumed error cross-covariances and dependencies vanish (compare equations in Sect. 4.1.3, 4.2.3 and 4.2.4). A refering statement was added in the manuscript (ll.557-561, new count):

   " *These two requirements are a logical summary of the mathematical derivations in Sect. 3 and 4 and are valid for all number of datasets $I \leq 3$. They provide the necessary conditions for the existence of a solution under the given assumptions (compare Sect. 4.1.1, 4.2.1, and 4.2.2)). Optimality and uniqueness of this solution w.r.t. different formulations and setups are achieved when - and exactly when - the required assumptions are accurate (i.e. vanishing uncertainties of assumed error statistics in Sect. 4.1.2, 4.1.3, 4.2.3, and 4.2.4).* "

   A rigorous mathematical proof of these conditions would require another level of math in this paper, which is mainly tailored to a theoretical-geophysical rather than a mathematical audience. Therefore, we decided to not include a rigorous proof in this paper.

Note that the given minimal conditions apply for setups *"were all error covariances and some error dependencies (or cross-covariances) are estimated"* (l.553, new count).

3. **I find that Sections 3 and 4, although they boil down to elementary algebraic manipulations, are intricate and difficult to follow.**

   a. **Eq. (22) expresses the basic links between innovation and error statistics (denoted respectively $\Gamma$ and X). Although algebraically obvious, it is the crux of the method, and should be stressed more strongly as such.**

      Reply3a: We agree with this comment and thank the reviewer for pointing this out. We pointed out the importance of Sect. 3.2 and specifically Eq. (20) at several locations in the manuscript:

      – In the description of the content of the study in Sect. 1 (ll.69-72, new count):
        *" ... the mathematical formulation for non-scalar error matrices is derived in Sect. 3 and Sect. 4, respectively. The derivation is based on the formulation of residual statistics as function of error statistics which is introduced in Sect. 3.2. While the exact formulations of error statistics in Sect. 3.3 remain underdetermined in real applications ... "*

      – In the description of the general framework in Sect. 2 (ll.96-97, new count):
        *" The main idea now is to express the known residual statistics as function of unknown error statistics (Sect. 3.2) and combine these equations to eliminate single error statistics (Sect. 3.3, Sect. 4). "*

      – In the introduction of Sect. 3 (ll.137-138, new count):
        *" The expression of residual statistics as function of error covariances and cross-covariances in Sect. 3.2 provides the basis for the subsequent mathematical theory. "*

      – In the introduction of Sect. 3.3 (ll.239-240, new count):
        *" These formulations are based on the relations between residual and error statistics in Eq. (20) and Eq. (22). "*

      – And in the section itself along with the discussion of the equation (ll.216-217, new count):
        *" This formulation of residual statistics as function of error statistics provides the basis for the complete theoretical derivation of error estimates in this study. "*

   b. **The derivation of Eq. 23 (ll. 211-213) is strange, since it suggests (l. 211) that one must go through the error statistics X to obtain the equation, while the latter expresses necessarily links between the innovation statistics Gamma, and can be easily be proved directly.**

      Reply3b: We see that this formulation is confusing. We rearranged the equation that the individual steps to show the equivalence between innovation statistics in a clearer way (Eq. (23), new count).

      The decision to use error statistics to show the equivalence was made upon its context in the work which is based on the relation between innovation and error statistics (also pointed out by the reviewer e.g. in his comment 3a above).

   c. **Eq. (34) is also strange in that in purports to show the 'equivalence' between two expressions for the error dependencies D. Those two expressions are basically obtained from Eq. (22), and the reader would think they must necessarily be the same. I presume the authors want to stress that inappropriate choice of the a priori chosen error cross-covariances can lead to inconsistencies.**

**But, rather than demonstrating consistency, it would be preferable to show an explicit example of inconsistency. Actually, my understanding is that Eqs (39-40) precisely show an example of inconsistency. If I am mistaken about the significance of Eq. (34), say more explicitly what that significance is.**

Reply3c: Indeed, the equivalence of the two expressions is not surprising. The formulation based on innovation covariances originates form Eq. (20) which is a special case of the basic equation for cross-covariances Eq. (22) (compare also Reply3a). However, the authors decided to show this equivalence to explicitly demonstrate the consistency of the two estimates of error dependencies. While this was obvious to the reviewer, we think that it will be useful to other readers. Given the fact that previous literature considers only one of the two formulations, either based on residual covariances or cross-covariances, this relation might not be clear to everyone. At the same time, the equivalence is an important result of the work, which refers to the last paragraph of the reviewers comment 3 (see below). We added the following comment to the description of Eq. (34) (ll.304-306, new count):

*" This consistency applies to the exact formulations of all symmetric error statistics (error covariances and dependencies) and results directly form the fact that the basic formulation of residual covariances in Eq. (20) is a special case of the formulation of residual cross-covariances in Eq. (22). "*

d. **The authors, for some unspecified reason, consider only the 'error-dependencies', i.e. the symmetric part of the error cross-correlations matrices (Eq. 20), and ignore the antisymmetric part. Why so ?**

Reply3d: The reference to manuscript is not entirely clear. In Sect.3 and 4, the authors carefully verified that asymmetric cross-covariance matrices and asymmetry matrices are considered wherever possible and if not, a related statement were made (eg. ll.254, ll.403, old version). What remains are the experiments in Sect. 5 which indeed only discuss the symmetric statistics (error covariances and dependencies) and not asymmetric error cross-covariances. This was the author's choice in order to restrict the section to it's main purpose of demonstrating the ability to retrieve error covariances as well as dependencies. Actually, the presented experiments were created with symmetric statistics which only enables the compressed visualization in Fig. 2-4 (showing only half of each matrix). The demonstration could be extended to explicitly show asymmetric error statistics, but we believe that showing dependencies is more convenient for this purpose (shorter and more intuitive); especially under consideration that the manuscript is already quite long. We added a referring sentence in the description of the experiments (ll.496-498, new count):

*" Similar results would be obtained from estimations from cross-covariances in Algorithm A2, but this short illustration is restricted to a general demonstration using symmetric statistics only. "*

and in the explanation of the plots (ll.500-501, new count):

*" Because all matrices involved are symmetric, it is sufficient to show only one half of each matrix. "*

e. **It is not clearly said why the number of independent innovation covariances and cross-covariances is equal to $N_I = (1/2)I(I-1)$ (that is rather simple, but must be said more clearly). The mutual dependence between those quantities is expressed by Eq. 23, the significance of which (in addition to my remark b above) should be stressed more strongly.**

Reply3e: We clarified the explanation for the number of independent innovation statistics (now denoted as "residual statistics" based on Reviewer1, ll.97-100, new count):

*" Because of $j \neq i$ for residuals, each of the $I$ datasets can be combined with all other $I-1$ datasets. As residual statistics also do not change with the order of datasets in the residual (see*

*Sect. 3.1), the number of known statistics of the system is also given by $N_I$ as defined in Eq. (1).*
*"*

A note on the significance of Eq. (23) - beeing the main equation in this Sect.3.2.3 - was added in the description of the general framework Sect. 2 (ll.101-101, new count):

*" It will be shown in Sect. 3.2.2 that residual cross-covariances contain generally the same information as residual covariances; thus the $N_I$ residual statistics can be given in form of residual covariances or cross-covariances. "*

**These are only examples of places that can cause confusion in the mind of a reader who is a newcomer to the approach described in the paper, as elementary as that approach may fundamentally be. I think Sections 3 and 4 could be rewritten in a clearer and more concise way, with more stress on the logic of the approach and on the two fundamental aspects upon which it is based. First, that the observed innovation covariances and crosscovariances are redundant. Second, the basic link between between the innovation and errors covariances and cross-covariances, expressed by Eq. (22) (or any other equivalent equation for that matter).**

Reply3: We hope that the corrections applied w.r.t.comment 3a-e already contribute significantly to the clarification. Firstly, a reference to the equivalence of residual covariances and cross-covariances was made in the formulation of the general framework (Sect. 2) for comment 3e. Secondly, the importance of the link between residual (innovation) and error statistics, which was emphasized at several locations in the manuscript for comment 3a - including the introduction (Sect. 3) and general framework (Sect. 2) and the introduction of the theoretical part (Sect. 3 and 3.3). Thought these changes, the formulation of the general framework in Sect. 2 now introduces explicitly the two fundamental aspects, puts them into context of the framework and provides references to the respective sections in the theory.

In addition to the modifications above, the list of new aspects in the introduction of Sect. 3 was extended w.r.t. the fundamental aspects suggested by the reviewer (ll.142-145, new count):

*" This first part of the mathematical theory includes the following new elements: (i) the separation of cross-statistics into a symmetric error dependency and an error asymmetry (Sect. 3.1), (ii) the general formulation of residual statistics as function of error statistics (Sect. 3.2.1 and 3.2.2), (iii) the demonstration of equivalence between residual covariances and cross-covariances (Sect. 3.2.3), (iv) the general formulation of exact relations between residual- and error statistics (Sect. 3.3). "*

And the list of Sect. 4 was modified accordingly (ll.313-316, new count):

*" In addition to the optimal extension to more than three datasets, this second part of the mathematical theory includes the following new elements: (i) the analysis of differences from residual covariance and cross-covariance estimates (Sect. 4.1.2), (ii) the determination of uncertainties caused by assumed error statistics (Sect. 4.1.3 and 4.2.3), and (iii) the comparison of the approximation from three- ("triangular estimation") and more ("sequential estimation") datasets (Sect. 4.2.4). "*

4. **And, for a final (but I think important) comment, any algebraic solution to system (22) will not be acceptable in then present context. It must also define a proper (symmetric non negative) global error covariance matrix (in particular, the estimated error covariance matrices $C_i$ of the various individual datasets, in addition to being symmetric, must be nonnegative). The authors hardly mention this point. Do the conditions (i-ii) stated in subsection (6.1) lead to a proper global covariance matrix ? Since system (22) expresses necessary conditions between error and innovation variances and covariances, I presume that if the a priori specified variances and cross-covariances are compatible with a globally symmetric non-negative**

matrix that is itself compatible with the $\Gamma_{i,j;k,l}$ 's (Eq. 22), the estimated variances and cross-covariances will also be. I do not ask the authors to necessarily give a full answer to that question, but it should be clearly mentioned and at least briefly discussed. In particular, if the authors do not have a full answer to that question, it should clearly stated as remaining an open question.

Reply4: We agree with the reviewer that this is an important point to consider when it comes to the application to real data. The equations in Sect. 3-4 and the minimal conditions in Sect. 6.1 do not ensure positive definiteness of the error covariance matrix.

There is already a comment on this in the original version of the manuscript in the formulation of uncertainties for 3 datasets (Sect. 4.1.3, ll.391-394, new count):

" *Thus, the estimated error covariance matrices might not be positive definite if the independent assumption between three datasets is not fulfilled. This phenomena was also described and demonstrated by Sjoberg et al. (2021) for scalar problems. However, the generalization to covariances matrices is expected to increase the occurrence of negative values were correlations between two entries of the state are low, thus relative differences and sampling errors become large.* "

As well as a general comment in the formulation of uncertainties for multiple datasets (Sect. 4.1.3, ll.484-486, new count):

" *Note that the absolute uncertainties presented here only account for uncertainties due to the underlying assumptions on error cross-statistics and not due to imperfect residual statistics occurring e.g. from finite sampling. An discussion of those effects for scalar problems can be found in Sjoberg et al. (2021).* "

Specifically, negative values appear in the estimates for the estimation with a large neglected dependency in the basic triangle (manuscript Fig. 4). Here, we added a note on the appearance of negative values (ll.535-537, new count):

" *This particular setup also demonstrates that uncertainties due to neglected dependencies can become larger than the actual true statistics (here e.g. in the error dependency of datasets $(2;4)$ for both estimation methods) which creates negative values in the estimate.* "

In addition, we added a short note in the discussion section when discussing the application to real data (ll.626-628, new count):

" *While the presented method ensures symmetry of error covariances, positive definiteness might not be fulfilled in real applications due to inaccurate assumptions or sampling uncertainties.* "

**It may that the response to some of the questions I raise above is available in the literature, in particular in the literature the authors mention. If so, please give precise references.**

**I would have a number of other comments, bearing on both scientific and editing aspects of the paper, but they are of lesser importance, and I will wait for a possible revised version for mentioning them.**

Reply: We performed several detailed corrections concerning wording, notation and scientific content following the suggestions of the other reviewer. If there are remaining comments in the new version of the manuscript, we are happy to implement them accordingly.

---

## Referee Report (RR1)

This new version of the paper is for me a substantial improvement. I had written in my previous review that I was confused by the way the paper was written and that I had difficulties in understanding what the authors had exactly done. I think I have now understood (expect for Section 5, see comment 2 below, and for possibly minor details). That is due to improvements in the paper (for instance, in agreement with the first comment made by R. Todling in his review, the systematic use of the word *residual* instead of *innovation*). It is also due to the fact that a second reading of a paper, after some time, very often leads to a clearer understanding.

The paper is original and instructive and may be useful for many applications, for instance, as mentioned by the authors in their conclusion, for the combined use of analyses or forecasts produced by different Numerical Weather Prediction centres. On the other hand, I think improvement is still necessary, not in the scientific content of that paper, but in the way it is written and in the conclusions that the authors draw from their results. My main comments and suggestions, in approximate order of decreasing importance, are given below. I could have included some of these in my previous review, but I did not either because I had not understood some aspects I have now understood better, or because I considered these comments and suggestions of secondary importance at that stage. My main two comments (points 1 and 2 respectively) bear on the optimality that the authors claim to have defined and on the presentation of the numerical results (Section 5)

1. The authors write (abstract, ll. 7-8) that their paper provides *a formulation of the minimal and optimal conditions to solve the problem* [*i.e.* the problem of estimating the statistics of the errors affecting collocated data] (see also section 6.2 *Optimal setup*, and statements on *e.g.* ll. 84, 544, 559, 645).

The minimal condition to solve the considered estimation problem is to define hypotheses that make that problem exactly determined. The authors indeed define an approach that satisfies that condition.

But concerning optimality, the above claim is largely exaggerated. I understand the authors mean that the uncertainty in the estimated statistics is in some sense minimum, but give no precise criterion by which criterion the corresponding optimality is defined. The 'optimal setup' is schematically described by Figure 6, and is based on *a priori* (and largely subjective) hypotheses as to the degree of correlation of the errors in the various datasets. That is by no means 'optimal' in any precise sense.

The sentence (ll. 600-601) *Thus, multiple independence trees can be defined …* clearly says that different choices of triangles and reference subsets can be defined, which cannot be all 'optimal'.

In addition, the authors state clearly (l. 627) that positive definiteness of the estimated covariance matrices might not be fulfilled with their approach. Positive definiteness is obviously necessary for 'optimality' of estimates of variances and covariances.

I think the authors should either state precisely by which measure they consider their approach is optimal, or (preferably) remove any claim of optimality.

2. I find Section 5, which presents results of numerical experiments, rather confusing and difficult to understand. The experiments that have been performed are not really

described. What were for each experiment the real error statistics that were used for producing the datasets and then computing the residual statistics ? What where then the error statistics that were *a priori* assumed in order to close the problem of estimation of the global error statistics ? Were those *a priori* assumed statistics consistent with the real ones ? I understand that was the case for the experiment whose results are presented in Fig. 2a, but not for the other ones, although that is not clearly said. If they were consistent, were all the *a posteriori* estimated statistics in agreement with the real ones ? I understand that was the case for Fig. 2a, (… *the two remaining dependencies are estimated accurately*, l. 511), with the consequence that the 'matrices' in the two bottom rows of Fig. 2a must be exactly symmetric, with in particular the matrices in the third row being full of zeroes. That seems visually to be the case, but is not mentioned explicitly, leaving the reader in some doubt.

In the other experiments, the assumed statistics were not consistent with the real ones. That was the case for Fig. 2b, about which the authors write (ll. 525-526) *As shown in Fig. 2b, the error covariance of dataset 4 is underestimated by half the neglected dependency between (2;4)*. How can that (including the quantitative assessment) be seen from Fig. 2b ? And how can the reader check that the errors in the *a posteriori* estimated statistics are themselves consistent with the corresponding estimations presented in Section 4 ? The reference to Eq. (42), (51) and (52) (l. 521) is not of much help.

Considering figures, they must help then reader, who must be able to distinguish, among the statements made by the authors, what can be seen in the figures, and what cannot (in the latter case, the simple mention *not shown* is necessary).

These are only examples. I do not suspect that anything is scientifically wrong or disputable in Section 5 (nor anywhere else in the paper for that matter). But I think the Section must rewritten, with a precise description of each of the experiments that have been performed, with explicit statement of the associated hypotheses, and with a precise description of the obtained results, as well as of the conclusions that must be drawn from these results.
The only difficulty left to the readers must be with a proper understanding of the approach taken by the authors and of what the latter have done. Anything that has to do with deciphering the figures and drawing the conclusions must require no additional effort from the reader.

3. As already mentioned in my previous review, Eq. (23) can be obtained directly without going through the error statistics **C**, **X** or **D**. The authors write in their response that they have rearranged the equation to show the equivalence between innovation and error statistics in a clearer way. My point is that there is simply no need to refer to a 'truth' or to 'errors' in order to obtain Eq. (23). The equivalence between innovation and error statistics has been clearly shown by Eqs (19-22). Introducing quantities in places where there are useless can only confuse the reader (the fact there is a link between innovation and error statistics is irrelevant at this precise point).

4. Ll. 303-304, *The equivalence demonstrates that the exact formulations of error statistics from residual covariances and cross-covariances are consistent to each other* (incidentally, the correct wording would be *consistent **with** each other*). I understand the authors stress the 'equivalence' of those formulations because they will show later that, under the assumption of 'independence', they may lead to different results. But the reader cannot

understand at this stage why it is useful to stress the equivalence. I suggest the authors write rather *The equivalence demonstrates that,* **as they must be***, the exact formulations ….* And the equivalence does not ultimately result from the fact that Eq. (20) is a special case of Eq. (22) (ll. 305-306), but from the basic definitions (4) and (5).

A similar argument holds before that for Eq. (32) and the comment that follows. I suggest to write (l. 289) *… the formulation of error covariances based on residual cross-covariances in Eq. (31) is,* **as it must be***, symmetric and equivalent …*

5. Ll. 553-554, *… there are two requirements for the setup of datasets* It does not seem to me that the need for either one of two stated requirements has been shown. It has only been shown that those two requirements are sufficient for solving the underlying estimation problem (actually, I understand the sentence ll. 569-570 as meaning that other possibilities exist).

6. L. 552, *… without introducing additional degrees of freedom.* That formulation is confusing. From what I understand, the purpose is fundamentally to *eliminate* degrees of freedom by introducing hypotheses that render exactly determined the problem of estimating the error statistics. It seems that you implicitly anticipate on the text (ll. 602-608) where it is suggested to combine different estimates.

7. Ll. 551-552, *The only restriction is that all assumed error statistics must be fully determined …* From what I understand, the wording *all* **additional** *assumed error statistics* would be more appropriate.

8. Concerning positive definiteness of the estimated error statistics, the authors write (ll. 393-394) *However, the generalization to covariances matrices is expected to increase the occurrence of negative values were* (incidentally, the proper spelling is *where*) *correlations between two entries of the state are low, thus relative differences and sampling errors become large.* I am not sure I understand what that means. Is it that the occurrence of non definite positiveness is likely to increase with the number $I$ datasets, or what ?

9. L. 370, *All three estimates become equivalent if the residual cross-covariances are symmetric* Yes, but it might be useful to mention here that these equivalent estimates may not be positive definite.
And l. 388, *Estimated error covariances might even contain negative values …* You must mean *might even not be positive definite …* .

10. Ll. 626-627, *While the presented method ensures symmetry of error covariances,* Not necessarily (Eqs 37 or 38 do not ensure symmetry). Say that symmetry can be enforced if necessary.

11. L. 2 (abstract), *… an ill-posed problem …* Not *ill-posed*, but *underdetermined*

12. Ll. 325-326, "*assumption of independence*". I mention that the word *independence* is not used here in its accepted standard meaning in probability theory. The hypothesis made here is actually an hypothesis of no correlation, which is a weaker property than independence. I suggest the authors briefly mention that fact.
And l. 325, *equals* is to be changed to *implies*

13. Ll. 440-441, … *absolute uncertainties of estimations from residual covariances and cross-covariances differ only in the uncertainties w.r.t. the basic triangle* … Well, these uncertainties depend also in the uncertainty in the hypothesis $\mathbf{D}_{i;\mathrm{ref}(i)} = 0$.

14. L. 315, *the determination of uncertainties* **resulting from possible errors in the**  *assumed error statistics*

15. L. 201, what are *unbiased error statistics* ? Error statistics are here the unknowns, not the data, and whether they are biased or not makes *a priori* no sense.

16. L. 323, $\mathbf{X}_{i;j} = 0 \Leftrightarrow \mathbf{D}_{i;j} = 0$ to be changed to $\mathbf{X}_{i;j} = 0 \Rightarrow \mathbf{D}_{i;j} = 0$
L. 414, similarly, the condition $\mathbf{X}_{i;\mathrm{ref}(j)} = 0$ is actually stronger than the assumption $\mathbf{D}_{i;\mathrm{ref}(i)} = 0$ made on l. 405. Be more consistent as to which hypotheses you make.

17. L. 558, $I \leq 3$, you must mean $I > 3$ ?

18. L. 318, $U_I \geq 0 \rightarrow D_I \geq 0$

19. L. 352, … *four equivalent formulations for each pair of other datasets* … . You have assumed here $I = 3$, so that there is only one pair of 'other datasets' (see also l. 358).

20. L. 34, what are exactly the *corners* ?

21. Ll. 40-41, sentence starting *Up to now,* … awkward. From what I understand, I suggest *Up to now, only scalar error variance estimation has been implemented in data assimilation with the TC method (e.g.* …

22. The word *exemplary* is used mistakenly in several places (the word designates something that is meant to be imitated, while the authors obviously think of illustrative examples)
L. 543, … *including an exemplary visualisation,* → … *including an illustrative visualisation,*
L. 566, *An exemplary setup* … → *An illustrative example of setup* …
Caption of Figure 5, *An illustrative example of visualization* …
See also l. 7 (abstract)

23. Ll. 35-36, … *this particular error estimation problem can only be closed under the assumption of optimally.* I do not understand what this means (and the proper wording should in any case be *assumption of optimality*)

24. Ll. 54-55 (and later) *error cross-variances* … You must mean *error covariances* ?

25. Ll. 65-66, … *affect different formulations of the estimated error statistics?* → … *affect different estimations of the error statistics?*

26. L. 613, I understand the *N-CN method* is what was called previously the *N-cornered hat method* (l. 603), or what ?

27. L. 24, *arises* → *raises*

28. L. 588, *effect* → *affect* (or *impact*)

29. L. 589, *triple* → *triplet* (the same correction is to be made elsewhere, *e.g.* l. 407 ; please check)

30. Ll. 574-575, *… can be interpreted **as being** similar to …*

31. L. 583, *… is comparably well known, …* word *comparably* inappropriate here.  I suggest  *… is known to some degree of accuracy, ……*

32. L. 544, *… algorithmic summary **for** the calculation …*

Although the authors have obviously been very careful with their notations, in particular as concerns indices, I have noticed a few typos

33. Ll. 177 and 184, $\mathbf{D}_{i;j}$ should be replaced with $\mathbf{D}_{i-j;k-l}$ and $\mathbf{Y}_{i-j;k-l}$ respectively

34. Eq. (32), (22) above first = sign should rather be (30)

35. L. 442, $m_{f-1}$ → $m_{F-1}$

And finally, it is the first time I have seen superscripts above equality signs to give reference to previous equations. I think that can be useful, and it is undoubtedly in the present case.

---

## Referee Report (RR2)

The authors have significantly modified their paper, and have taken my comments and suggestions into account. They have generalized their 'triangular' approach to estimation in the form of an interesting 'polygonal' approach (none of my suggestions). I think the paper is now acceptable for publication. I nevertheless suggest a number of modifications of mostly editing character which should not require substantial work.

1. From a purely scientific point of view, I have only one request for correction which, without being critical for the paper, must be made.

The authors write, concerning positive definiteness (ll. 406-407) ... *the generalization to covariances matrices is expected to increase the occurrence of negative values in off-diagonal elements.* Positive definiteness is not a question of the signs of off-diagonal elements. It is that a covariance matrix, in addition to being symmetric (and to have as such real eigenvalues) must have only positive eigenvalues (or nonnegative ones, depending on the precise definition one takes). For instance, the 2-2 matrix

$$\begin{array}{|c|c|} \hline 1 & a \\ \hline a & 1 \\ \hline \end{array}$$

is definite positive if $a^2 < 1$, and is not if $a^2 > 1$, independently of the sign of $a$.

Along the same line of ideas, I make the following remarks

1*a*. L. 727, I think the need for positive definiteness will always be there, independently of the purpose or use (using a 'covariance matrix' that is not positive definite can lead to absurd results, for instance in data assimilation)

1*b*. And I suggest to change the words *positive definiteness might not be fulfilled* (ll. 785-786) to *there remains the risk that positive definiteness might not be fulfilled*

2. L. 685, ... *not every possible combination of error statistics to be estimated provides a solution.* This statement, which is fundamental for the paper, is mentioned explicitly only in the conclusion. It should be mentioned in the introduction (for instance between ll. 47 and 48 of the present version, or after l. 67), with appropriate references (possibly Vogelzang and Stoffelen, 2001, or Gruber *et al.*, 2016).
In addition, from I understand, a better formulation would be ... *not every possible **a priori choice** of error statistics to be estimated provides a solution.*

On the number of occasions, the wording is improper, imprecise or awkward

3. Ll. 71-72, … *the exact formulations of error statistics in Sect. 3.3 remain underdetermined* … to be changed to … *the **complete estimation** of error statistics in Sect. 3.3 remains underdetermined* … (it is the estimation as a whole that is underdetermined in a real situation, not the exact estimation)

4. Ll. 323-324, sentence starting *It includes* … seems awkward to me. From what I understand, I suggest *In order to achieve a complete estimation of error statistics, the Section includes the estimation, either direct or sequential, of additional error covariances, as well as of some error cross-statistics* (although it seems to me that the words *additional* and *some* are not here necessary, since it is the whole of error statistics that are in the end to be estimated. But I may be mistaken)

5. L. 326, *the analysis of differences from residual covariance- and cross-covariance estimates*, … The formulation may be confusing. I think it would be preferable to restrict the word *estimate* to estimates of the error statistics, not of the residual statistics, which are directly computed from the data and which (contrary to the estimation of the error statistics) do not require *a priori* hypotheses. I suggest to write simply *the analysis of differences from residual covariances and cross-covariances*.
A similar change may have to be made elsewhere. Please check.

6. L. 337, … *the assumption of zero **error** cross-covariance* …

7. Ll. 384-385, … *non**e** of the estimates ensures positive definiteness of the **estimated** error covariances.*

8. Ll. 415-416, I suggest *As described in Sect. 2, for I > 3 datasets, $A_1 > 0$ gives the number of error cross-statistics which can potentially be a priori assumed in addition to all error covariances*

9. L. 437, … *pairwise-independent datasets* …, do you mean two successive datasets in the series are independent, or more than that ?

10. L. 437 again, … *where the referring error covariances* ... What do you mean here by *referring* ?

11. L. 464, … ***polygonal** error covariance* …

12. L. 523-524, sentence starting *The sequential estimation* … makes no real sense to me. In any case, there is a useless repetition of the words *error covariance*, and the proper wording is *similar … to* (not *similar … as*)

13. L. 546, … *the true error statistics* … Ambiguous. The *a priori* assumed error statistics, or the empirical statistics obtained from the 20,000-sample ? The difference may be

small in the present case, but I presume you mean the latter, which would be the only ones available in a real situation. And they would not be *error statistics*, but *residual statistics*.

14. Ll. 565-566, … *the independent assumption in the basic triangle(1;2;3) is an accurate approximation of the true error dependencies*. Well, have not the true error dependencies assumed to be zero by construction ?

15. L. 579, … *true error dependencies are occurring …* → … *true error dependencies are used …*

16. L. 591, … *This neglected positive error dependency …* In what is that neglected error dependency positive ?

17. L. 594-595, … *these three matrices are underestimated w.r.t. the true statistics by the half of the neglected error dependency matrix …* That these matrices are underestimated by half of the neglected error dependency matrix is not visible from the figure. Say it clearly, and say how you know it. Is it consistent with error estimates that have been given in Section 4 ? If yes, say it clearly and make reference to the relevant equations. If not, explain.
A similar comment applies to the text ll. 615-617.

18. L. 597-598, Sentence starting *This experiment demonstrates …* Move the word *accurately* to … *to estimate accurately*.

19. L. 598, … *accurate reproduction* → *accurate estimation*

20. L. 606 … *a true dependency …* → … *a non-zero dependency …*

21. Similarly, l. 607, … *three true error dependencies …* → … *three non-zero error dependencies …*

22. L. 830, … *the need for estimating **possible** asymmetric error cross-covariances …* (cross-covariance matrices, if they are square in the first place, will in general by asymmetric, but there is no need for them to be so).

The English, although perfectly understandable in most places, is occasionally incorrect

23. In many places (starting on l. 272) the word *uneven* is used to qualify an integer that is not even. The proper word is *odd*

24. L. 30, *This method **has been** widely used …*

25. L. 48, … *the estimation […] **has been** well-established for decades, …*

26. L. 54, … *were the first **to propose** the additional estimation …*

27. L. 338, *thereafter* → *hereafter*

28. L. 498, *partly* → *partially*

29. L. 534, … *the capabilities **for** estimat**ing** …*

30. L. 651, *If the other two error covariances 1 and 2 **had** also been estimated*

Additional corrections to be made

31. L. 2, … *is **an** underdetermined problem* …

32. L. 70, word *respectively* seems here useless

33. L. 96, l. 189, … *as function**s** of* ...

34. L. 461, … *each subsequent dataset**s** j > i* … (singular)

35. L. 477, … *and  $m_{G-1}$ **the** reference of* …

36. L. 546, suppress the word *be*

37. L. 556, … *to estimate the error covariance* …

38. L. 626, … *the uncertainties **in the** two estimated error dependencies* …

39. L. 526, … *of **the** other two error dependencies* …

40. L. 724, *dat**a**sets*

41. Ll. 767-768, … ***a** flexible setup* …

42. L. 807, Title of Appendix ***Algorithms*** (plural)

43. L. 201, … *or unbiased datasets* …. It has already been said (l. 148) that unbiasedness would be assumed in the paper. Unless the authors want to stress particularly that point (and, if so, they must do it explicitly), this kind of repetition is useless, and may even be confusing.

---

## Editor Decision (ED1)

Three referees have sent their reports on the latest version of the paper. Referees 1 and 2 are the same ones as those of previous version. In particular, referee 1 is still Ricardo Todling. Referee 3, whom I had solicited in order to get an additional opinion, is an expert of the estimation methods that are studied in the paper.

R. Todling writes that he is satisfied with the revision and recommends acceptance of the paper as it stands.

Referee 3 considers that the paper is of significant scientific interest and is worth publishing. He only asks for minor revisions, which all bear on editing aspects.

Referee 2 is more critical. He had written in his previous review that he had difficulties in understanding exactly what the authors had done. He writes that he finds now the paper much easier to understand. He has no criticism on the basic science, but he nevertheless writes that two aspects need significant modifications. He considers that the 'optimality' claimed by the authors as to their approach is unjustified. And he also considers that Section 5, which presents results of numerical experiments, is difficult to understand and requires substantial rewriting. He also adds a number of suggestions on editing aspects.

Please revise your paper according to the recommendations of referees 2 and 3. Please respond to all of these recommendations. In case you disagree with, or decide not to follow, one of them, please state precisely your reasons for that.

Not being a native English speaker, I am not fully qualified for commenting on the language, but it is clear that it needs improvements. Referee 3 makes a number of suggestions in this respect. The paper, if it is accepted, will go through copy-editing by the technical editor. But it is preferable to correct it as much as possible at the present stage. Please check the language systematically (or have it checked by a qualified person).

I am looking forward to receiving a new version of your paper. I may submit it to one or two of the referees.

---

## Author Response (AR2)

**Response to Report 2 (reviewer 3):**

General Reply: We thank the reviewer for the insightful remarks and hope that we could reply and adopt the manuscript in a sufficient way.

**This study addresses the challenge of accurately determining error statistics for data assimilation and other uses. The work presented here builds on a growing body of geo-physical applications of estimating error statistics using collocated observations. A detailed mathematical theory is presented to demonstrate how three or more data sets, with limited assumptions and no knowledge of truth, may be used to simultaneously and/or sequentially calculate error covariances and error cross-covariances for all data sets. The authors present illustrative examples of its application and a useful conceptual model for how it could be applied.**

**I find the work to be quite complex, in a way that I appreciate. That is, the level of detail is sufficient without being overbearing. And I believe this adds an intriguing method (or at least considerable detail to existing methods) to the literature on error estimation from multiple data sets. It's compelling to think on how one might apply the method to existing data, especially the determination of the three data sets that comprise the basic triangle and the subsequent order of the calculation. However, I think the authors have addressed this with sufficient detail for the present work.**

**I recommend a number of minor and technical edits to the text prior to publication.**

**Minor comments:**

- **The use of scalar D_i for number of estimated error statistics and the matrix D_i;j for error dependency matrices and residual dependency matrices is unfortunate. I think it's clear enough, and perhaps this is common notation in the literature of which I'm not aware. But, the authors may consider changing the scalar D_i notation for clarity.**

  Reply: The scalar quantity $D_I$ was never defined in literature, so there is no restriction to the use of a specific character. Actually, the choice of using of scalar $D_I$ similar to the error dependency matrix $D_{i;j}$ was made on purpose because the scalar $D_I$ it is the number of estimated error dependencies $D_{i;j}$ (of cross-covariances in general). But we agree that the double-use of $D$ might be confusing, so we replaced the scalar $D_I$ by $A_I$ (refering to the number of **A**dditionally estimated error cross-statistics) though the manuscript, incl l.108, Eq.(3), Fig. 1, ...

- **Section 4.2 might be better titled as "Approximation for more than three datasets" as Section 4.1, which uses three datasets, already has "multiple" datasets.**

  Reply: Replaced as suggested (l.409).

- **Figs. 2, 3, and 4: the bottom axes of some of the subplots are cut off in this version. Please check that this is not the case in the final figures.**

  Reply: Thank you for the careful evaluation. However, there is no bottom axis at the third row of Fig. 2a,2b,3a,3b,4a,4b because the lower axis refers to the lower-right part of each subplot which in empty in the third row. During the 1st review, one reviewer got confused expecting the lower-right part of the third row to be zero-values. So we decided to remove the bottom axis in these subplots for clarification (among adding thin gray lines in the lower-right part). We now added a note on the

empty lower-right subplots in the third row in caption of Fig.2 and in the description of the plots (ll.576-577):

*" Note that the lower-right part of each subplot in the 3rd row does not contain any data. "*

- **The "innocov" and "innocross" terms in algorithms A1 and A2 should be defined. They refer to the residual covariances and residual cross-covariances, correct? It's not clear with this naming, so please either change the terms or provide a (brief) definition. Are these perhaps terms from a previous version of the manuscript?**

  Reply: Thank you for this hint. Based on the 1st review, we renamed the innovation covariance and cross-covariance to residual covariances and cross-covariance in the manuscript, but forgot to update the algorithms. We replaced the terms accordingly in Alg. A1 and A2 and converted both algorithms to a template-conform format without changing its content.

- **Many instances of "were" being used in place of "where".**

  Reply: Thank you. It was corrected thought the whole document.

- **Line 15: remove comma between "both" and "background"**

  Reply: Done.

- **Line 24: "raises" instead of "arises"**

  Reply: Corrected.

- **Line 27: remove "being"**

  Reply: Done.

- **Line 30: remove "and" ("but has only")**

  Reply: Done.

- **Line 36: "optimality"?**

  Reply: Corrected.

- **Line 62: "where each approach"?**

  Reply: Corrected.

- **Line 69: spelling of "unknowns"**

  Reply: Corrected.

- **Line 114: suggest "as is the case"**

  Reply: Replaced with *"as in the case"* (l.114).

- **Fig. 1 caption: spelling of "and" (instead of "ans")**

  Reply: Corrected.

- **Line 257: suggest "other datasets and a"**

  Reply: Corrected.

- **Line 282: spelling of "datasets"**

  Reply: Corrected, also at all other locations in the manuscript. Thank you.

- **Line 318: should this be "D_i" instead of "U_i"?**

  Reply: Yes. Corrected, thank you.

- **Line 405, 448: perhaps "partial independence"?**

  Reply: Corrected.

- **Line 430: spelling of "cross-covariances"**

  Reply: Corrected.

- **Line 489: suggest "20,000" (comma instead of period)**

  Reply: Corrected.

- **Line 520: suggest "basic triangle are illustrated"**

  Reply: Corrected.

- **Fig. 3 caption: should be "but with a neglected"**

  Reply: Corrected.

- **Line 563: spelling of "solvability"**

  Reply: Corrected.

- **Fig. 6 caption: would it be better to state the distance as representing "the accuracy of assumed independence "?**

  Reply: Fig.5 and 6 are defined in the general case where error dependencies (or error cross-covariances) are assumed in some way (compare caption of Fig.5). In this context, the distance of dataset in Fig.6 indicates the accuracy of the related dependency; where the independence assumption is one choice of setting the assumed error dependencies to zero. Therefor, we kept the current formulation.

- **Line 588: should be "affect" rather than "effect"**

  Reply: Corrected.

- **Line 639: "major challenge"**

  Reply: Corrected.

Response to Report 3 (reviewer 2):

General Reply: We thank the reviewer for the very thoughtful and detailed evaluation and valuable remarks. We hope that we could reply and adopt the manuscript in a sufficient way.

Before providing detailed replies to each comment below, the main modifications of the manuscript are summarized here (compare replies to individual comments for more details):

- reformulation of Sec.5 for clarification of experiments (compare report3, comment2)

- reformulation of Sec.6.2, removing the claim for optimality of the solution (compare report3, comment1)

- reformulation and extension of Sec.6.1, placing the claim for requirement of the two conditions by a conceptual description of their sufficiency to solve the problem (compare report3, comment5)

- extension of the triangular estimate of error covariances to a polynomial estimate mainly in Sec.3.3.1, 4.2.1, 4.2.4, 4.2.5 (compare report3, comment5)

From our point of view these changes improved the manuscript significantly without changing its main statements.

**This new version of the paper is for me a substantial improvement. I had written in my previous review that I was confused by the way the paper was written and that I had difficulties in understanding what the authors had exactly done. I think I have now understood (expect for Section 5, see comment 2 below, and for possibly minor details). That is due to improvements in the paper (for instance, in agreement with the first comment made by R. Todling in his review, the systematic use of the word *residual* instead of *innovation*). It is also due to the fact that a second reading of a paper, after some time, very often leads to a clearer understanding.**

**The paper is original and instructive and may be useful for many applications, for instance, as mentioned by the authors in their conclusion, for the combined use of analyses or forecasts produced by different Numerical Weather Prediction centres. On the other hand, I think improvement is still necessary, not in the scientific content of that paper, but in the way it is written and in the conclusions that the authors draw from their results. My main comments and suggestions, in approximate order of decreasing importance, are given below. I could have included some of these in my previous review, but I did not either because I had not understood some aspects I have now understood better, or because I considered these comments and suggestions of secondary importance at that stage. My main two comments (points 1 and 2 respectively) bear on the optimality that the authors claim to have defined and on the presentation of the numerical results (Section 5).**

1. **The authors write (abstract, ll. 7-8) that their paper provides a *formulation of the minimal and optimal conditions to solve the problem* [i.e. the problem of estimating the statistics of the errors affecting collocated data] (see also section 6.2 *Optimal setup*, and statements on e.g. ll. 84, 544, 559, 645).**

   **The minimal condition to solve the considered estimation problem is to define hypotheses that make that problem exactly determined. The authors indeed define an approach that satisfies that condition.**

But concerning optimality, the above claim is largely exaggerated. I understand the authors mean that the uncertainty in the estimated statistics is in some sense minimum, but give no precise criterion by which criterion the corresponding optimality is defined. The 'optimal setup' is schematically described by Figure 6, and is based on a priori (and largely subjective) hypotheses as to the degree of correlation of the errors in the various datasets. That is by no means 'optimal' in any precise sense.

The sentence (ll. 600-601) *Thus, multiple independence trees can be defined ...* clearly says that different choices of triangles and reference subsets can be defined, which cannot be all 'optimal'.

In addition, the authors state clearly (l. 627) that positive definiteness of the estimated covariance matrices might not be fulfilled with their approach. Positive definiteness is obviously necessary for 'optimality' of estimates of variances and covariances.

I think the authors should either state precisely by which measure they consider their approach is optimal, or (preferably) remove any claim of optimality.

Reply: We agree with the reviewers concern about the use of the word "optimal" in the context of the setup of the problem. As noticed by the reviewer, this mainly refers to Sec.6.2 "optimal setup" and some other locations in the manuscript which refer to this section. Actually, this section aims to provide some guidelines for the selection of a setup among the possible setups which were determined in Sec.6.1 "minimal conditions". Given the large number of possible setups - which actually also include the definition of multiple independence trees in the reviewers reference to ll. 600 (old count) - we think that it is important to stress that there might be significant differences in real applications were assumptions remain inaccurate and that it is therefore crucial to select an appropriate setup. The guidelines in this section only refer to the aim of "minimizing" the uncertainties introduced by assumptions. The reviewer is right that positive definiteness is one another important criterion which would be required for truly optimal covariance estimates.

We extended the introduction of Sec.6.2 for clarification (ll.724-729):
" *The general rules given in Sect. 6.1 allow for multiple different setups of datsets which all solve the error estimation problem. However in real applications, theremight be significant differences in estimated error statistics fromdifferent setups as observed e.g. by Vogelzang and Stoffelen (2021) in the scalar case. The optimal selection is specific for each application and may depend on several requirements related to the actual purpose or use (e.g. available knowledge, need for positive definiteness, accuracy of each estimate). This section provides some general guidelines on the selection of an appropriate setup among the various possible solutions w.r.t. the uncertainties introduced by statistical assumptions.* "

We also went though the manuscript and reformulated all locations where the word "optimal" appeared in the context of the setup of the problem:

- abstract (l.7): *"guidelines for setting up and solving the problem"*
- introduction (ll.65-66): *"And what are the general conditions to set up and solve the problem?"*
- and (ll.83) *"and provides guidelines for the setup of those"*
- title of Sec.6 (l.667): *"Conceptual summary and guidelines"*
- introduction of Sec.6 (l;.670-671): *"... Sect. 6.2 formulates guidelines for the selection of an appropriate setup of datasets under imperfect assumptions."*
- Sec.6.1 (l.557-561, old count) removed
- title of Sec.6.2 (l.723): *"Selection of setup"*
- Sec.6.2 (ll.735-736): *"In order to achieve sufficiently accurate estimates, ... "*

- and (l.743): *". . . , the setup visualized in Fig. 5 is not an appropriate selection."*
- conclusions (l.645): *"optimal"* removed.

2. **I find Section 5, which presents results of numerical experiments, rather confusing and difficult to understand. The experiments that have been performed are not really described. What were for each experiment the real error statistics that were used for producing the datasets and then computing the residual statistics ? What where then the error statistics that were a priori assumed in order to close the problem of estimation of the global error statistics ? Were those a priori assumed statistics consistent with the real ones ? I understand that was the case for the experiment whose results are presented in Fig. 2a, but not for the other ones, although that is not clearly said. If they were consistent, were all the a posteriori estimated statistics in agreement with the real ones ? I understand that was the case for Fig. 2a, (. . . *the two remaining dependencies are estimated accurately*, l. 511), with the consequence that the 'matrices' in the two bottom rows of Fig. 2a must be exactly symmetric, with in particular the matrices in the third row being full of zeroes. That seems visually to be the case, but is not mentioned explicitly, leaving the reader in some doubt.**

    **In the other experiments, the assumed statistics were not consistent with the real ones. That was the case for Fig. 2b, about which the authors write (ll. 525-526) *As shown in Fig. 2b, the error covariance of dataset 4 is underestimated by half the neglected dependency between (2;4)*. How can that (including the quantitative assessment) be seen from Fig. 2b ? And how can the reader check that the errors in the a posteriori estimated statistics are themselves consistent with the corresponding estimations presented in Section 4 ? The reference to Eq. (42), (51) and (52) (l. 521) is not of much help.**

    **Considering figures, they must help then reader, who must be able to distinguish, among the statements made by the authors, what can be seen in the figures, and what cannot (in the latter case, the simple mention *not shown* is necessary).**

    **These are only examples. I do not suspect that anything is scientifically wrong or disputable in Section 5 (nor anywhere else in the paper for that matter). But I think the Section must rewritten, with a precise description of each of the experiments that have been performed, with explicit statement of the associated hypotheses, and with a precise description of the obtained results, as well as of the conclusions that must be drawn from these results.**

    **The only difficulty left to the readers must be with a proper understanding of the approach taken by the authors and of what the latter have done. Anything that has to do with deciphering the figures and drawing the conclusions must require no additional effort from the reader.**

    Reply: Thank you for this detailed comment, we understand that the Section was not clear enough. We reformulated Sec. 5, focusing especially on a clear introduction and description of the experiments, including the formulation of the setup, the related assumptions and the estimation procedure. The description of the results in Subsec.5.1 to 5.3 was also reformulated to assist the reader in the interpretation. We tried to make clear where the descried results can actually be seen in the figures (the references to specific panels are given in brackets) and also give some references to the referring equations in the theoretical sections Sec.3 and 4. This also improves the connection of this section to the rest of the manuscript. We also extended the formulation of conclusions of the experiment. We believe that this improves the readability of the Section significantly and hope that it answers all of the reviewers questions sufficiently.

Because this reformulation increased the length of Sec.5, we decided to have a separate subsection for each of the three experiments, i.e. divide Sec.5.2 into two subsections.

The modification affects the complete Sec.5, which we did not copy here. Instead, the general changes are listed below:

- reformulation introduction of Sec.5 for clarification (ll.534-567)
- addition of paragraph in each subsection which introduces the experiments (ll.579-584, ll.605-610, ll.639-642)
- reformulation results of each subsection (ll.585-596, ll.611-634, ll.642-661)
- division of Sec.5.2 into two subsections; rename Sec.5.2 for experiment 2 *"Small uncertainties in basic triangle"* (l.604) and define new Sub.5.3 for experiment 3 *"Large uncertainties in basic triangle"* (l.633)
- addition of a concluding paragraph in each subsection (ll.597-603, ll.635-637, ll.661-666)
- addition *"Experiment XY:"* in captions of Fig.2,3,4
- extension of figure captions (i.e. caption Fig.2): *" For each subplot, gray asterisks in the upper-left part of the first row indicate that these error dependencies are assumed to be zero in the estimation. Note that the lower-right part of each subplot in the third row does not contain any data. "*

3. **As already mentioned in my previous review, Eq. (23) can be obtained directly without going through the error statistics C, X or D. The authors write in their response that they have rearranged the equation to show the equivalence between innovation and error statistics in a clearer way. My point is that there is simply no need to refer to a 'truth' or to 'errors' in order to obtain Eq. (23). The equivalence between innovation and error statistics has been clearly shown by Eqs (19-22). Introducing quantities in places where there are useless can only confuse the reader (the fact there is a link between innovation and error statistics is irrelevant at this precise point).**

Reply: Thank you for the clarification. We now understand the reviewers point and show the equivalence in Eq. (23) directly from the definitions - which is actually more consistent to the rest of Sec.3.2 - without using the true state or error statistics (ll.222-228 and Eq.(23) ):

*" For $k = i$, the residual dependency between $i-j$ and $i-l$ can be expressed as combination of three residual covariances: "*

$$\Gamma_{i;l} + \Gamma_{j;i} - \Gamma_{j;l} \overset{(6)}{=} \overline{\left[x_i(p) - x_l(p)\right]\left[x_i(q) - x_l(q)\right]} + \overline{\left[x_j(p) - x_i(p)\right]\left[x_j(q) - x_i(q)\right]} - \overline{\left[x_j(p) - x_l(p)\right]\left[x_j(q)\right.}$$

$$= \overline{\left[x_i(p)\right]\left[x_i(q)\right]} - \overline{\left[x_i(p)\right]\left[x_l(q)\right]} - \overline{\left[x_l(p)\right]\left[x_i(q)\right]} + \overline{\left[x_l(p)\right]\left[x_l(q)\right]}$$

$$+ \overline{\left[x_j(p)\right]\left[x_j(q)\right]} - \overline{\left[x_j(p)\right]\left[x_i(q)\right]} - \overline{\left[x_i(p)\right]\left[x_j(q)\right]} + \overline{\left[x_i(p)\right]\left[x_i(q)\right]}$$

$$- \overline{\left[x_j(p)\right]\left[x_j(q)\right]} + \overline{\left[x_j(p)\right]\left[x_l(q)\right]} + \overline{\left[x_l(p)\right]\left[x_j(q)\right]} - \overline{\left[x_l(p)\right]\left[x_l(q)\right]}$$

$$= \overline{\left[x_i(p) - x_j(p)\right]\left[x_i(q) - x_l(q)\right]} + \overline{\left[x_i(p) - x_l(p)\right]\left[x_i(q) - x_j(q)\right]}$$

$$\overset{(4)}{=} \Gamma_{i-j;i-l} + \Gamma_{i-l;i-j}$$

4. **Ll. 303-304,** *The equivalence demonstrates that the exact formulations of error statistics from residual covariances and cross-covariances are consistent to each other* **(incidentally, the correct wording would be** *consistent with each other***). I understand the authors stress the 'equivalence' of those formulations because they will show later that,**

under the assumption of 'independence', they may lead to different results. But the reader cannot understand at this stage why it is useful to stress the equivalence. I suggest the authors write rather *The equivalence demonstrates that, as they must be, the exact formulations* .... And the equivalence does not ultimately result from the fact that Eq. (20) is a special case of Eq. (22) (ll. 305-306), but from the basic definitions (4) and (5). A similar argument holds before that for Eq. (32) and the comment that follows. I suggest to write (l. 289) ... *the formulation of error covariances based on residual crosscovariances in Eq. (31) is, as it must be, symmetric and equivalent* ...

Reply: Corrected as suggested, with updated equation numbers (31)→(34) (ll.301-302): *" the formulation of error covariances based on residual cross-covariances in Eq. (34) is, as it must be, symmetric and equivalent to the formulation based on residual covariances from Eq. (25). "*
and (ll.315-318): *" The equivalence demonstrates that, as they must be, the exact formulations of error statistics from residual covariances and cross-covariances are consistent with each other. This consistency applies to the exact formulations of all symmetric error statistics (error covariances and dependencies) and results from the consistent definitions of residual covariances and cross-covariances in Eq. (4) and (6). "*

5. **Ll. 553-554, ...** *there are two requirements for the setup of datasets* **It does not seem to me that the need for either one of two stated requirements has been shown. It has only been shown that those two requirements are sufficient for solving the underlying estimation problem (actually, I understand the sentence ll. 569-570 as meaning that other possibilities exist).**

Reply: We understand that this point is not sufficiently clear in the manuscript. The two requirements are logical conclusions from the theoretical sections before (Sec. 3+4), but they do not provide a sufficient mathematical foundation that shows that there are not other possible solutions of the problem.

Indeed, it would be very interesting to approach the closure problem (i.e. solvability, optimality, ...) from a purely mathematically point of view. However, we believe that a rigorous mathematical proof exceeds the scope of this journal and the purpose (and length) of this paper. We hope to address this problem from a purely mathematical point-of-view in an upcoming study.

A) Here, we decided to give a logical argumentation why these two "requirements" are sufficient solutions of the problem without claiming the necessary need of those. The specific modifications to the manuscript are as follows:

- We replaced the word "requirement" by "rules" and the claim for the need for the two "requirements" by a statement that these "rules" are sufficient for solving the problem (l.669): *"Section 6.1 summarises the general assumptions and provides rules for the minimal conditions ... "*

- Sec.6.1 was reformulated to guide the reader though the argumentation which leads to the two "requirements" (now "rules"). Note that these changes already include the modifications referring to part B of this reply.

  - (ll.674-676): *" This section provides a conceptual discussion of different conditions which need to be fulfilled in order to be able to solve the error estimation problem. The discussion is based on the previous sections, but formulated in a qualitative way without providing mathematical details. "*

  - (ll.684-686): *" The number of error statistics that can be estimated for a given number of datasets ($N_I$) was introduced in Sect. 2. However not every possible combination of error statistics to be estimated provides a solution. The following discussion only considers setups where all error covariances and as many error cross-statistics as possible are estimated. "*

  - (ll.684-705), logical argumentation (see manuscript)

  - (ll.706-708): *" Based on this, two general rules for the setup of datasets can be formulated which ensure the solvability of the problem in the case where all error covariances and as many error cross-statisitcs as possible (cross-covariances or dependencies) are estimated: ... "*

  - remove statements on requirements which were below l.713 (new count)

- other locations in the manuscript which refer to the claim for minimal requirements were also modified accordingly:
    - Sec.1 Introduction (ll.82-83): *" It includes the formulation and illustration of rules to solve the problem for an arbitrary number of datasets and provides guidelines for the setup of those. "*
    - Sec.2 General framework (l.105): *" Sect. 6.1 provides guidelines which ensure the solvability of the problem for a minimal number of assumptions. "*

For clarification, the "illustrative example" (reformulated in accordance to comment below) is an example setup to demonstrate which these two requirements could be fulfilled. It is one option of the large amount of different possible setups which fulfill the requirements and is not meant to indicate that other possibilities exist which would not fulfill the requirements. We added a note to clarify this (ll.718-719):
*" Note that this is one of many possible setups which are determined by the two rules above. "*

B) Related to these changes, we noticed that the current formulation misses one important generalization, which actually defines a larger group of possible solutions of the estimation problem. Up to now, the "direct estimation" of error covariances, with no other error covariances available, was only created by a "triangular estimation" from the combination of 3 residual covariances ( eg. Eq. (27) ). However, this can be generalized to a "polynomial estimation" by combining a closed series of $F$ residual covariances, for any $F$ uneven and larger or equal 3 (and smaller or equal the number of datasets $I$ – see the new parts in the manuscript for a detailed description). We believe that this generalization is an important aspect which needs to be considered in the manuscript and adopted the manuscript accordingly.

This generalization does not change the main content of the study but required a number of minor modifications thought the manuscript, most of them related to the replacement of words, e.g. replacing "triangular estimate" and "basic triangle" by "polynomial estimate" and "basic polynom", respectively. The most significant modifications are the addition of the generalized equations for:

- the generalized exact formulation of error covariances from residual covariances in Sec.3.3.1 (ll.269-277):
*" Equation (26) can be generalized by replacing the closed series of the 3 dataset-pairs $(i; j), (j; k), (k; i)$ with a closed series of $F$ dataset-pairs $(i_1; i_2), (i_2; i_3), \cdots, (i_{F-1}; i_F), (i_F; i_1)$, for any $3 \leq F \leq I$ (and $I$ the number of datasets):*

$$\mathbf{C}_{\tilde{i_1}} \overset{(20)}{=} \sum_{f=1}^{F-1} (-1)^{f-1} \left[ \mathbf{\Gamma}_{i_f; i_{f+1}} + \mathbf{D}_{\tilde{i_f}; i_{\tilde{f+1}}} \right] + (-1)^{F-1} \left[ \mathbf{\Gamma}_{i_F; i_1} + \mathbf{D}_{\tilde{i_F}; \tilde{i_1}} \right] + (-1)^F \mathbf{C}_{\tilde{i_1}}$$

*Because of changing signs, Eq. (28) can only be solved for the error covariance $\mathbf{C}_{\tilde{i_1}}$ if $F$ is uneven. If $F$ is even, $\mathbf{C}_{\tilde{i_1}}$ cancels out and cannot be eliminated, and Eq. (28) could be solved for one error dependency instead. If $F$ is uneven, the generalized formulation for $\mathbf{C}_{\tilde{i_1}}$ becomes:*

$$\mathbf{C}_{\tilde{i_1}} \overset{(28)}{=} \frac{1}{2} \left[ \underbrace{\left( \sum_{f=1}^{F-1} (-1)^{f-1} \mathbf{\Gamma}_{i_f; i_{f+1}} \right) + \mathbf{\Gamma}_{i_F; i_1}}_{\text{"independent contribution"}} + \underbrace{\left( \sum_{f=1}^{F-1} (-1)^{f-1} \mathbf{D}_{\tilde{i_f}; i_{\tilde{f+1}}} \right) + \mathbf{D}_{\tilde{i_F}; \tilde{i_1}}}_{\text{"dependent contribution"}} \right] \quad , \forall \ F \ uneven \ \wedge \ 3 \leq F$$

*where Eq. (27) results for $F = 3$ with indices $i_1 = i$, $i_2 = j$ and $i_3 = k$. Note that in any case, the number of assumed and estimated error statistics remains consistent with the general framework in Sect. 2. "*

- the generalized approximative form for direct estimation of error covariances from residual covariances (new Sec.4.2.1, ll.424-432):
*" For more than three datasets $I > 3$, the estimation from three residual covariances in Eq. (39) can be generalized to estimations of error covariances from a closed series of $F$ residual covariances (compare Sect. 3.3.1). For any uneven $F$ with $3 \leq F \leq I$, each error covariance can be estimated under the assumption of vanishing error dependencies along the closed series of datasets $\mathbf{D}_{\tilde{i_f}; i_{\tilde{f+1}}} \ \forall \ f \in [1, F-1]$*

and $\mathbf{D}_{i_F;\tilde{1}}$:

$$\mathbf{C}_{\tilde{i_1}} \underset{\{inF\}}{\overset{(28)}{\approx}} \frac{1}{2}\left[\left(\sum_{f=1}^{F-1}(-1)^{f-1}\mathbf{\Gamma}_{i_f;i_{f+1}}\right)+\mathbf{\Gamma}_{i_F;i_1}\right] \qquad , \forall \; F \; uneven \; \wedge \; 3 \leq F \leq I$$

where $"\underset{\{inF\}}{\approx}"$ indicates the assumption of neglectable error dependencies along the series of datasets.
As shown in Sect. 2, the problem cannot be closed for less than 3 datasets, even under the independent assumption. For $F = 3$ datasets, Eq. (39) is a special case of Eq. (48) with indices $i_1 = i$, $i_2 = j$ and $i_3 = k$. "

- the uncertainty of the generalized direct error covariance estimate in Sec.4.2.4 (ll.464-468):
  " As generalization of Eq. (45), the absolute uncertainty $\Delta\mathbf{C}_{\tilde{i_1}}$ of a polynomial error covariance estimate introduced by the assumption of pairwise-independence along the closed series of $F$ datasets, with $F$ uneven and $3 \leq F \leq I$, is given by:

$$\Delta\mathbf{C}_{\tilde{i_1}}\Big|_{(47)} := \mathbf{C}_{\tilde{i_1}}\Big|_{true} - \mathbf{C}_{\tilde{i_1}}\Big|_{(47)} \overset{(28),(47)}{=} \frac{1}{2}\left[\left(\sum_{f=1}^{F-1}(-1)^{f-1}\Delta\mathbf{D}_{\tilde{i_f};\tilde{i_{f+1}}}\right)+\Delta\mathbf{D}_{i_F;\tilde{i_1}}\right] \qquad , \forall \; F \; uneven \; \wedge \; 3 \leq$$

  Due to the changing sign of error dependencies along the series of datasets, the absolute uncertainty of the error covariance estimates does not necessary increase with the size of the polygon $F$. "

- the equivalence of the generalized direct error covariance estimate in Sec.4.2.5 (ll.505-512):
  " This can also be generalized for the estimation of any error covariance $\mathbf{C}_{\tilde{i_2}}\big|_{\vdash}$ given its reference $\mathbf{C}_{\tilde{i_1}}\big|_{\bigcirc}$ estimated with the polygonal formulation for a closed series of $F$ pairwise-independent datasets for any uneven $F$ with $3 \leq F \leq I$:

$$\mathbf{C}_{\tilde{i_2}}\big|_{\vdash} \underset{\{inI\}}{\overset{(48)}{\approx}} \mathbf{\Gamma}_{i_1;i_2} - \mathbf{C}_{\tilde{i_1}}\big|_{\bigcirc}$$

$$\underset{\{inF\}}{\overset{(47)_{i_1}}{\approx}} \mathbf{\Gamma}_{i_1;i_2} - \frac{1}{2}\left[\left(\sum_{f=1}^{F-1}(-1)^{f-1}\mathbf{\Gamma}_{i_f;i_{f+1}}\right)+\mathbf{\Gamma}_{i_F;i_1}\right]$$

$$= \frac{1}{2}\left[\left(\sum_{f=2}^{F-1}(-1)^{f-2}\mathbf{\Gamma}_{i_f;i_{f+1}}\right)-\mathbf{\Gamma}_{i_F;i_1}+\mathbf{\Gamma}_{i_1;i_2}\right] \underset{\{inF\}}{\overset{(47)_{i_2}}{\approx}} \mathbf{C}_{\tilde{i_2}}\big|_{\bigcirc} \qquad , \forall \; F \; uneven \; \wedge \; 3 \leq F \leq I$$

  The consistency between direct and sequential error covariance estimates results directly from their common underlying definition of residual covariances in Eq. (20) and holds not only for the approximate formulations but similarly for the full expressions including error dependencies (compare Sect. 3.3.1). "

The remaining modification are as follows:

- reformulation in abstract (ll.10-12): " The presented generalized estimation of full error covariance- and cross-covariance matrices between dataset does not necessarily accumulate uncertainties of assumptions among error estimations of multiple datasets. "

- reformulation of the general introduction of Sec.4 (ll.322-324): " An extension for more than three datasets based on a minimal number of assumptions is introduced in Sect. 4.2. It includes the estimation of additional error covariances, either directly or sequentially, and some error cross-statistics to estimate a maximum amount of error statistics. "
  and (ll.327-328): " ... and (iii) the comparison of the approximations from direct and sequential estimates (Sect. 4.2.5). "

- reformulate introduction of Sec.4.2 (ll.401-407): " As described in Sect. 2, $A_I > 0$ gives the number of error cross-statistics which can potentially be estimated in addition to all error covariances for $I > 3$ datasets. Consequentially, the independent assumption between all pairs of datasets can be relaxed to

a "partial independence assumption" where one independent dataset-pair is required for each dataset I. The estimation of error covariances can be generalized in two ways: Firstly, the direct formulation for three datasets in Sect. 4.1.1 is generalized to a direct estimation of more than three datasets in Sect. 4.2.1. Secondly, Sect. 4.2.2 introduces the sequential estimation of error covariances of any additional dataset. This estimation procedure of additional error covariances is denoted as "sequential estimation" because is requires the error covariance estimate of a prior dataset, in contrast to the "direct estimation" from an independent triplet of datasets ("triangular estimation" in Sect. 4.1) or generally from a closed series of pairwise-independent datasets ("polygonal estimation" in Sect. 4.2.1). "

- generalized formulation of the use of a basis pentagon as basis for the sequential covariance estimate in Sec.4.2.2 (ll.436-438): " Similarly, a "basic polygon" can be defined from a closed series of F pairwise-independent datasets, where the referring error covariances can be directly estimated from Eq. (48). "

- moving aspects of the sequential estimation from the general introduction of Sec.4.2 to Sec.4.2.2 (l.439-442): " For each additional dataset $i$ with $F < i < I$, its cross-statistics to one prior dataset $ref(i) < i$ is needed to be assumed in order to close the problem. This prior dataset $ref(i)$ is denoted as "reference dataset" of dataset $i$. With this, the remaining error covariances can be estimated from residual covariances under the partial independence assumption $\mathbf{X}_{\tilde{i};r\tilde{e}f(i)} = 0$. "

- use index "G" instead of "F" for the sequential uncertainty of error covaiances to avoid confusion (ll.477-480)

- note on generalization of comparison to sequential estimate in Sec.4.2.5 (ll.527-529): _This holds similarly for any polygonal estimation, where the additional independence assumption which closes the series of pairwise-independent datasets has to be of similar accuracy as the other independent assumptions._ "

- generalization formulation of the 1st rule for the setup of datasets in Sec. 6.1 (ll.709-711): " all error cross-statistics along a closed series of dataset-pairs, with the number of involved datasets uneven and larger or equal three, are needed (this closed series of datasets is called "basic polygon" or "basic triangle" in case of three datasets), and ... "

- addition of a basic pentagon in the illustrative example: Fig.5(plot + title + title of Fig.6 accordingly) + (ll.721-722): " Alternatively, the basic triangle could be replaced e.g. by a basic polygon of five datasets ("basic pentagon": 1;2;3;5;4), if the dependency(3;5) is assumed instead of the dependency(3;1). "

- generalization of the general rules for the selection of a setup: Sec.6.2 (ll.733-735): " Because uncertainties in error estimate do not necessarily sum up for a large basic polynom or along a branch of the independence tree (compare Sect. 4.2.4), a large residual-to-dependency ratio w.r.t. of the assumed cross-statistics is more important than a low number of intermediate datasets. "

- generalization of Algorithm for residual covariances in Apx.A: Alg.A1(title and algorithm) + title of Alg.A2 + (ll.817-818): " In this algorithm, the generalized formulation of a basic polygon of $F \leq I$ residuals, for any uneven $F \geq 3$, is used for the estimation of the first error covariance. "
+ (ll.821-822): " This algorithm uses a basic triangle as example for a basic polygon for the estimation first error covariance. " + (ll.831-832): " Note that the generalized basic polygon can also be used for the estimation of the first error covariance in Algorithm 2. "

- (minor) addition of the word "sequential" when refering to sequential error covariance estimates in contrast to the generalized direct estimates: Sec.4.2.2 (l.433) + (l.439) + Sec.4.2.4 (l.469) + (l.473) + (l.483)

- (minor) replacement of "3" direct estimates by "F": Sec.4.2.4 (l.469) + Sec.4.2.5 (l.516)

- (minor) replacement of "basic triangle" to "basic polygon" and "independent triangle" by "pariwise-independent polygon": Sec.4.2.4 (l.476) + (ll.478-479) + Sec.4.2.5 (ll.514-515) + (l.518) + Sec.6.2 (ll.745-746) + (l.748) + (l.750) + (ll.752-754) + (l.758) + (l.760) + (l.752) + Apx.A (l.810)

6. **L. 552, ...** _without introducing additional degrees of freedom._ **That formulation is confusing. From what I understand, the purpose is fundamentally to eliminate degrees of freedom by**

introducing hypotheses that render exactly determined the problem of estimating the error statistics. It seems that you implicitly anticipate on the text (ll. 602-608) where it is suggested to combine different estimates.

Reply: This information was added to emphasize the following: If an assumed error statistic is formulated as function of an additional unknown, this additional unknown cannot be determined in the estimation scheme as it induces an additional degree-of-freedom which makes the problem underdetermined. We see that this formulation might actually be confusing here, we removed it form the sentence (ll.681-682):
" The only restriction is that all assumed error statistics must be fully determined by other error statistics or predefined values. "

7. **Ll. 551-552, *The only restriction is that all assumed error statistics must be fully determined* ... From what I understand, the wording all additional assumed error statistics would be more appropriate.**

Reply: In this manuscript the word "additional" is referring to all datasets which are not part of the basic triangle (or basic poylgon). In this particular discussion it would also be possible to formulate e.g. an assumed error cross-covariance between two datasets in the basic triangle as function of their error covariances. Thus this sentence refers to all assumed error statistics, and not only the "additional" ones. Therefor, we decided to keep the current formulation.

8. **Concerning positive definiteness of the estimated error statistics, the authors write (ll. 393-394) *However, the generalization to covariances matrices is expected to increase the occurrence of negative values were* (incidentally, the proper spelling is *where*) *correlations between two entries of the state are low, thus relative differences and sampling errors become large.* I am not sure I understand what that means. Is it that the occurrence of non definite positiveness is likely to increase with the number I datasets, or what ?**

Reply: This refers to the fact that - in comparison to scalar variances at each location - covariance matrices often contain small values where the spatial correlation between eg two spatially distant locations is small. This results in a small covariance between the two locations, and uncertainties in the assumptions (eg independence) or sampling noise can become larger than the actual true covariance which may lead to negative values in the estimated covariance at these locations. We reformulated the sentence for clarification (ll.406-408):
" However, the generalization to covariances matrices is expected to increase the occurrence of negative values in off-diagonal elements. Because spatial correlations and thus true covariances may become small compared to uncertainties in the assumptions or sampling noise, estimated covariances at these locations might become negative. "

9. **L. 370, *All three estimates become equivalent if the residual cross-covariances are symmetric* Yes, but it might be useful to mention here that these equivalent estimates may not be positive definite.**
**And l. 388, *Estimated error covariances might even contain negative values* ... You must mean *might even not be positive definite* ... .**

Reply: We added a refering note (ll.384-385):
" However, non of the estimates ensures positive definiteness of the error covariances. "

And reformulated as suggested (l.401):
" Estimated error covariances might even not be positive definite if ... "

10. **Ll. 626-627, *While the presented method ensures symmetry of error covariances*, Not necessarily (Eqs 37 or 38 do not ensure symmetry). Say that symmetry can be enforced if necessary.**

Reply: Right. We reformulated for clarification (l.785):
" While the presented method can be formulated to provide symmetric error covariances, ... "

Additionally, we also added this point in the description of the Algorithms in Apx.A (ll.827-831):
*" The decision to estimate error statistics from residual covariances (Algorithm A1) or cross-covariances (Algorithm A2) depends on the availability of residual statistics, the need for symmetric estimations of error covariances – which is only intrinsically guaranteed in Algorithm A1 –, and the need for estimating asymmetric error cross-covariances – which can only be estimated with Algorithm A2 (compare Sect. 3.3.1). "*

11. **L. 2 (abstract), ... *an ill-posed problem* ... Not *ill-posed*, but *underdetermined***

    Reply: Replaced as suggested.

12. **Ll. 325-326, *"assumption of independence"*. I mention that the word independence is not used here in its accepted standard meaning in probability theory. The hypothesis made here is actually an hypothesis of no correlation, which is a weaker property than independence. I suggest the authors briefly mention that fact.**
    **And l. 325, *equals* is to be changed to *implies***

    Reply: Thank you for this important point. We reformulated the sentence for clarification and replaced the word as suggested (ll.337-338):
    *" Because the assumption of zero cross-covariance implies zero error correlation which is often used as proxy for independence, it is denoted as "assumption of independence" or "independence assumption" thereafter. "*

13. **Ll. 440-441, ... *absolute uncertainties of estimations from residual covariances and cross-covariances differ only in the uncertainties w.r.t. the basic triangle* ... Well, these uncertainties depend also in the uncertainty in the hypothesis $\mathbf{D}_{i;ref(i)} = 0$.**

    Reply: That's right, the mentioned uncertainties include also $\mathbf{D}_{i;ref(i)} = 0$. However, the uncertainty induced by this assumption is the same among the two estimations (Eq. (54),(55)). The sentence refers only to the differences of the two estimations: "absolute uncertainties ... differ only in ...".

14. **L. 315, *the determination of uncertainties resulting from possible errors in the  assumed error statistics***

    Reply: Corrected (l.327).

15. **L. 201, what are unbiased error statistics ? Error statistics are here the unknowns, not the data, and whether they are biased or not makes a priori no sense.**

    Reply: Corrected to "unbiased datasets" (l.201).

16. **L. 323, $\mathbf{X}_{i;j} = 0 \Leftrightarrow \mathbf{D}_{i;j} = 0$ to be changed to $\mathbf{X}_{i;j} = 0 \Rightarrow \mathbf{D}_{i;j} = 0$**
    **L. 414, similarly, the condition $\mathbf{X}_{i;ref(j)} = 0$ is actually stronger than the assumption $\mathbf{D}_{i;ref(i)} = 0$ made on l. 405. Be more consistent as to which hypotheses you make.**

    Reply: Thank you for pointing this out. We corrected l.335 (new count) as proposed and replaced $\mathbf{D}_{i;ref(i)}$ by $\mathbf{X}_{i;ref(i)}$ in l.442 (new count) as well as l.520 (new count). In addition, the description of the "independence assumption" was adopted accordingly, now referring to vanishing error cross-covariance $\mathbf{X}_{i;ref(i)}$ (compare comment 12 above). With this correction, the subsequent formulations e.g. in l.343 (new count) and l.444 (new count), remain correct. We checked the consistency thought the whole manuscript. The "independence assumption" now refers solely to vanishing error cross-covariance $\mathbf{X}_{i;j}$ which induces vanishing error dependencies $\mathbf{D}_{i;j}$.

17. **L. 558, $I \leq 3$, you must mean $I > 3$ ?**

    Reply: Replaced by $I \geq 3$. Thank you.

18. **L. 318, $U_I \geq 0 \rightarrow D_I \geq 0$**

    Reply: Corrected.

19. **L. 352, ... _four equivalent formulations for each pair of other datasets ... ._ You have assumed here I = 3, so that there is only one pair of 'other datasets' (see also l. 358).**

Reply: Right, this is an artefact from a more general description. We removed "for each pair of other datasets" in both cases (l.363, l.370):
_" ... every error covariance from residual cross-covariances has four equivalent formulations which provide the same result in the exact case ... "_
_" Equations (36) to (38) provide three different estimates of an error covariance matrix. "_

20. **L. 34, what are exactly the _corners_ ?**

Reply: The "corners" refer to the three datasets when used in the (G)3CH method. We see that this remains unclear for readers who are not fimiliar with this method. We reformulated avoiding the used of the word "corners" (ll.34-35):
_" They show that when the G3CH method is applied to the observations, background and analysis of variational assimilation procedures, ... "_

21. **Ll. 40-41, sentence starting _Up to now,_ ... awkward. From what I understand, I suggest _Up to now, only scalar error variance estimation has been implemented in data assimilation with the TC method (e.g._ ...**

Reply: Thank you, we reformulated based on the reviewers suggestion (ll.40-41):
_" Up to now, there are only a few applications of scalar error variance estimation in data assimilation with the TC method ... "_

22. **The word _exemplary_ is used mistakenly in several places (the word designates something that is meant to be imitated, while the authors obviously think of illustrative examples)**
**L. 543, ... _including an exemplary visualisation,_ → ... _including an illustrative visualisation,_**
**L. 566, _An exemplary setup_ ... → _An illustrative example of setup_ ...**
**Caption of Figure 5, _An illustrative example of visualization_ ...**
**See also l. 7 (abstract)**

Reply: Corrected at all 4 locations.

23. **Ll. 35-36, ... _this particular error estimation problem can only be closed under the assumption of optimally._ I do not understand what this means (and the proper wording should in any case be assumption of optimality)**

Reply: The optimality (not optimally) refers to the analysis being optimal, which is a common assumption in a posteriori evaluation methods in data assimilation. We reformulated for clarification (ll.35-36):
_" ... this particular error estimation problem can only be closed under the assumption that the analysis is optimal. "_

24. **Ll. 54-55 (and later) _error cross-variances_ ... You must mean _error covariances_ ?**

Reply: According to the notation and definition of this manuscript, the error interaction between two scalar datasets is denoted as "cross-variance" because "cross-" always refers to the interaction between datasets (like residual or error cross-covariance, ll.150-156) whereas "variances" are the diagonal elements of "covariances". In the scalar case, which was e.g. formulated by Zwieback et al 2012, all covariances reduce to scalar variances. Thus, the term "cross-variances" is used when refering to scalar error interactions in order to be consistent with the notation in the rest of this manuscript. We extended the sentence to clarify this difference in wording (ll.54-55) which should also avoid misunderstanding lateron:
_" Zwieback et al.,2012 were the first proposing the additional estimation of the scalar error cross-variances between two selected datasets (which they denote as covariances) ... "_

25. **Ll. 65-66, ... *affect different formulations of the estimated error statistics?* → ... *affect different estimations of the error statistics?***

    Reply: Corrected.

26. **L. 613, I understand the N-CN *method* is what was called previously the *N-cornered hat method* (l. 603), or what ?**

    Reply: We assume that the reviewer refers to the use of "N-CH method" which is indeed the short form of the "N-cornered hat method". We added the abbreviation where the method was first mentioned (l.760) and the full name where the abbreviation is first mentioned in the conclusions (l.770).

27. **L. 24, *arises* → *raises***

    Reply: Corrected.

28. **L. 588, *effect* → *affect* (or *impact*)**

    Reply: Corrected to "affect".

29. **L. 589, *triple* → *triplet* (the same correction is to be made elsewhere, e.g. l. 407 ; please check)**

    Reply: Corrected everywhere, where it appeared.

30. **Ll. 574-575, ... *can be interpreted as being similar to* ...**

    Reply: Corrected.

31. **L. 583, ... *is comparably well known,* ... word *comparably* inappropriate here. I suggest ... *is known to some degree of accuracy,* ......**

    Reply: Corrected.

32. **L. 544, ... *algorithmic summary for the calculation* ...**

    Reply: Corrected.

    **Although the authors have obviously been very careful with their notations, in particular as concerns indices, I have noticed a few typos**

33. **Ll. 177 and 184, $\mathbf{D}_{i;j}$ should be replaced with $\mathbf{D}_{i-j;k-l}$ and $\mathbf{Y}_{i-j;k-l}$ respectively**

    Reply: Corrected, thank you! (l.177, 184)

34. **Eq. (32), (22) above first = sign should rather be (30)**

    Reply: Eq. (32) can be derived either from using Eq. (32) (new version = (29) old verion), (33) (new version = (30) old verion) or (23) (new version = (22) old version). We replaced Eq. (22) by both, Eq. (32) and (33) because each of them refers to one of the innovation cross-covariances used here (see Eq.(35) in new version).

35. **L. 442, $m_{f-1} \rightarrow m_{F-1}$**

    Reply: Corrected, now $m_{G-1}$ after changing index according to comment 5 - reply part B above. (l.480)

**And finally, it is the first time I have seen superscripts above equality signs to give reference to previous equations. I think that can be useful, and it is undoubtedly in the present case.**

Reply: Thank you for the supporting comment.

---

## Author Response (AR3)

Response to Report 1 (reviewer 2):

The authors have significantly modified their paper, and have taken my comments and suggestions into account. They have generalized their 'triangular' approach to estimation in the form of an interesting 'polynomial' approach (none of my suggestions). I think the paper is now acceptable for publication. I nevertheless suggest a number of modifications of mostly editing character which should not require substantial work.

General Reply: We are grateful for the thoughtful re-evaluation and valuable remarks.

1. From a purely scientific point of view, I have only one request for correction which, without being critical for the paper, must be made.

   The authors write, concerning positive definiteness (ll. 406-407) ... *the generalization to covariances matrices is expected to increase the occurrence of negative values in off-diagonal elements.* Positive definiteness is not a question of the sign of off-diagonal elements. It is that a covariance matrix, in addition to being symmetric (and to have as such real eigenvalues) must have only positive eigenvalues (or nonnegative ones, depending on the precise definition one takes). For instance, the 2-2 matrix

   | 1 | a |
   |---|---|
   | a | 1 |

   is definite positive if $a^2 < 1$, and is not if $a^2 > 1$, independent of the sign of $a$.

   Reply: Thank you for pointing this out. The paragraph was actually meant to be related to the occurrence of negative elements in the covariance matrices which we incidentally mixed up with positive definiteness. We removed the mentioning of positive definiteness in the first sentence of this paragraph (underlined below) and deleted the sentence where it was mentioned again. The rest of the paragraph was left unchanged (ll. 401-407, compare also corrections to comment 5 below):

   " *Estimated error covariances* *might even contain negative values* *if error dependencies are large compared to the true error covariance of a dataset. If the true error covariances differ significantly among highly correlated datasets, the neglected error dependency between two datasets might become much larger than the smaller error covariance, e.g.* $\Delta\mathbf{D}_{\tilde{k};\tilde{i}} - \Delta\mathbf{D}_{\tilde{j};\tilde{k}} \approx 0$, $\frac{1}{2}\Delta\mathbf{D}_{\tilde{i};\tilde{j}} > \mathbf{C}_{\tilde{i}}\big|_{true}$. *This phenomena was also described and demonstrated by Sjoberg et al. (2021) for scalar problems. However, the generalization to covariances matrices is expected to increase the occurrence of negative values in off-diagonal elements. Because spatial correlations and thus true covariances may become small compared to uncertainties in the assumptions or sampling noise, estimated error covariances at these locations might become negative.* "

   Along with the same line of ideas, I make the following remarks

   1a. L. 727, I think the need for positive definiteness will always we there, independently of the purpose or use (using a 'covariance matrix' that is not positive definite can lead to absurd results, for instance in data assimilation)

   Reply: Agreed. We removed the item, the sentence now reads (ll. 738-739):

   " *The optimal selection is specific for each application and may depend on several requirements related to the actual purpose or use (e.g. available knowledge, accuracy of each estimate).* "

1b. **And I suggest to change the words** *positive definiteness might not be fulfilled* (ll. 785-786) **to** *there remains the risk that positive definiteness might not be fulfilled*

Reply: In alignment with the main reply to comment 1 above, we replaced the use of "positive definiteness" by "negative values". The sentence reads now (ll. 796-797):

*" . . . there remains a risk that negative values might occur for real applications due to inaccurate assumptions or sampling uncertainties. "*

2. **L. 685, . . .** *not every possible combination of error statistics to be estimated provides a solution.* **This statement, which is fundamental for the paper, is mentioned explicitly only in the conclusions. It should be mentioned in the introduction (for instance between ll. 47 and 48 or the present version, or after l. 67), with appropriate references (possibly Vogelsang and Stoffelen, 2001, or Gruber et al., 2016).**

   **In addition, from I understand, a better formulation would be . . .** *not every possible a priori choice of error statistics to be estimated provides a solution.*

Reply: We agree that this fundamental aspect needs to be stressed clearly in the manuscript. Concerning the introduction, there is already a sentence mentioning that this point was observed by Vogelsang and Stoffelen,2021, however formulated slightly differently (ll. 58-59): *" They observed that the problem can not be solved for all possible combinations of cross-variances to be estimated. "*

We also modified the formulation based on the reviewers suggestion and added the reference to Vogelzang and Stoffelen,2021 also here (ll. 696-698):

*" However not every possible choice of error statistics to be estimated provides a solution, which was also observed by Vogelzang and Stoffelen (2021) in the scalar case. "*

**On the number of occasions, the wording is improper, imprecise or awkward**

3. **Ll. 71-72, . . .** *the exact formulations of error statistics in Sect. 3.3 remain underdetermined . . .* **to be changes to . . .** *the complete estimation of error statistics in Sect. 3.3 remains underdetermined . . .* **(it is the estimation as a whole that is underdetermined in a real situation, not the exact estimation)**

Reply: We agree, but we decided to keep the expression "exact formulation" here because it is the title of the section that is referred to. We reformulated the expression based on the reviewers remark to make clear that it refers to the complete estimation (ll. 71):

*" While the exact formulation for estimating error statistics in Sect.3.3 . . . "*

4. **Ll. 323-324, sentence starting** *It includes . . .* **seems awkward to me. From what I understand, I suggest** *In order to achieve a complete estimation of error statistics, the Section includes the estimation, either direct or sequential, of additional error covariances, as well as of some error cross-statistics* **(although it seems to me that the words** *additional* **and** *some* **are not here necessary, since it as the whole of error statistics that are in the end to be estimated. But I may be mistaken)**

Reply: The sentence was modified based on the reviewers suggestion, but skip the first part of the suggested sentence because we think that is is not essential. It reads now (ll. 323-324):

*" It includes the estimation, either direct or sequential, of additional error covariances, as well as of some error cross-statistics. "*

We use the words "additional" for error statistics of all datasets which are not part of the basic polygon thought the manuscript, and so also here. The word "some" refers to the fact that only a

part of the error cross-covariances can be estimated, in contrast to the ones that have to be assumed a priori.

5. **L. 326, *the analysis of differences from residual covariance- and cross-covariance estimates,* ... The formulation might be confusing. I think it would be preferable to restrict the word *estimate* to estimates of the error statistics, not to the residual statistics, which are directly computed form the data and which (contrary to the estimation of the error statistics) do not require *a priori* hypotheses. I suggest to write simply *the analysis of differences from residual covariances and cross-covariances.***

**A similar change may have to be made elsewhere. Please check.**

Reply: It refers to the differences of error estimates which are achieved either from residual covariances or residual cross-covariances, which was wrongly formulated. We corrected the sentence accordingly (ll. 326-327):
" *(i) the analysis of differences between error estimates from residual covariances and cross-covariances (Sect. 4.1.2) ...* "

Additionally, we checked the rest of the document and added the word "error" for estimated error statistics at some locations for clarification: l. 407, 475, 489

6. **L. 337, ... *the assumption of zero error cross-covariance* ...**

Reply: Corrected.

7. **Ll. 384-385, ... *none of the estimates ensures positive definiteness of the estimated error covariances.***

Reply: Corrected.

8. **Ll. 415-416, I suggest *As described in Sect. 2, for $I > 3$ datasets, $A_I > 0$ gives the number of error cross-statistics which can potentially be a priori assumed in addition to all error coviariances***

Reply: As described in the replay to comment 5 above (and in accordance to the reviewers suggestion in comment 5), the word "estimated" is used for error statistics which are estimated with the proposed method, in contrast to error statistics which are need to be assumed a priori and a priori computed residual statistics. We therefore keep the current words "estimated" instead of "a priori assumed" (reviewers suggestion). Except that, we reformulated the sentence as suggested (ll. 414-415):
" *As described in Sect. 2, for $I > 3$ datasets, $A_I > 0$ gives the number of error cross-statistics which can potentially be estimated in addition to all error covariances.* "

9. **L. 437, ... *pairwise-independent datasets* ..., do you mean two successive datasets in the series are independent, or more than that ?**

Reply: Yes, but additionally the last and first elements have to be independent to "close" the series, we added the description in the sentence (ll. 435-437):
" *Similarly, a "basic polygon" can be defined from a closed series of F pairwise-independent datasets, where each two successive datasets in the series as well as the last and first element are independent from each other (compare Sect. 4.2.1).* "

10. **L. 437 again, … *where the referring error covariances* … What to you mean here by *referring* ?**

    Reply: We reformulated consistent with the prior comment for clarification (ll. 437-438):
    *" Then, the error covariance of each dataset in the series can be directly estimated from Eq. (48). "*

11. **L. 464, … *polygonal error covariance* …**

    Reply: Corrected (l. 464). We also checked the rest of the document for the incidental use of the word "polynom\*" and corrected it accordingly (l. 758).

12. **L. 523-524, sentence starting *The sequential estimation* … makes no real sense to me. In any case, there is a useless repetition of the words *error covariance*, and the proper wording is *similar … to* (not *similar … as*)**

    Reply: We removed the incidental repetition of the words "error covariance" and reformulated for clarification (ll. 523-524):
    *" The sequential estimation of an error covariance becomes favourable if the error covariance estimate of its reference dataset is as least as accurate as the assumed dependency between these two datasets $\left(\Delta\mathbf{C}_{\bar{j}} \to \Delta\mathbf{D}_{\bar{i};\bar{j}}\right)$.*
    *"*

13. **L. 546, … *the true error statistics* … Ambiguous. The *a priori* assumed error statistics, or the empirical statistics obtained from the 20,000-sample ? The difference may be small in the present case, but I presume you mean the latter, which would be the only ones available in a real simulation. And they would not be *error statistics* but *residual statistics*.**

    Reply: We understand the reviewers concern regarding the "truth" of error statistics. This sentence indeed refers to the fact that the presented experiments are based on artificial data with predefined error statistics. In contrast to real applications where only residual error statistics can be calculated from the available data, the residual covariances were here calculated by these predefined error statistics. We understand that this was not formulated clear enough in the manuscript and added the following sentences in the introduction of the experiments for clarification (ll. 537-542):
    *" The experiments use predefined error statistics which are artificially generated to fulfill certain properties concerning error covariances and dependencies. Although also being generated by a finite sample of 20,000 realizations, these predefined error statistics are used to calculate residual statistics and thus represent the true error statistics that would be unknown in real applications. Here, the artificial generation of sampled true error statistics – denoted as "true error statistics" hereafter – allows for an evaluation of uncertainties of the "estimated error statistics" that are estimated with the proposed method. "*

14. **Ll. 565-566, … *the independent assumption in the basic triangle(1;2;3) is an accurate approximation of the true error dependencies.* Well, have not the true error dependencies assumed to be zero by construction ?**

    Reply: Indeed, the true error dependencies are zero by construction. We reformulated the sentence to state this more clearly (ll. 568-570):
    *" In the first experiment in Sect. 5.1, the true error dependencies are constructed to fulfill the independent assumption in the basic triangle$(1; 2; 3)$. "*

    Please note that the true error statistics are constructed in these experiments and don't include any assumption in contrast to the "assumed" estimations of those statistics.

15. **L. 579, ... *true error dependencies are occurring* ... → ... *true error dependencies are used* ...**

    Reply: Replaced with "are generated" to stress the fact that the true statistics are artificially generated in these experiments (l. 583).

16. **L. 591, ... *This neglected positive error dependency* ... In what is that neglected error dependency positive ?**

    Reply: We removed the word "positive" to avoid confusion (l. 595).

17. **L. 594-595, ... *these three matrices are underestimated w.r.t. the true statistics by the half of the neglected error dependency matrix* ... The these matrices are underestimated by half of the neglected error dependency matrix is not visible from the figure. Say it clearly, and say how you know it. Is it consistent with error estimates that have been given in Section 4 ? If yes, say it clearly and make reference to the relevant equations. If not, explain.**
    **A similar comment applies to the text ll. 615-617.**

    Reply: We see that this point is not sufficiently described in the manuscript. The fact the matrices are underestimated by half the neglected error dependency can actually be seen from the equations as well as in the figure – as good as different values can be specified by the colortable. However we agree that this is not obvious for the reader and some more description is needed to guide the reader to see that conclusion. We reformulated the two paragraphs to provide more visual guidance and state more explicitly the relation to the equations in Sect.4. (ll. 596-600):
    *" This agrees with the experimental results shown in Fig. 2b where the neglected error dependency$(2;4)$ with diagonal values around 1.2 (orange colors in upper-left part of 1st row, column 5) induces an absolute uncertainty of the estimated error covariance 4 with diagonal values around 0.6 (purple colors in 3rd row, column 4). The sign of the uncertainty which corresponds to the underestimation can be seen from comparing the true and estimated error covariances matrices of dataset 4 (2nd row, column 4). "*

    And similarly (ll.624-629):
    *" For the two datasets involved 2 and 3, the neglected positive dependency$(2;3)$ is transferred with the same sign, leading to an underestimation of their error covariances (row 2, column 2-3). In contrast, the impact on the error covariance of the remaining dataset in the triangle 1 is reversed, leading to an overestimation of the true error covariance (2nd row, column 1). As expected from Eq. (45), the magnitude of uncertainty of the three estimated error covariances with diagonal elements around 0.4 (light purple colors in 3rd row, colomn 1-3) is half the neglected error dependency with diagonal elements around 0.2 (dark purple colors in upper-left part of 1st row, column 3). "*

18. **L. 597-598, Sentence starting *This experiment demonstrates* ... Move the word *accurately* to ... *to estimate accurately.***

    Reply: Corrected.

19. **L. 598, ... *accurate reproduction* → *accurate estimation***

    Reply: Corrected.

20. **L. 606 ... *a true dependency* ... → ... *a non-zero dependency* ...**
    Reply: Corrected.

21. **Similarly, l. 607,** ... *three true error dependencies* ... → ... *three non-zero error dependencies*

    Reply: Corrected, also in a similar case in l.630.

22. **L. 830,** ... *the need for estimating possible asymmetric error cross-covariances* ... **(cross-covariance matrices, if they are square in the first place, will in general be asymmetric, but there is no need for them to be so)**

    Reply: Yes. Here, we just wnated to point out that cross-covariances are (or may be) asymmetric statistics, and these asymmetric components cannot be estimated from residual covariances (as described in Sec.3.3.1). We added this information for clarification (l. 841):
    " ... *the need for estimating asymmetric components of error cross-covariances* ... "

**The English, although perfectly understandable in most places, is occasionally incorrect**

23. **In may places (starting on l. 272) the word** *uneven* **is used to quantify an integer that is not even. The proper word is** *odd*

    Reply: Yes, thank you! We replaced it everywhere in the manuscript.

24. **L. 30,** *This method has been widely used* ...

    Reply: Corrected.

25. **L. 48,** ... *the estimation [. . . ] has been well-established for decades,* ...

    Reply: Corrected.

26. **L. 54,** ... *were the first to propose the additional estimation* ...

    Reply: Corrected.

27. **L. 338,** *thereafter* → *hereafter*

    Reply: Corrected.

28. **L. 498,** *partly* → *partially*

    Reply: Corrected.

29. **L. 534,** ... *the capabilities for estimating* ...

    Reply: Corrected.

30. **L. 651,** *If the other two error covariances 1 and 2 had also been estimated*

    Reply: Corrected.

**Additional corrections to be made**

31. **L. 2,** ... *is an underdetermined problem* ...
    Reply: Corrected.

32. **L. 70, word** *respectively* **seems here useless**
    Reply: Yes, removed.

33. **L. 96, l. 189,** ... *as functionS of* ...
    Reply: Corrected at both locations.

34. **L. 461,** ... *each subsequent datasetS* $j > i$ ... **(singular)**
    Reply: Corrected.

35. **L. 477,** ... *and * $m_{G-1}$ *the reference of* ...
    Reply: Corrected.

36. **L. 546, suppress the word** *be*
    Reply: Done.

37. **L. 556,** ... *to estimate the error covariance* ...
    Reply: Corrected.

38. **L. 626,** ... *the uncertainty in the two estimated error dependencies* ...
    Reply: Corrected.

39. **L. 526,** ... *of the other two error dependencies* ...
    Reply: Corrected.

40. **L. 724,** *datasets*
    Reply: Corrected.

41. **Ll. 767-768,** ... *a flexible setup* ...
    Reply: Corrected.

42. **L. 807, Title of Appendix** *Algorithms* **(plural)**
    Reply: Corrected.

43. **L. 201,** ... *or unbiased datasets* ... **It has already been said (l. 148) that unbiasedness would be assumed in the paper. Unless the authors want to stress particularly that point (and, if so, they must do it explicitly), this kind of repetition is useless, and may even be confusing.**
    Reply: Agreed. We removed the part on unibased datsets; it reads now (l. 201):
    *" ... without any further assumption like independent errors. "*

In addition to the reviewers suggestions above, we corrected the abbreviation "Sec." by "Sect." which occurred in Sect.5-6.

---

## Author Response (AR4)

Response to Report 1 (reviewer 2):

I regret I still have an objection against publication of this new version of the paper. It is relative to the first point I raised in my previous review, concerning the positive definite character of the covariance matrices obtained through the estimation approach described in the paper. The authors have not in my opinion correctly responded to my comment, and I actually suspect there is a misunderstanding between us. As I said in my previous review, this point is not critical for the paper, but it is nevertheless in a sense fundamental, and the paper should not mislead potential readers.

The authors are (rightly) concerned by the possibility that the 'variance' of components of the dataset vectors (or of some linear combinations of those components) might become negative. They write (ll. 404-407) *However, the generalization to covariances matrices is expected to increase the occurrence of negative values in off-diagonal elements. Because spatial correlations and thus true covariances may become small compared to uncertainties in the assumptions or sampling noise, estimated error covariances at these locations might become negative.*

This seems to imply that it is the occurrence of negative off-diagonal elements in a 'covariance' matrix which might produce negative 'variances'. That is a misconception. Taking again the example of the 2x2 matrix considered in my previous review,

$$A \equiv \begin{array}{|c|c|} \hline 1 & a \\ \hline a & 1 \\ \hline \end{array}$$

If $A$ is to be the covariance matrix of a 2-vector $(x_1, x_2)^T$ with zero expectation, then necessarily

$$E(x_1^2) = E(x_2^2) = 1, \quad E(x_1 x_2) = a \tag{1}$$

where $E(.)$ denotes statistical expectation.

For any $\beta_1$ and $\beta_2$, the quantity $q = \beta_1 x_1 + \beta_2 x_2$ has variance

$$E[(\beta_1 x_1 + \beta_2 x_2)^2] = \beta_1^2 + \beta_2^2 + 2a\beta_1\beta_2 \tag{2}$$

which must be $\geq 0$ for any $(\beta_1, \beta_2)$. This is verified iff $a^2 \leq 1$ (with equality if $a^2 = 1$), independently of the sign of a.

It is the signs of the eigenvalues of a symmetric matrix that matters for the matrix to be a covariance matrix or not, not the signs of the entries in the matrix. The matrix $A$ has eigenvalues $1 + a$ and $1 - a$, both of which are $\geq 0$ if $a_2 \leq 1$, while one is negative if $a^2 > 1$. The matrix is symmetric non-negative for $a^2 \leq 1$, and strictly positive definite if $a^2 < 1$.

I have not looked at the connection of the above with the inequality on ll. 403-404 of the paper, but the text that follows is erroneous, at least in part. The correction must be made in the paper.

It is true that the expressions *symmetric definite positive matrices* or *symmetric nonnegative matrices* can be misleading in that they might be understood as referring to the signs of the entries in the matrices. But the established vocabulary of linear algebra is unambiguous, and these expressions refer to the signs of the eigenvalues of the matrices.

I really do not think I am mistaken in the above, but if the authors think I am, and that it is the signs of the entries in the covariance matrices that matter, I am ready to look at their objections.

And it remains of course that nothing in the paper guarantees that the obtained 'covariance matrices' are definite positive (or even non-negative)

Reply: Thank you very much for the clarification. In this paragraph, we intent to only discuss the sign of individual elements in the covariance matrix (only referring to non-negative elements, not positive definiteness of covariance matrix). But we agree that the important aspect for a covariance matrix is the positive definiteness (determined by the sign of the eigenvalues) rather than the sign of individual elements. We therefore added a note on positive definiteness based on the reviewers explanations (ll.407-408):
" *However, the occurrence of negative elements does not affect the positive definiteness of a covariance matrix, which is determined by the sign of its eigenvalues.* "

Note that the not guaranteed positive definiteness was mentioned shorty before (ll.384-385) so we decided to not repeat it here.

Concerning the other comments and suggestions in my previous review, the authors have responded to them correctly. But I still have a few suggestions for editing corrections.

- **L. 294, text is awkward. From what I understand, I suggest ... *results from setting to the same value both elements of one pair of datasets in the expression (22) of residual crosscovariances.***

  Reply: Corrected based on the reviewers suggestion (l.294):
  " *Each of the four possibilities results from setting both indices of one pair of datasets in definition of residual cross-covariances in Eq. (22) to the same value.* "

- **L. 456, ... *including their asymmetric components.***

  Reply: Corrected as suggested (l.457).

- **L. 527, text is awkward. Change to ... *has accuracy similar to that of the other two assumptions.***

  Reply: Corrected based on the reviewers suggestion (ll.527-528):
  " *... if the accuracy of the additional independence assumption is similar to that of the other two assumptions.* "

- **L. 545, *The four datasets induces 10 error statistics ... → The statistics of the four datasets depend on 10 error statistical moments***

  Reply: Corrected based on the reviewers suggestion, but we decided to avoid the word "statistical moments" to avoid potential confusion as this term is not used elsewhere in the manuscript (ll.546):
  " *The error statistics of the four datasets consist of 10 matrices ...* "

In addition to the reviewers suggestions above, we performed two purely technical corrections in the introduction: comma added before *as well as* in l.7, and plural of *datasets* in l.11 .